# Analysis of Bootstrap and Subsampling
# in High-dimensional Regularized Regression

**Lucas Clarté**[1]   **Adrien Vandenbroucque**[1,2]   **Guillaume Dalle**[1,2,3]   **Bruno Loureiro**[4]   **Florent Krzakala**[2]   **Lenka Zdeborová**[1]

[1] École Polytechnique Fédérale de Lausanne (EPFL), SPOC laboratory, CH-1015 Lausanne, Switzerland
[2] École Polytechnique Fédérale de Lausanne (EPFL), IdePHICS laboratory, CH-1015 Lausanne, Switzerland
[3] École Polytechnique Fédérale de Lausanne (EPFL), INDY laboratory, CH-1015 Lausanne, Switzerland
[4] Département d'Informatique, École Normale Supérieure - PSL & CNRS, 45 rue d'Ulm, F-75230 Paris cedex 05, France

## Abstract

We investigate popular resampling methods for estimating the uncertainty of statistical models, such as subsampling, bootstrap and the jackknife, and their performance in high-dimensional supervised regression tasks. We provide a tight asymptotic description of the biases and variances estimated by these methods in the context of generalized linear models, such as ridge and logistic regression, taking the limit where the number of samples and dimension of the covariates grow at a comparable fixed rate. Our findings are three-fold: i) resampling methods are fraught with problems in high dimensions and exhibit the double-descent-like behavior typical of these situations; ii) only when the sampling ratio is large enough do they provide consistent and reliable error estimations (we give convergence rates); iii) in the over-parametrized regime relevant to modern machine learning practice, their predictions are not consistent, even with optimal regularization.

## 1 INTRODUCTION

Estimating and quantifying errors is a central aspect of statistical practice. Nevertheless, a solid understanding of how uncertainty can be reliably quantified in modern machine learning practice is largely missing, despite being a key endeavor towards a reliable use of these methods across sensitive applications. This paper delves into a comprehensive mathematical analysis of conventional resampling methods to estimate uncertainty, such as subsampling, the bootstrap and the jackknife, specifically in the context of high-dimensional regression and classification tasks.

Let $Z_1, \cdots, Z_n \sim p_\theta$ denote $n$ independent samples from a parametric probability distribution. Given an estimator $\hat{\theta}$ of $\theta$ (e.g. the maximum likelihood estimator), one is interested

not only in the absolute performance of $\hat{\theta}$ but also in estimating how reliable it is, e.g. error bars. In particular, even if the estimator is consistent, i.e. $\hat{\theta} \to \theta$ when $n \to \infty$, having access only to a finite amount of data $n$ introduces uncertainty in our estimation $\theta$. A central question in statistics is *how to quantify this uncertainty* [Wasserman, 2004].

A classical family of non-parametric methods developed to address this question are *resampling methods* [Tibshirani and Efron, 1993, James et al., 2023], which consist in estimating the statistics of interest from the empirical distribution $p_n = \frac{1}{n} \sum_{i=1}^n \delta_{Z_i}$. Our goal is to investigate the statistical properties of three popular resampling methods in the context of the most widespread machine learning task: *supervised learning*. Here the samples are given by pairs $Z_i = (\boldsymbol{x}_i, y_i)$ from a joint distribution $p_\theta(\boldsymbol{x}, y)$, with $\boldsymbol{x}_i \in \mathbb{R}^d$ being the covariates and $y_i \in \mathcal{Y} \subset \mathbb{R}$ the labels. Given the parameter $\hat{\theta}$ learned by a fitting model, say ridge or logistic regression, the goal is to estimate the actual bias and variance of $\hat{\theta}$.

We focus on the *high-dimensional* regime, where both the number of samples $n$ and their dimension $d$ are comparatively large, with a fixed ratio $\alpha = n/d$. We provide a tight asymptotic description of the biases and variances estimated by resampling methods for generalized linear models, such as ridge and logistic regression or any M-estimator. We show that resampling methods are fraught with problems in high-dimensions, either overestimating or underestimating the mean and variances. Reliable error estimation can only be reached in the regime when $\alpha \gg 1$, for which we provide asymptotic rates of convergences. However, in the overparametrized regime $\alpha < 1$, relevant to modern machine learning practice, the predictions of resampling methods are clearly off, even when optimally regularizing.

## 2 SETTING & MOTIVATION

We consider the class of generalized linear estimation problems, where the goal is to estimate a parameter $\boldsymbol{\theta}_\star \in \mathbb{R}^d$

*Accepted for the 40th Conference on Uncertainty in Artificial Intelligence* (UAI 2024).

from $n$ independent samples $\mathcal{D} = \{(\boldsymbol{x}_i, y_i)_{i \in [n]}\}$ drawn from the following distribution:

$$y_i \sim p(\cdot | \boldsymbol{\theta}_\star^\top \boldsymbol{x}_i), \qquad \boldsymbol{x}_i \sim \mathcal{N}(0, 1/d \boldsymbol{I}_d) \qquad (1)$$

for a general likelihood $p(y|z)$. Therefore, in this case, the joint distribution reads $p_{\boldsymbol{\theta}_\star}(\boldsymbol{x}, y) = p(y|\boldsymbol{\theta}_\star^\top \boldsymbol{x})p(\boldsymbol{x})$. For concreteness, we assume $\boldsymbol{\theta}_\star \sim \mathcal{N}(0, \boldsymbol{I}_d)$. In the following, we focus on the (regularized) maximum likelihood estimator:

$$\hat{\boldsymbol{\theta}}_\lambda(\mathcal{D}) = \underset{\boldsymbol{\theta} \in \mathbb{R}^d}{\arg\min} \sum_{i=1}^n -\log p\left(y_i | \boldsymbol{\theta}^\top \boldsymbol{x}_i\right) + \frac{\lambda}{2} \|\boldsymbol{\theta}\|_2^2 \quad (2)$$

also known as *empirical risk minimizer* in the context of supervised machine learning, where the loss function coincides with minus the empirical log-likelihood: $\ell(y, z) = -\log p(y|z)$. When it is clear from the context, we omit the training data dependence $\mathcal{D}$ in the MLE estimator and write $\hat{\boldsymbol{\theta}}_\lambda$.

We will focus on two particular examples of generalized linear estimation: ridge and logistic regression. Ridge regression is a regression problem $\mathcal{Y} = \mathbb{R}$, which corresponds to the Gaussian likelihood $p(y|z) = \mathcal{N}(y|z, \Delta)$ of mean $z$ and variance $\Delta$ (or equivalently the square loss function $\ell(y, z) = \frac{1}{2\Delta}(y - z)^2$) for $\Delta > 0$. Instead, logistic regression is a binary classification problem $\mathcal{Y} = \{-1, +1\}$ which corresponds to a logit likelihood $p(y|z) = \sigma(yz)$ for $\sigma(t) = (1 + e^{-t})^{-1}$ the logistic function (this corresponds to the logistic or cross-entropy loss function $\ell(y, z) = \log(1 + e^{-yz})$).

Note that the estimation problem introduced above is well-specified, and therefore enjoys strong mathematical guarantees in the classical statistical regime where $n \to \infty$ at fixed $d$. For instance, a well-known result is the asymptotic normality of the MLE for $\lambda = 0$ [Wasserman, 2004]:

$$\sqrt{n}\left(\hat{\boldsymbol{\theta}}_0 - \boldsymbol{\theta}_\star\right) \overset{(d)}{\to} \mathcal{N}(0, \mathcal{I}^{-1}), \qquad n \to \infty \quad (3)$$

where $\mathcal{I} \in \mathbb{R}^{d \times d}$ is the Fisher information matrix, in particular implying consistency and calibration of the maximum likelihood estimator. However, those guarantees break down when the number of samples is comparable with the dimension of the covariates $n = \Theta(d)$. This is precisely the regime of interest in our work, and applying it to resampling methods will be our goal in the following.

## 2.1 WHAT STATISTICIANS WANT

"Bias" and "variance" depend on the underlying data sampling process, and therefore, different notions co-exist, whether one takes, for instance, a frequentist or Bayesian viewpoint. Below, we define these different quantities, which resampling methods try to approximate.

**Frequentist bias and variance —** In the classical frequentist approach, the statistician seeks to estimate the bias and variance with respect to the data sampling process. This induces the classical *bias-variance decomposition* of the mean squared error for the estimator $\hat{\boldsymbol{\theta}}_\lambda$:

$$\mathrm{MSE}(\hat{\boldsymbol{\theta}}_\lambda) = \frac{1}{d} \mathbb{E}_{\mathcal{D}, \boldsymbol{\theta}_\star} \left[\|\hat{\boldsymbol{\theta}}_\lambda - \boldsymbol{\theta}_\star\|^2\right] = \mathrm{Bias}_{\mathcal{D}}^2(\hat{\boldsymbol{\theta}}_\lambda) + \mathrm{Var}_{\mathcal{D}}(\hat{\boldsymbol{\theta}}_\lambda)$$

with:

$$\mathrm{Bias}_{\mathcal{D}}^2(\hat{\boldsymbol{\theta}}_\lambda) = \frac{1}{d} \left\|\mathbb{E}_{\mathcal{D}, \boldsymbol{\theta}_\star}\left[\hat{\boldsymbol{\theta}}_\lambda\right] - \boldsymbol{\theta}_\star\right\|^2 \quad (4)$$

$$\mathrm{Var}_{\mathcal{D}}(\hat{\boldsymbol{\theta}}_\lambda) = \frac{1}{d} \mathbb{E}_{\mathcal{D}, \boldsymbol{\theta}_\star}\left[\left\|\hat{\boldsymbol{\theta}}_\lambda - \mathbb{E}_{\mathcal{D}, \boldsymbol{\theta}_\star}\left[\hat{\boldsymbol{\theta}}_\lambda\right]\right\|^2\right]. \quad (5)$$

We emphasize that in this case, the expectations are taken with respect to sampling of the full data set $\mathcal{D} = \{(\boldsymbol{x}_i, y_i)_{i \in [n]}\} \sim p_{\boldsymbol{\theta}_\star}^{\otimes n}$.

**Conditional bias and variance —** Alternatively, in a supervised learning setting one can define the bias and variance only with respect to the sampling of the labels $y_i \sim p(\cdot | \boldsymbol{x}_i^\top \boldsymbol{\theta}_\star)$, i.e. conditionally on the covariates $\boldsymbol{x}_i$. This is known as a *fixed design* analysis. We will refer to the corresponding notions as *conditional* bias and variance:

$$\mathrm{Bias}_{\mathcal{D}|\boldsymbol{X}}^2(\hat{\boldsymbol{\theta}}_\lambda) = \frac{1}{d} \left\|\mathbb{E}_{\mathcal{D}}[\hat{\boldsymbol{\theta}}_\lambda | \boldsymbol{X}] - \boldsymbol{\theta}_\star\right\|^2 \quad (6)$$

$$\mathrm{Var}_{\mathcal{D}|\boldsymbol{X}}(\hat{\boldsymbol{\theta}}_\lambda) = \frac{1}{d} \mathbb{E}_{\mathcal{D}} \left\|\hat{\boldsymbol{\theta}}_\lambda - \mathbb{E}[\hat{\boldsymbol{\theta}}_\lambda | \boldsymbol{X}]\right\|^2, \quad (7)$$

where for convenience we defined the covariate matrix $\boldsymbol{X} \in \mathbb{R}^{n \times d}$ with rows given by the covariates $\boldsymbol{x}_i \in \mathbb{R}^d$.

**Bayesian estimator and variance —** Finally, it is natural to compare the maximum likelihood estimator above with the best estimator (in mean squared error) conditioned on the full training data $\mathcal{D}$, also known as the *Bayes-optimal* estimator. It requires, however, the knowledge of the *a priori* distribution of the "true" weights.

$$\hat{\boldsymbol{\theta}}_{\mathrm{bo}} = \underset{\hat{\boldsymbol{\theta}} \in \mathbb{R}^d}{\arg\min} \, \mathbb{E}\left[\|\hat{\boldsymbol{\theta}} - \boldsymbol{\theta}_\star\|^2\right] = \mathbb{E}[\boldsymbol{\theta}|\mathcal{D}] \quad (8)$$

where the conditional expectation is taken with respect to the posterior distribution:

$$p(\boldsymbol{\theta}|\mathcal{D}) \propto \mathcal{N}(\boldsymbol{\theta}|0, \boldsymbol{I}_d) \prod_{i=1}^n p(y_i | \boldsymbol{\theta}^\top \boldsymbol{x}_i) \quad (9)$$

Note that, by definition, $\hat{\boldsymbol{\theta}}_{\mathrm{bo}}$ is an unbiased and calibrated estimator of $\boldsymbol{\theta}_\star$ [Clarté et al., 2023b]. Nevertheless, it captures the irreducible variance due to the fact we have a finite sample $\mathcal{D}$ of the population distribution:

$$\mathrm{Var}_{\mathrm{bo}} = \frac{1}{d} \mathbb{E}\left[\|\boldsymbol{\theta} - \boldsymbol{\theta}_{\mathrm{bo}}\|^2 \, |\mathcal{D}\right] \quad (10)$$

where, again, the expectation is taken over the posterior distribution $p(\boldsymbol{\theta}|\mathcal{D})$.

## 2.2 RESAMPLING ESTIMATES

A central problem in statistics is the estimation of the biases (4) & (6) and variances (5) & (7), which involve population expectations, from a finite number of samples $\mathcal{D} = \{(\boldsymbol{x}_i, y_i)_{i \in [n]}\}$. Resampling methods are a popular class of statistical procedures that fit a family of $B$ estimators $\hat{\boldsymbol{\theta}}_b \equiv \hat{\boldsymbol{\theta}}_\lambda(\mathcal{D}_b^\star)$ from resampled data $\mathcal{D}_b^\star$ generated from the original samples $\mathcal{D} = \{(\boldsymbol{x}_i, y_i)_{i \in [n]}\}$, and from which the bias and variance of $\hat{\boldsymbol{\theta}}_\lambda$ can be estimated:

$$\widehat{\text{Bias}}^2 = \frac{1}{d} \left\| \frac{1}{B} \sum_{b=1}^B \hat{\boldsymbol{\theta}}_b - \hat{\boldsymbol{\theta}}_\lambda \right\|^2, \tag{11}$$

$$\widehat{\text{Var}} = \frac{1}{dB} \sum_{b=1}^B \left\| \hat{\boldsymbol{\theta}}_b - \frac{1}{B} \sum_{b=1}^B \hat{\boldsymbol{\theta}}_b \right\|^2 \tag{12}$$

In this work, we will focus on the following methods:

**- Pair bootstrap:** Consists in resampling $\mathcal{D}_b^\star$ from $\mathcal{D}$ with sample replacements, or in other words, sampling $\mathcal{D}_b^\star = \{(\boldsymbol{x}_{b,i}^\star, y_{b,i}^\star)_{i \in [n]}\} \sim p_n^{\otimes n}$ from the empirical distribution.

**- Residual bootstrap:** Akin to the pair bootstrap method, but for the conditional distribution $p(y|z)$. In practice, one first fits an estimator $\hat{\boldsymbol{\theta}}_\lambda(\mathcal{D})$ on the original samples (the MLE (2) in our setting), and given a statistical model for $\hat{p}(y|z)$, one resamples only the labels from $\hat{p}(y|\hat{\boldsymbol{\theta}}_\lambda(\mathcal{D})^\top \boldsymbol{x}_i)$, generating new datasets $\mathcal{D}_b^\star = \{\boldsymbol{x}_i, y_{b,i}^\star\}_{i=1}^n$. This allows for the estimation of conditional statistical errors.

**- Subsampling:** Consists of generating new datasets $\mathcal{D}_b^\star$ of a smaller size $\lfloor rn \rfloor$ by subsampling $\mathcal{D}$ without replacement, where $r \in (0,1)$. While bootstrap creates datasets of the right size but from the wrong distribution (as elements of $\mathcal{D}$ are duplicated), subsampling relies on data of the wrong size but from the right distribution.[1]

**- Jackknife:** Consists of creating $B = n$ datasets $\mathcal{D}_b^\star = \{(\boldsymbol{x}_i, y_i)_{i \neq b}\}$, each of which leaves a single sample out. Note that when $n \to \infty$, as in our high-dimensional regime, this is equivalent to subsampling with $r \to 1$.

For notational convenience, we will refer to these statistics as $\widehat{\text{Bias}}^2_t, \widehat{\text{Var}}_t$ with $t \in \{\text{pb}, \text{rb}, \text{ss}, \text{jk}\}$ for pair (pb) and residual bootstrap (rb), subsampling (ss) and jackknife (jk).

## 3 CONTRIBUTIONS & RELATED WORK

The resampling methods above have been widely studied in the classical statistical literature, with whole books dedicated to proving their mathematical soundness [Efron, 1979, Efron and Tibshirani, 1986, Davison and Hinkley, 1997]. However, as discussed in Section 2 most of the classical guarantees hold in the regime where the quantity of data

---

[1]Since the $\mathcal{D}_b^\star$'s are independent conditionally on $\mathcal{D}$.

$n$ available to the statistician is large in comparison with data dimension $d$ — a regime that falls short in the context of modern machine learning practice. Of particular importance was the work of Karoui and Purdom [2018] who have pointed out the lack of consistency of the bootstrap method for *unregularized* least squares, in the *underparametrized regime* $n > d$. One of our goals in this manuscript is to fill the gap, providing a complete evaluation of the aforementioned methods (beyond bootstrap), including the effect of regularization and over-parametrization.

More precisely, our **main contributions** are:

• We provide a closed-form expression for the biases and variances in the proportional high-dimensional limit where $n, d \to \infty$ at fixed rate $\alpha = n/d$ for all the cases discussed in Section 2: the pair and residual biases and variances and their bootstrap, subsample, and jackknife estimates. Our result holds for generic log-concave likelihoods (corresponding to convex losses) and convex regularizers.

• Our formulas are derived from mapping to a Generalized Approximate Message Passing (GAMP) scheme admitting a rigorous asymptotic characterization in terms of *state evolution* equations [Bayati and Montanari, 2011a,b, Javanmard and Montanari, 2014, Emami et al., 2020, Loureiro et al., 2021]. We believe this derivation has an interest on its own, as we show how simultaneously tracking *coupled* GAMP trajectories provides the biases and variances for all the resampling methods. Our construction is quite generic and can be extended to other variants of interest.

• Our examination into the effectiveness and limitations of these methods yields three key insights. Firstly, we demonstrate that resampling techniques face significant challenges in high-dimensional contexts, resulting in a double-descent behavior typical of such scenarios. Secondly, we find that these methods yield consistent and reliable error estimates only when the ratio $\alpha$ is sufficiently large, for which we also present convergence rates. Thirdly, in the overparametrized regime where $\alpha < 1$, the predictions remain inconsistent despite optimal regularization.

**Further related work —** Resampling methods are a classical topic in statistics. The jackknife method was introduced in Quenouille [1956], refined by Tukey [1958] and analysed by Efron and Stein [1981]. Bootstrap was introduced by Efron [1979], and studied in the context of least squares estimation in Freedman [1981], Wu [1986].

The asymptotic theory of high-dimensional statistical generalized linear problems has witnessed a burst of activity over the last decades. Pioneered by the statistical physics community in the late 80s [Gardner and Derrida, 1989, Opper et al., 1990, Krogh and Hertz, 1991, Seung et al., 1992, Kabashima and Shinomoto, 1992], it is now an established field of research encompassing applications to machine learning, statistics, and signal processing among others [Bayati and Montanari, 2011b, El Karoui et al., 2013,

Donoho and Montanari, 2016, Thrampoulidis et al., 2015, 2018, Dobriban and Wager, 2018, Sur and Candès, 2019, 2020, Gerbelot et al., 2020, Takahashi and Kabashima, 2022, Loureiro et al., 2021, 2022, Bellec and Zhang, 2023, Bellec, 2023]. Bayes-optimal generalization guarantees for generalized linear models were established by Donoho et al. [2013], Krzakala et al. [2012], Barbier et al. [2019], Maillard et al. [2020]. Sur and Candès [2020] have shown that, besides not being well-defined when $n < d$, the unregularized maximum likelihood estimator is biased [El Karoui et al., 2013, Karoui, 2013, Bean et al., 2013, Sur and Candès, 2019, Bellec et al., 2022] for $n > d$. One consequence is that the variance of the MLE underestimates the true variance of $\boldsymbol{\theta}_\star$, leading to an overconfident prediction [Bai et al., 2021a,b, Clarté et al., 2023b]. Indeed, Clarté et al. [2023b,a] highlighted the importance of properly regularizing the MLE in the high-dimensional regime, showing that cross-validation over $\lambda$ can mitigate some of these issues. Clarté et al. [2023] showed that post-training *temperature scaling* can mitigate overconfidence, regardless of the regularization used.

Bagging (the combination of subsampling with ensembling) has been studied in the high-dimensional regime by [Sollich and Krogh, 1995, Krogh and Sollich, 1997, LeJeune et al., 2020, Patil et al., 2023, Du et al., 2023, Chen et al., 2023, Ando and Komaki, 2023, Patil and LeJeune, 2023]. Ensembling has also been investigated in the context of the random features model as a tool to decouple the different sources of randomness [D'Ascoli et al., 2020, Lin and Dobriban, 2021, Adlam and Pennington, 2020, Loureiro et al., 2023]. The performance of bootstrap averaging has been studied in the context of Gaussian Processes and Support Vector Machines using the replica method by Malzahn and Opper [2002, 2003]. A replicated AMP algorithm for computing bootstrap averages of GLMs was proposed by Takahashi and Kabashima [2019] and studied in the context of LASSO [Obuchi and Kabashima, 2019] and Elastic Net [Takahashi, 2023].

Finally, we note that resampling methods in the context of generalized linear models are not just theoretical abstractions but are actually used in machine learning practice. For instance, Musil et al. [2019] use subsampling to estimate the uncertainty in kernel regression for the energy of molecular compounds. Their observation that subsampling yields a better uncertainty estimation than Bootstrap or Gaussian processes is one motivation for the present work.

# 4 MAIN TECHNICAL RESULTS

The key observation in the results that follow is that in order to asymptotically characterize the biases and variances associated with any of the resampling methods in Section 2, it is sufficient to characterize only a few correlations. For

example, the resampling variance (12):

$$\widehat{\mathrm{Var}} = \frac{1}{d}\left(\frac{1}{B}\sum_{k=1}^{B}\|\hat{\boldsymbol{\theta}}_k\|^2 - \frac{1}{B^2}\sum_{k,k'=1}^{B}\hat{\boldsymbol{\theta}}_k^\top\hat{\boldsymbol{\theta}}_{k'}\right). \quad (13)$$

Assuming the data sets $\mathcal{D}_k^\star$ are independently resampled from $\mathcal{D}$, it is then enough to characterize the norm of $\hat{\boldsymbol{\theta}}_1$ and the correlation between two independent (conditionally on $\mathcal{D}$) resampled estimators $\hat{\boldsymbol{\theta}}_1^\top \hat{\boldsymbol{\theta}}_2$ - with all the rest being statistically similar. The results that follow precisely characterize these quantities asymptotically. Finally, the methods defined in Section 2 naturally divide into two categories: estimators for the statistics of the joint distribution $p_{\boldsymbol{\theta}_\star}(\boldsymbol{x}, y)$ (we refer to them as *pair resampling*) and for the conditional distribution $p(y|\boldsymbol{\theta}_\star^\top \boldsymbol{x})$ (we refer to them as *conditional* or *residual resampling*). Below, we start by discussing our results for the former.

## 4.1 PAIR RESAMPLING

The key idea is to reframe the regularized MLE problem (2) as a *weighted empirical risk minimization* (wERM) problem:

$$\hat{\boldsymbol{\theta}}_\lambda(\mathcal{D}, \boldsymbol{p}) = \arg\min_{\boldsymbol{\theta} \in \mathbb{R}^d} \sum_{i=1}^n -p_i \log p\left(y_i|\boldsymbol{\theta}^\top \boldsymbol{x}_i\right) + \lambda/2\|\boldsymbol{\theta}\|^2 \quad (14)$$

where for each sample $(\boldsymbol{x}_i, y_i) \in \mathcal{D}$, we have introduced a sample weight $p_i$. When $p_i = 1$ for all $i \in [n]$, this reduces to standard MLE (2), which we sometimes refer to as full resampling (abbreviated fr). However, by taking the $p_i$'s at random from a judiciously chosen distribution, we can asymptotically cover all pair resampling methods from Section 2.

Indeed, it is immediate to see that by choosing $p_i \in \{0, 1\}$ at random from a Bernoulli distribution with probability $r \in (0, 1]$, the wERM (14) asymptotically corresponds to doing subsampling. Intuitively, this can be seen as throwing a coin for each sample $i \in [n]$ in order to decide whether to include it in the subsampled batch $\mathcal{D}_{\mathrm{ss}}^\star$, which on average will contain precisely $r$ samples. The jackknife estimator can then be obtained as the $r \to 1^-$ limit of subsampling.

Similarly, pair bootstrap is asymptotically equivalent to taking $p_i \sim \mathrm{Pois}(1)$ independently. Indeed, for finite $n$, pair bootstrap exactly corresponds to taking $\boldsymbol{p} \in \mathbb{R}^n$ from the multinomial distribution $\mathrm{Multinomial}(n, 1/n)$. As $n \to \infty$, this is marginally equivalent to choosing $p_i \sim \mathrm{Pois}(1)$ independently [Karoui and Purdom, 2018, Section 3.1].

To summarize, each resampling method can be thought of as applying sampling weights which are i.i.d., with distributions defined as

$$\begin{cases} \mu_{\mathrm{pb}}(p) & := \frac{1}{ep!} \\ \mu_{\mathrm{ss(r)}}(p) & := r^p(1-r)^{1-p} \text{ for } r \in (0,1). \end{cases} \quad (15)$$

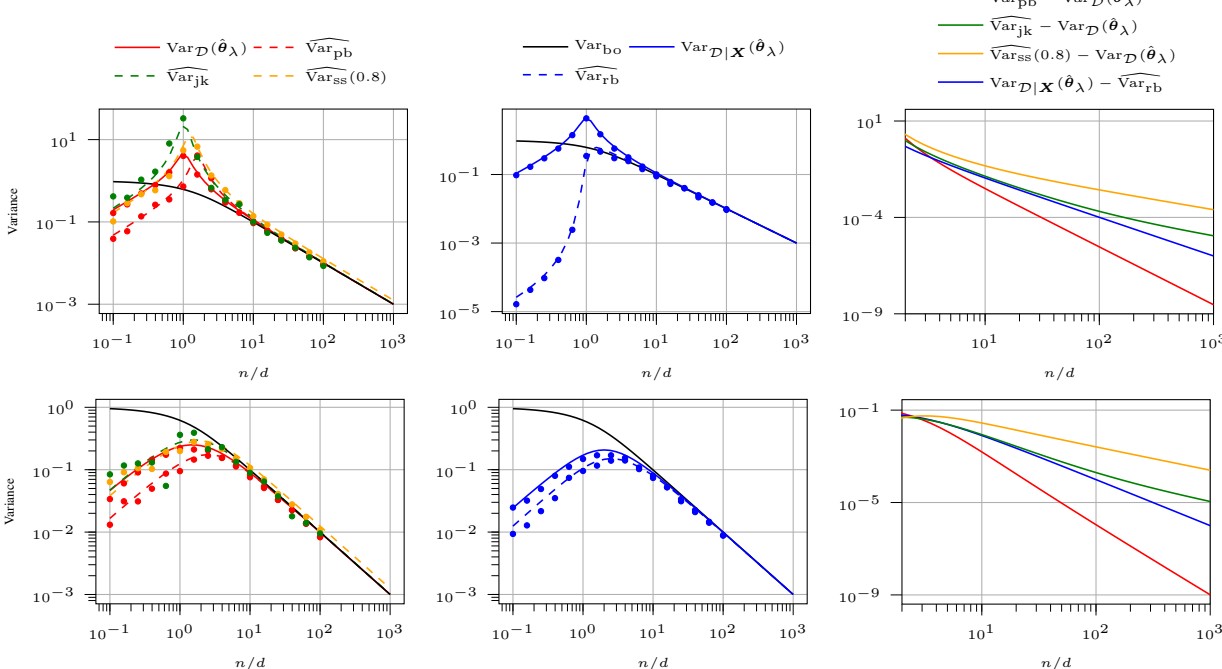

Figure 1: Variances for ridge regression at $\lambda = 10^{-2}$ (Top) and $\lambda = 1$ (Bottom). Left: variance of pair resampling methods and of Bayes-posterior. Middle: variance of conditional resampling and residual bootstrap. Right: difference between the true variances $\mathrm{Var}_{\mathcal{D}}(\hat{\boldsymbol{\theta}}_\lambda)$, $\mathrm{Var}_{\mathcal{D}|\boldsymbol{X}}(\hat{\boldsymbol{\theta}}_\lambda)$ and their estimation. Dots are simulations done at $d = 200$, with $B = 10$ resamples for bootstrap and subsampling.

We note that a key assumption which permits to retrieve our result is that for a particular resampling method, the sample weights $p_i$, $i \in [n]$ are *i.i.d.*. We are now ready to state our first two results for pair resampling. For the sake of clarity, we state our results for ridge regression and refer to Appendix A for the derivation of our results and a statement for general convex loss and penalties.

In the following, the asymptotic values of correlations needed to compute biases and variances will be referred to as *overlaps*. For $\mathrm{t} \in \{\mathrm{pb}, \mathrm{ss}, \mathrm{jk}\}$, these overlaps read:

$$
\begin{cases}
Q_{11}^{\mathrm{t}} := \lim_{n,d\to\infty} \mathbb{E}_{\boldsymbol{\theta}_\star, \mathcal{D}, \boldsymbol{p}}\left[\|\hat{\boldsymbol{\theta}}_\lambda(\mathcal{D}, \boldsymbol{p})\|^2\right] \\
Q_{12}^{\mathrm{t}} := \lim_{n,d\to\infty} \mathbb{E}_{\boldsymbol{\theta}_\star, \mathcal{D}}\left[\|\mathbb{E}_{\boldsymbol{p}}[\hat{\boldsymbol{\theta}}_\lambda(\mathcal{D}, \boldsymbol{p})]\|^2\right] \\
Q_{11}^{\mathrm{fr}} := \lim_{n,d\to\infty} \mathbb{E}_{\boldsymbol{\theta}_\star, \mathcal{D}}\left[\|\hat{\boldsymbol{\theta}}_\lambda(\mathcal{D})\|^2\right] \\
Q_{12}^{\mathrm{fr}} := \lim_{n,d\to\infty} \mathbb{E}_{\boldsymbol{\theta}_\star}\left[\|\mathbb{E}_{\mathcal{D}}[\hat{\boldsymbol{\theta}}_\lambda(\mathcal{D})]\|^2\right] \\
Q_{12}^{\mathrm{fr,t}} := \lim_{n,d\to\infty} \mathbb{E}_{\boldsymbol{\theta}_\star, \mathcal{D}, \boldsymbol{p}}\left[\hat{\boldsymbol{\theta}}_\lambda(\mathcal{D})^\top \hat{\boldsymbol{\theta}}_\lambda(\mathcal{D}, \boldsymbol{p})\right] \\
m_1^{\mathrm{t}} := \lim_{n,d\to\infty} \mathbb{E}_{\boldsymbol{\theta}_\star, \mathcal{D}, \boldsymbol{p}}\left[\hat{\boldsymbol{\theta}}_\lambda(\mathcal{D}, \boldsymbol{p})^\top \boldsymbol{\theta}_\star\right] \\
m_1^{\mathrm{fr}} := \lim_{n,d\to\infty} \mathbb{E}_{\boldsymbol{\theta}_\star, \mathcal{D}}\left[\hat{\boldsymbol{\theta}}_\lambda(\mathcal{D})^\top \boldsymbol{\theta}_\star\right]
\end{cases} \quad , \quad (16)
$$

where $\boldsymbol{p} = (p_1, \ldots, p_n) \overset{\mathrm{i.i.d.}}{\sim} \mu_{\mathrm{t}}$ and fr refers to full resampling. In what follows, these overlaps will be written in a matrix and vector form

$$
\begin{cases}
\boldsymbol{Q}^{\mathrm{t}} &= \begin{bmatrix} Q_{11}^{\mathrm{t}} & Q_{12}^{\mathrm{t}} \\ Q_{12}^{\mathrm{t}} & Q_{11}^{\mathrm{t}} \end{bmatrix} \\
\boldsymbol{Q}^{\mathrm{fr,t}} &= \begin{bmatrix} Q_{11}^{\mathrm{fr}} & Q_{12}^{\mathrm{fr,t}} \\ Q_{12}^{\mathrm{fr,t}} & Q_{11}^{\mathrm{t}} \end{bmatrix} \\
\boldsymbol{Q}^{\mathrm{fr}} &= \begin{bmatrix} Q_{11}^{\mathrm{fr}} & Q_{12}^{\mathrm{fr}} \\ Q_{12}^{\mathrm{fr}} & Q_{11}^{\mathrm{fr}} \end{bmatrix} \\
\boldsymbol{m}^{\mathrm{t}} &= [m_1^{\mathrm{t}}, m_1^{\mathrm{t}}]^\top \\
\boldsymbol{m}^{\mathrm{fr,t}} &= [m_1^{\mathrm{fr}}, m_1^{\mathrm{t}}]^\top
\end{cases} \quad (17)
$$

Intuitively, for $\mathrm{t} \in \{\mathrm{pb}, \mathrm{ss}, \mathrm{jk}\}$ the matrix $\boldsymbol{Q}^{\mathrm{t}} \in \mathbb{R}^{2\times 2}$ represents the Gram matrix of two estimators trained on two independent resamples of the same training data $\mathcal{D}$. Similarly, $\boldsymbol{Q}^{\mathrm{fr}}$ is a Gram matrix between two estimators trained two datasets sampled independently from the same teacher $\boldsymbol{\theta}_\star$. Moreover, the vector $\boldsymbol{m}^{\mathrm{t}}$ contains the correlation between estimators trained with method t and $\boldsymbol{\theta}_\star$. Our main technical result is a characterization of these quantities in the high-dimensional limit.

**Theorem 4.1** (Biases and Variances for pair resampling in ridge regression). *Let $\mathcal{D} = \{(\boldsymbol{x}_i, y_i)_{i\in[n]}\}$ denote $n$ independent samples drawn from model (1) with log-concave likelihood $p(y|z)$. In the high-dimensional proportional regime $n, d \to \infty$ with $n/d = \alpha$, the overlaps of interest (17) are given by the unique solution $\boldsymbol{m} \in \mathbb{R}^2$,*

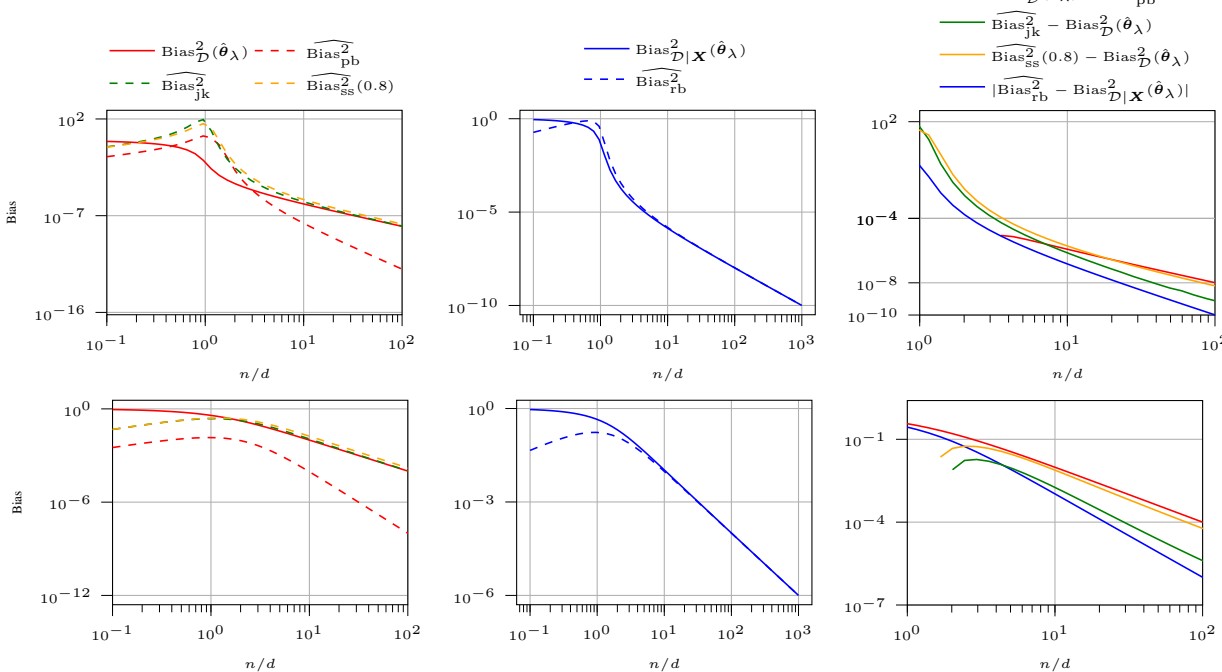

Figure 2: Bias of ridge regression and its estimation using pair bootstrap and subsampling at $\lambda = 10^{-2}$ (Top) and $\lambda = 1$ (Bottom). Left: bias of pair resampling methods. Middle: conditional bias and bias of residual bootstrap. Right: difference between the various biases.

$\boldsymbol{Q} \in \mathbb{R}^{2\times 2}, \boldsymbol{V} \in \mathbb{R}^2$ *to the following set of self-consistent equations:*

$$\begin{cases} \boldsymbol{m} &= \left(\lambda \boldsymbol{I}_2 + \hat{\boldsymbol{V}}\right)^{-1} \hat{\boldsymbol{m}} \\ \boldsymbol{Q} &= \left(\lambda \boldsymbol{I}_2 + \hat{\boldsymbol{V}}\right)^{-1} \left(\hat{\boldsymbol{m}}\hat{\boldsymbol{m}}^\top + \hat{\boldsymbol{Q}}\right) \left(\lambda \boldsymbol{I}_2 + \hat{\boldsymbol{V}}\right)^{-1\top} \\ \boldsymbol{V} &= \left(\lambda \boldsymbol{I}_2 + \hat{\boldsymbol{V}}\right)^{-1} \end{cases}$$

$$(18)$$

$$\begin{cases} \hat{\boldsymbol{m}} = \alpha \mathbb{E}_{\boldsymbol{p}} \left[\boldsymbol{G}(\boldsymbol{p})\right] \mathbf{1}_2 \\ \hat{\boldsymbol{Q}} = \alpha \mathbb{E}_{\boldsymbol{p}} \left[\boldsymbol{G}(\boldsymbol{p}) \left((v_\star + \Delta)\mathbf{1}_{2\times 2} + \boldsymbol{BQB}^\top\right) \boldsymbol{G}(\boldsymbol{p})^\top\right] \\ \hat{\boldsymbol{V}} = \alpha \mathbb{E}_{\boldsymbol{p}} \left[\boldsymbol{G}(\boldsymbol{p})\right] \end{cases}$$

$$(19)$$

*for a careful choice of the joint distribution of* $\boldsymbol{p} = (p_1, p_2)$. *In the above,* $\boldsymbol{G}(\boldsymbol{p}) = (\boldsymbol{I}_2 + \boldsymbol{PV})^{-1}\boldsymbol{P}$ *with* $\boldsymbol{P} = \mathrm{Diag}(\boldsymbol{p})$, $\boldsymbol{B} = \mathbf{1}_2 \boldsymbol{m}^\top \boldsymbol{Q}^{-1} - \boldsymbol{I}_2$ *and* $v_\star = 1 - \boldsymbol{m}^\top Q^{-1}\boldsymbol{m}$.

*Then, the following holds:*

• *the variance of resampling method* $\mathrm{t} \in \{\mathrm{pb}, \mathrm{ss}, \mathrm{jk}\}$ *is given by*

$$\widehat{\mathrm{Var}}_t = Q_{11}^{\mathrm{t}} - Q_{12}^{\mathrm{t}}, \qquad (20)$$

*where overlaps with superscript* t *are obtained by solving* (18), (19) *using joint distribution* $\mu(p_1, p_2) = \mu_{\mathrm{t}}(p_1) \cdot \mu_{\mathrm{t}}(p_2)$.

• *the true variance is given by*

$$\mathrm{Var}_{\mathcal{D}}(\hat{\boldsymbol{\theta}}_\lambda) = Q_{11}^{\mathrm{fr}} - Q_{12}^{\mathrm{fr}}, \qquad (21)$$

*where overlaps with superscript* fr *(indicating full resampling) are obtained by solving* (18), (19) *using joint distribution*

$$\mu(p_1, p_2) = (\mathbb{1}(p_1 = 0, p_2 = 1) + \mathbb{1}(p_1 = 1, p_2 = 0)).$$

• *the squared bias of resampling method* t *is given by*

$$\widehat{\mathrm{Bias}_{\mathrm{t}}^2} = Q_{11}^{\mathrm{fr}} + Q_{12}^{\mathrm{t}} - 2Q_{12}^{\mathrm{fr},\mathrm{t}}, \qquad (22)$$

*where overlaps with superscript* t, fr *are obtained by solving* (18), (19) *using distribution* $\mu(p_1, p_2) = \mu_{\mathrm{t}}(p_1) \cdot \mathbb{1}\{p_2 = 1\}$ *for* $p_1, p_2$.

• *the true squared bias is given by*

$$\mathrm{Bias}_{\mathcal{D}}^2(\hat{\boldsymbol{\theta}}_\lambda) = 1 - 2m_1^{\mathrm{fr}} + Q_{12}^{\mathrm{fr}}. \qquad (23)$$

The details for the derivations of Theorem 4.1 are shown in Appendix A.2.

**The specific case of subsampling** To make Theorem 4.1 more concrete, we consider in this paragraph the particular case of subsampling, for which Equations (18) and (19) can be written in a more succint form. Indeed, for subsampling

with ratio $r$, the overlaps $m_1^{\mathrm{ss}}$, $Q_{11}^{\mathrm{ss}}$ and $Q_{12}^{\mathrm{ss}}$ are given by

$$
\begin{cases}
m_1^{\mathrm{ss}} & = 1 - \lambda v \\
Q_{11}^{\mathrm{ss}} & = (m_1^{\mathrm{ss}})^2 \cdot \frac{\alpha r + 1 + \Delta - 2m_1^{\mathrm{ss}}}{\alpha r - (m_1^{\mathrm{ss}})^2} \; , \\
Q_{12}^{\mathrm{ss}} & = (m_1^{\mathrm{ss}})^2 \cdot \frac{\alpha + 1 + \Delta - 2m_1^{\mathrm{ss}}}{\alpha - (m_1^{\mathrm{ss}})^2}
\end{cases}
\tag{24}
$$

where $v = \frac{1 - \lambda - \alpha r + \sqrt{(\alpha r + \lambda - 1)^2 + 4\lambda}}{2\lambda}$, as detailed in Appendix C.1.3. With this representation, the dependency of the overlaps on the different parameters such as $\alpha$ and on the subsampling ratio $r$ becomes much more explicit. We note in particular that the overlap $Q_{11}^{\mathrm{ss}}$ of one of the subsampling estimator with itself, depends only on the subsampled data it has seen, explaining the dependency on $\alpha r$. On the other hand, the overlap $Q_{12}^{\mathrm{ss}}$ involves both subsampling estimators, so that a dependency on $\alpha$ also appears since all samples are considered.

## 4.2 CONDITIONAL RESAMPLING

Similar to pair resampling, we leverage the fact that the conditional bias and variance, together with the estimates by residual bootstrap, can be written in terms of correlations between estimators. The key difference here is that the covariates $\boldsymbol{x}_1, \ldots, \boldsymbol{x}_n$ remain constant, and only the labels are resampled. Focusing on linear regression, in the case of residual resampling (abbreviated rr), the labels are sampled from the true distribution $y_i^\star \sim \mathcal{N}(\boldsymbol{\theta}_\star^\top \boldsymbol{x}_i, \Delta)$, whereas for residual bootstrap, we use the ERM estimator to approximate this distribution and $y_i^\star \sim \mathcal{N}(\hat{\boldsymbol{\theta}}_\lambda^\top \boldsymbol{x}_i, \tilde{\Delta})$ with $\tilde{\Delta}$ an estimator of $\Delta$. Similarly to pair bootstrap, we now just need the correlation between $B$ estimators $\hat{\boldsymbol{\theta}}_{\lambda,b}$ trained on resampled datasets $\mathcal{D}_b^\star = \{(\boldsymbol{x}_i, y_{i,b}^\star)_{i=1}^n\}$. This can be done by considering the minimization problem (26). Despite minimizing each $\hat{\boldsymbol{\theta}}_{\lambda,b}$ independently, they see the same covariates $\boldsymbol{x}_i$. In Appendix B.1, we discuss how this correlation can be exactly captured by designing a particular approximate message passing, and also provide more details and an extension to more generic losses. As in the previous section, we first define the overlaps of interest

$$
\begin{cases}
Q_{11}^{\mathrm{rb}} & := \lim_{n,d\to\infty} \mathbb{E}_{\boldsymbol{\theta}_\star, \mathcal{D}} \left[ \mathbb{E}_{\boldsymbol{y}^\star | \mathcal{D}} \left[ \| \hat{\boldsymbol{\theta}}_\lambda(\boldsymbol{X}, \boldsymbol{y}^\star) \|^2 \right] \right] \\
Q_{12}^{\mathrm{rb}} & := \lim_{n,d\to\infty} \mathbb{E}_{\boldsymbol{\theta}_\star, \mathcal{D}} \left[ \| \mathbb{E}_{\boldsymbol{y}^\star | \mathcal{D}} [\hat{\boldsymbol{\theta}}_\lambda(\boldsymbol{X}, \boldsymbol{y}^\star)] \|^2 \right] \\
Q_{11}^{\mathrm{rr}} & := \lim_{n,d\to\infty} \mathbb{E}_{\boldsymbol{\theta}_\star, \mathcal{D}} \left[ \| \hat{\boldsymbol{\theta}}_\lambda \|^2 | \boldsymbol{X} \right] \\
Q_{12}^{\mathrm{rr}} & := \lim_{n,d\to\infty} \mathbb{E}_{\boldsymbol{\theta}_\star} \left[ \| \mathbb{E}_{\mathcal{D}} [\hat{\boldsymbol{\theta}}_\lambda | \boldsymbol{X}] \|^2 \right] \\
m_1^{\mathrm{rb}} & := \lim_{n,d\to\infty} \mathbb{E}_{\boldsymbol{\theta}_\star, \mathcal{D}} \left[ \hat{\boldsymbol{\theta}}_\lambda(\mathcal{D})^\top \mathbb{E}_{\boldsymbol{y}^\star | \mathcal{D}} \left[ \hat{\boldsymbol{\theta}}_\lambda(\boldsymbol{X}, \boldsymbol{y}^\star) \right] \right] \\
m_1^{\mathrm{rr}} & := \lim_{n,d\to\infty} \mathbb{E}_{\boldsymbol{\theta}_\star} \left[ \mathbb{E}_{\mathcal{D}} \left[ \hat{\boldsymbol{\theta}}_\lambda | \boldsymbol{X} \right]^\top \boldsymbol{\theta}_\star \right].
\end{cases}
\tag{25}
$$

and the minimization problem for conditional resampling

$$
\hat{\boldsymbol{\theta}}_{\lambda,b} = \arg\min_{\boldsymbol{\theta} \in \mathbb{R}^d} \sum_{i=1}^n -\log p(y_{b,i}^\star | \boldsymbol{\theta}^\top \boldsymbol{x}_i) + \lambda/2 \|\boldsymbol{\theta}\|^2, \tag{26}
$$

where $b = 1, \ldots, B$.

**Theorem 4.2** (Biases and Variances for conditional resampling in ridge regression). *Let $\mathcal{D} = \{(\boldsymbol{x}_i, y_i)_{i \in [n]}\}$ denote $n$ independent samples drawn from model (1) with log-concave likelihood $p(y|z)$. In the high-dimensional proportional regime $n, d \to \infty$ with $n/d = \alpha$, the overlaps of interest (25) for $\mathrm{t} \in \{\mathrm{rr}, \mathrm{rb}\}$ are given by :*

$$
\begin{cases}
m_1^{\mathrm{t}} & = \tilde{\rho}(1 - \lambda v) \\
Q_{11}^{\mathrm{t}} & = (m_1^{\mathrm{t}})^2 \cdot \frac{\alpha \tilde{\rho} + \tilde{\rho} + \tilde{\Delta} - 2m_1^{\mathrm{t}}}{\alpha \tilde{\rho}^2 - (m_1^{\mathrm{t}})^2} \\
Q_{12}^{\mathrm{t}} & = (m_1^{\mathrm{t}})^2 \cdot \frac{\alpha \tilde{\rho} + \tilde{\rho} - 2m_1^{\mathrm{t}}}{\alpha \tilde{\rho}^2 - (m_1^{\mathrm{t}})^2}
\end{cases}
\tag{27}
$$

*where $v = \frac{1 - \lambda - \alpha + \sqrt{(\alpha + \lambda - 1)^2 + 4\lambda}}{2\lambda}$. The quantities $\tilde{\Delta}, \tilde{\rho}$ take different values depending on whether bootstrap is performed or not, as detailed below. Then, the following holds:*

- *the variance of residual bootstrap is given by*

$$
\widehat{\mathrm{Var}_{\mathrm{rb}}} = Q_{11}^{\mathrm{rb}} - Q_{12}^{\mathrm{rb}}, \tag{28}
$$

*where $Q_{11}^{\mathrm{rr}}, Q_{12}^{\mathrm{rr}}$ are obtained by solving (27) using $\tilde{\rho} = Q_{11}^{\mathrm{fr}}$ and $\tilde{\Delta} = (1 + \Delta - 2m_1^{\mathrm{fr}} + Q_{11}^{\mathrm{fr}})/(1 + V_{11}^{\mathrm{fr}})^2$. Note that the overlaps with superscript* fr *are specified in Theorem 4.1.*
- *the true variance $\mathrm{Var}_{\mathcal{D}|\boldsymbol{X}}(\hat{\boldsymbol{\theta}}_\lambda)$ is given by*

$$
\mathrm{Var}_{\mathcal{D}|\boldsymbol{X}}(\hat{\boldsymbol{\theta}}_\lambda) = Q_{11}^{\mathrm{rr}} - Q_{12}^{\mathrm{rr}}, \tag{29}
$$

*where $Q_{11}^{\mathrm{rr}}, Q_{12}^{\mathrm{rr}}$ are obtained by solving (27) using $\tilde{\rho} = 1, \tilde{\Delta} = \Delta$.*
- *the squared bias of residual bootstrap*

$$
\widehat{\mathrm{Bias}_{\mathrm{rb}}^2} = Q_{11}^{\mathrm{fr}} + Q_{12}^{\mathrm{rb}} - 2m_1^{\mathrm{rb}} \tag{30}
$$

- *the true conditional squared bias is given by*

$$
\mathrm{Bias}_{\mathcal{D}|\boldsymbol{X}}^2(\hat{\boldsymbol{\theta}}_\lambda) = 1 - 2m_1^{\mathrm{rr}} + Q_{12}^{\mathrm{rr}}. \tag{31}
$$

The details for the derivations of Theorem 4.2 are shown in Appendix B and Appendix C. Compared to pair resampling, residual resampling does not involve introducing sample weights, only the labels are resampled from a conditional distribution. However, for residual bootstrap, the main idea is that the target weights $\boldsymbol{\theta}_\star$ are replaced by $\hat{\boldsymbol{\theta}}_\lambda$. Moreover, for ridge regression, we approximate the variance $\Delta$ by the averaged residual:

$$
\tilde{\Delta} = \frac{1}{n} \sum_{i=1}^n (y_i - \hat{\boldsymbol{\theta}}_\lambda^\top \boldsymbol{x}_i)^2 \tag{32}
$$

In the high-dimensional regime, the analytical expression of this training error is given by the overlaps of state-evolution, and $\tilde{\Delta} = (1 + \Delta - 2m_1^{\mathrm{fr}} + Q_{11}^{\mathrm{fr}})/(1 + V_{11}^{\mathrm{fr}})^2$. The derivation of

| Pair resampling rates | | | Residual resampling rates | | |
|---|---|---|---|---|---|
| | Rate | Error | | Rate | Error |
| $\mathrm{Var}_{\mathcal{D}}(\hat{\boldsymbol{\theta}}_\lambda)$ | $1/\alpha$ | – | $\mathrm{Var}_{\mathcal{D}\mid\boldsymbol{X}}(\hat{\boldsymbol{\theta}}_\lambda)$ | $1/\alpha$ | – |
| $\widehat{\mathrm{Var}}_{\mathrm{ss}}$ | $1/\alpha$ | $1/\alpha$ | $\widehat{\mathrm{Var}}_{\mathrm{rb}}$ | $1/\alpha$ | $1/\alpha^2$ |
| $\widehat{\mathrm{Var}}_{\mathrm{jk}}$ | $1/\alpha$ | $1/\alpha^2$ | | | |
| $\widehat{\mathrm{Var}}_{\mathrm{pb}}$ | $1/\alpha$ | $1/\alpha^3$ | $\mathrm{Bias}^2_{\mathcal{D}\mid\boldsymbol{X}}(\hat{\boldsymbol{\theta}}_\lambda)$ | $1/\alpha^2$ | – |
| $\mathrm{Bias}^2_{\mathcal{D}}(\hat{\boldsymbol{\theta}}_\lambda)$ | $1/\alpha^2$ | – | $\widehat{\mathrm{Bias}^2_{\mathrm{rb}}}$ | $1/\alpha^2$ | $1/\alpha^3$ |
| $\widehat{\mathrm{Bias}^2_{\mathrm{ss}}}$ | $1/\alpha^2$ | $1/\alpha^2$ | | | |
| $\widehat{\mathrm{Bias}^2_{\mathrm{jk}}}$ | $1/\alpha^2$ | $1/\alpha^3$ | | | |
| $\widehat{\mathrm{Bias}^2_{\mathrm{pb}}}$ | $1/\alpha^4$ | $1/\alpha^2$ | | | |

Table 1: Summary of large $\alpha$ rates for ridge regression (see Appendix C.2 for details).

this expression can be found in Loureiro et al. [2021]. We end this section by observing that so far, we considered only the variance on the weights. However, one could be interested in other types of variances such as *predictive variance*, which we discuss in Appendix D.

## 5 DISCUSSIONS AND MAIN FINDINGS

In this section we discuss the consequences of the technical results from Section 4 on the performance of resampling methods, and compare with empirical values. We refer to Appendix E for more details on the plots.

### 5.1 RIDGE REGRESSION

**Variance –** Figure 1 shows the different variances for ridge regression. We consider two important choices of regularization: $\lambda = 10^{-2}$ to approximate the behavior of unpenalized estimators, and $\lambda = \Delta = 1$ which is the optimal value of $\lambda$: this regularization minimizes the generalization error of $\hat{\boldsymbol{\theta}}_\lambda$ and its test error is the same as the Bayes-optimal estimator. As explained in Section 2.2, the variance of Jackknife is approximated by doing subsampling with $r = 0.99$. Note that the subsampling variances with ratio $r$ are rescaled by a factor $1 - r$. We compare our theoretical predictions with numerical experiments on Gaussian data and observe an excellent agreement. For $\lambda = 10^{-2}$ in the regime where $n > d$, our results are qualitatively consistent with Karoui and Purdom [2018], who showed that pair (respectively residual) bootstrap overestimates (resp. underestimates) the variance. On the other hand, our results allow us to study the variances at $d > n$. In this regime, we observe that both pair and residual bootstrap suffer from under-coverage: for residual bootstrap, it is easy to understand why, as without regularization $d > n$ the ERM interpolates the training data. Thus, the residual is exactly 0, and the residual bootstrap thus fatally underestimates the true level of noise in the data. On the other hand, subsampling and Jackknife are closer to

$\mathrm{Var}_{\mathcal{D}}(\hat{\boldsymbol{\theta}}_\lambda)$ than pair bootstrap, and as is classically known Efron and Stein [1981], the Jackknife estimate provides an upper bound of the true variance. On the right panel, we see that all variances converge to 0 with rate $1/\alpha$, and pair bootstrap converges to $\mathrm{Var}_{\mathcal{D}}(\hat{\boldsymbol{\theta}}_\lambda)$ the fastest. On the bottom row of Figure 1, we observe that optimal regularization greatly mitigates the under-coverage of bootstrapping, most notably for residual bootstrap. We thus conclude that for small values $n/d$, bootstrap fails to accurately capture the true variances, and appropriately regularizing partially mitigates this issue.

Note that conditioned on $\mathcal{D}$ and if the data generating process is known, the Bayes-optimal posterior variance $\mathrm{Var}_{\mathrm{bo}}$ is the best estimation of uncertainty on the weights. As in Theorem 4.1 and 4.2, this variance can be obtained by solving a corresponding set of self-consistent equations [Clarté et al., 2023b]. We observe that at large $\alpha$, all variances agree with $\mathrm{Var}_{\mathrm{bo}}$. However, at optimal $\lambda$ and small $n/d$, resampling will underestimate the actual posterior variance.

**Bias –** In Figure 2, we plot the bias of the different resampling methods for ridge regression with regularization $\lambda \in \{10^{-2}, 1\}$. For the Jackknife and subsampling, the estimation of the squared bias is rescaled by a factor $(1 - r)^2$. We observe that as $\alpha \to \infty$, $\mathrm{Bias}^2_{\mathcal{D}}(\hat{\boldsymbol{\theta}}_\lambda)$ and $\widehat{\mathrm{Bias}^2_{\mathrm{pb}}}$ converge to zero, as expected by the consistency of the MLE estimator (3). However, $\widehat{\mathrm{Bias}^2_{\mathrm{pb}}}$ converges as $1/\alpha^4$, while $\mathrm{Bias}^2_{\mathcal{D}}(\hat{\boldsymbol{\theta}}_\lambda) \sim 1/\alpha^2$, and pair bootstrap underestimates the true bias. We deduce that in our model, subsampling or Jackknife should thus be preferred to estimate $\mathrm{Bias}^2_{\mathcal{D}}(\hat{\boldsymbol{\theta}}_\lambda)$.

### 5.2 LOGISTIC REGRESSION

Our results extend beyond ridge regression, and the quantities of interest can be computed for any convex loss. Figure 3 displays the true variances and their estimation for regularized logistic regression with $\lambda \in \{10^{-2}, 1\}$, similarly to Figure 1. However, contrary to the ridge case, $\lambda = 1$ yields the maximum-a-posteriori estimator but does not minimize the misclassification error.

Qualitatively, we observe similar results as for ridge regression : at large $\alpha$, all methods consistently estimate the true variance and the Jackknife provides an upper bound of $\mathrm{Var}_{\mathcal{D}}(\hat{\boldsymbol{\theta}}_\lambda)$. Moreover, at low $\alpha$, regularization improves the estimation of the variance, even though $\lambda$ is not optimal.

Finally, at $\lambda = 0.01$ for both ridge and logistic regression, we observe a local maximum in the true and resampled bias and variance around $d = n$. This behavior is reminiscent of the double-descent behavior observed e.g. in random features models or neural networks : the test error achieves a local maximum at the interpolation threshold where the model can perfectly fit the training data, then decreases with the number of parameters. Moreover, we see that regulariza-

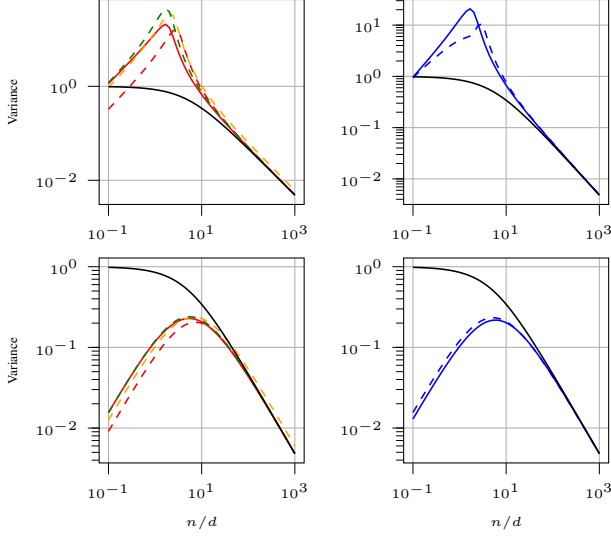

Figure 3: Variance for logistic regression at $\lambda = 10^{-2}$ (Top) and $\lambda = 1$ (Bottom). Left: variance of full resampling, pair bootstrap, subsampling. Right: variance of label resampling, residual bootstrap. See Figure 1 for the legend.

tion can mitigate this "double-descent" phenomenon.

# 6 CONCLUSION & PERSPECTIVES

In this work, we have provided an exact asymptotic comparison of the uncertainty estimations provided by different resampling methods, in the context of high-dimensional regularized maximum likelihood with generalized linear models.

Our results highlight the limitations of these methods in the high-dimensional regime relevant to modern machine learning practice and discuss how cross-validation can, to some extent, mitigate some of these limitations.

Avenues for future work are manifold. For instance, how would our results change in a misspecified scenario? Can structure in the data help or hinder resampling methods? These interesting questions are left for future investigation.

## ACKNOWLEDGEMENTS

This research was supported by the Swiss National Science Foundation grant SNFS OperaGOST, 200021_200390 and the NCCR MARVEL, a National Centre of Competence in Research, funded by the Swiss National Science Foundation (grant number 205602) and the Choose France - CNRS AI Rising Talents program.

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

# A  DERIVATION OF THE RESULTS FOR PAIR RESAMPLING

In this appendix we show how the self-consistent equations (18) and (19) can be derived from the state-evolution equation of GAMP (Generalized Approximate Message Passing), and how to extend them to generic log-concave losses.

As stated in Section 4, the key observation is that in order to asymptotically characterize the biases and variances associated with any of the resampling methods in Section 2, it is sufficient to characterize only the correlation $\hat{\boldsymbol{\theta}}_\lambda(\mathcal{D}_b^\star)^\top \hat{\boldsymbol{\theta}}_\lambda(\mathcal{D}_{b'}^\star)$ between two resampled datasets $\mathcal{D}_b^\star, \mathcal{D}_{b'}^\star$. Indeed, the resampling variances can be written

$$\widehat{\mathrm{Var}} = \frac{1}{d} \left( \frac{1}{B} \sum_{b=1}^{B} \|\hat{\boldsymbol{\theta}}_b\|^2 - \frac{1}{B^2} \sum_{b,b'=1}^{B} \hat{\boldsymbol{\theta}}_b^\top \hat{\boldsymbol{\theta}}_{b'} \right). \tag{33}$$

It is natural to study these variances in the limit $B \to \infty$. In that limit, $\widehat{\mathrm{Var}}$ converges to

$$\widehat{\mathrm{Var}} = \frac{1}{d} \mathbb{E}_{\mathcal{D}^\star} \left[ \|\hat{\boldsymbol{\theta}}(\mathcal{D}^\star)\|^2 \right] - \frac{1}{d} \mathbb{E}_{\mathcal{D}^\star, \mathcal{D}^{\star'}} \left[ \hat{\boldsymbol{\theta}}(\mathcal{D}^\star) \hat{\boldsymbol{\theta}}(\mathcal{D}^{\star'}) \right]$$

where the expectations are over resampled dataset conditioned on $\mathcal{D}$ and where the resampling depends on the method considered. In a similar way for the bias

$$\widehat{\mathrm{Bias}^2} = \frac{1}{d} \left\| \frac{1}{B} \sum_{b=1}^{B} \hat{\boldsymbol{\theta}}_b - \hat{\boldsymbol{\theta}} \right\|^2$$

$$\overset{B \to \infty}{\to} \frac{1}{d} \left( \|\hat{\boldsymbol{\theta}}\|^2 + \left\| \mathbb{E}_{\mathcal{D}^\star} \left[ \hat{\boldsymbol{\theta}}(\mathcal{D}^\star) \right] \right\|^2 \right)$$

To do so, we observe that computing the ERM estimator on a resampled dataset $\mathcal{D}^\star$ is equivalent to solving an wERM problem Equation (14), where for each sample $(\boldsymbol{x}_i, y_i) \in \mathcal{D}$, we introduce a sample weight $p_i$. The distribution on the sample weights depends on the way $\mathcal{D}$ is resampled: for example, with $p_i = 1$ for all $i \in [n]$, this reduces to standard MLE (2). On the other hand, by choosing $p_i \in \{0, 1\}$ at random from a Bernoulli distribution with probability $r \in (0, 1]$, the wERM (14) asymptotically corresponds to doing subsampling. Also, pair bootstrap is asymptotically equivalent to taking $p_i \sim \mathrm{Pois}(1)$ independently. The problem is thus to compute the correlation between estimators $\hat{\boldsymbol{\theta}}_\lambda(\mathcal{D}, \boldsymbol{p})$ trained with different, possibly correlated vectors $\boldsymbol{p}$.

The use of GAMP for deriving high-dimensional asymptotics characterization is now a classic rigorous tool, that has been used in many situations [Bayati and Montanari, 2011b, Javanmard and Montanari, 2014, Sur et al., 2019, Emami et al., 2020, Loureiro et al., 2021, 2023, Gerbelot et al., 2022]. The idea is to proceed in two steps: i) to propose a GAMP algorithm that solves the optimisation problem asymptotically, and ii) to use the fact that GAMP performance can be tracked with a rigorous state evolution Bayati and Montanari [2011a], Gerbelot and Berthier [2023] . This was, to the best of our knwoldege, introduced first in [Bayati and Montanari, 2011b] for studying the LASSO risk. We shall not repeat the proof technique, and refer the reader to [Loureiro et al., 2021, 2023] for details with our current notation. Our results directly uses Thm. 1 in [Loureiro et al., 2021] or Thm 2.1 in Loureiro et al. [2023].

The novelty of our approach consists in adapting these results to the bootstrap situation by introducing sample weights $\boldsymbol{p}$ and studying the performance of GAMP for several estimators. The properties of the estimators are given by the distribution on the weights $\boldsymbol{p}$. All previous proof still trivially apply: indeed the state evolution theorems generalize to vector estimations Javanmard and Montanari [2013], and, since GAMP is applied to two problems in parallel, the convergence guarantees still independently apply to each of them. A similar strategy was used in Loureiro et al. [2023].

Consider a convex loss function $\ell$ and regularizer $r$, and the following empirical risk minimization problem

$$(\hat{\boldsymbol{\theta}}_1, \ldots, \hat{\boldsymbol{\theta}}_B) = \arg \min_{\boldsymbol{\theta}_1, \ldots, \boldsymbol{\theta}_B \in \mathbb{R}^d} \mathcal{L}(\boldsymbol{\theta}_1, \ldots, \boldsymbol{\theta}_B) \tag{34}$$

where

$$\mathcal{L}(\boldsymbol{\theta}_1, \ldots, \boldsymbol{\theta}_B) := \sum_{\mu=1}^{n} \ell_{\boldsymbol{p}}(y_\mu, \boldsymbol{\theta}_1^\top \boldsymbol{x}_\mu, \ldots, \boldsymbol{\theta}_B^\top \boldsymbol{x}_\mu) + \sum_{b=1}^{B} r(\boldsymbol{\theta}_b) \tag{35}$$

---
**Algorithm 1** GAMP with sample weights
---

**Input:** $\boldsymbol{X} \in \mathbb{R}^{n \times d}$, $\boldsymbol{y} \in \mathbb{R}^n$, and $\boldsymbol{p}_\mu \in \mathbb{R}^B$ for $1 \leq \mu \leq n$

**Initialize:** $\boldsymbol{g}_{\text{out}\,\mu}^{(0)} = \boldsymbol{0}$ for $1 \leq \mu \leq n$, $\quad \boldsymbol{A}_i^{(0)} = \boldsymbol{I}_B$ for $1 \leq i \leq d$

**Initialize:** $\hat{\boldsymbol{\theta}}_i^{(1)} \in \mathbb{R}^B$ and $\hat{\boldsymbol{C}}_i^{(1)} \in \mathbb{R}^{B \times B}$ for $1 \leq i \leq d$

**Repeat for** $t = 1, 2, \ldots$**:**

    // Update of the means $\boldsymbol{\omega}_\mu \in \mathbb{R}^B$ and covariances $\boldsymbol{V}_\mu \in \mathcal{S}_B^+$ for $1 \leq \mu \leq n$:

    $\boldsymbol{\omega}_\mu^{(t)} = \sum_{i=1}^d X_{\mu,i}\hat{\boldsymbol{\theta}}_i^{(t)} - X_{\mu,i}^2 \left(\boldsymbol{A}_i^{(t-1)}\right)^{-1} \hat{\boldsymbol{C}}_i^{(t)} \boldsymbol{A}_i^{(t-1)} \boldsymbol{g}_{\text{out}\,\mu}^{(t-1)} \mid \boldsymbol{V}_\mu^{(t)} = \sum_{i=1}^d X_{\mu,i}^2 \hat{\boldsymbol{C}}_i^{(t)}$

    // Update of $\boldsymbol{g}_{\text{out}\,\mu}$ and $\partial_\omega \boldsymbol{g}_{\text{out}\,\mu}$ for $1 \leq \mu \leq n$ :

    $\boldsymbol{g}_{\text{out}\,\mu}^{(t)} = \boldsymbol{g}_{\text{out}} \left(\boldsymbol{\omega}_\mu^{(t)}, y_\mu, \boldsymbol{V}_\mu^{(t)}, \boldsymbol{p}_\mu\right) \mid \partial_{\boldsymbol{\omega}} \boldsymbol{g}_{\text{out}\,\mu}^{(t)} = \partial_{\boldsymbol{\omega}} \boldsymbol{g}_{\text{out}} \left(\boldsymbol{\omega}_\mu^{(t)}, y_\mu, \boldsymbol{V}_\mu^{(t)}, \boldsymbol{p}_\mu\right)$

    // Update of means $\boldsymbol{b}_i \in \mathbb{R}^B$ and covariances $\boldsymbol{A}_i \in \mathbb{R}^{B \times B}$ for $1 \leq i \leq d$ :

    $\boldsymbol{A}_i^{(t)} = -\sum_{\mu=1}^n X_{\mu,i}^2 \partial_\omega \boldsymbol{g}_{\text{out}\,\mu}^{(t)} \mid \boldsymbol{b}_i^{(t)} = \boldsymbol{A}_i^{(t)} \hat{\boldsymbol{\theta}}_i^{(t)} + \sum_{\mu=1}^n X_{\mu,i} \boldsymbol{g}_{\text{out}\,\mu}^{(t)}$

    // Update of the estimated marginals $\hat{\boldsymbol{\theta}}_i \in \mathbb{R}^B$ and $\hat{\boldsymbol{C}}_i \in \mathbb{R}^{B \times B}$ for $1 \leq i \leq d$ :

    $\hat{\boldsymbol{\theta}}_i^{(t+1)} = \boldsymbol{f}_a(\boldsymbol{b}_i^{(t)}, \boldsymbol{A}_i^{(t)}) \mid \hat{\boldsymbol{C}}_i^{(t+1)} = \partial_{\boldsymbol{b}} \boldsymbol{f}_a(\boldsymbol{b}_i^{(t)}, \boldsymbol{A}_i^{(t)})$

**Until convergence**

**Output:** $\hat{\boldsymbol{\theta}}_1, \ldots, \hat{\boldsymbol{\theta}}_d$ and $\hat{\boldsymbol{C}}_1, \ldots, \hat{\boldsymbol{C}}_d$

---

and

$$\ell_{\boldsymbol{p}}(y, z_1, \ldots, z_B) := \sum_{b=1}^B p_b \ell(y, z_b) \tag{36}$$

We define a *channel function* associated to the function $\ell$ :

$$\boldsymbol{g}_{\text{out}}(y, \boldsymbol{\omega}, \boldsymbol{V}, \boldsymbol{p}) = \boldsymbol{V}^{-1} \left(\text{prox}_{\boldsymbol{V}, \ell_{\boldsymbol{p}}(y, \cdot)}(\boldsymbol{\omega}) - \boldsymbol{\omega}\right), \tag{37}$$

where the proximal operator is

$$\text{prox}_{\boldsymbol{V}, \ell_{\boldsymbol{p}}(y, \cdot)}(\boldsymbol{\omega}) = \arg \min_{\boldsymbol{z} \in \mathbb{R}^B} \left(\frac{1}{2}(\boldsymbol{z} - \boldsymbol{\omega})^\top \boldsymbol{V}^{-1}(\boldsymbol{z} - \boldsymbol{\omega}) + \ell_{\boldsymbol{p}}(y, \boldsymbol{z})\right). \tag{38}$$

Let us also define the *denoising function* associated to the regularizer $r$:

$$\boldsymbol{f}_a(\boldsymbol{b}, \boldsymbol{A}) = \text{prox}_{\boldsymbol{A}^{-1}, r}(\boldsymbol{A}^{-1} \boldsymbol{b}) = \arg \min_{\boldsymbol{z} \in \mathbb{R}^B} \left(\frac{1}{2}(\boldsymbol{z} - \boldsymbol{A}^{-1}\boldsymbol{b})^\top \boldsymbol{A}(\boldsymbol{z} - \boldsymbol{A}^{-1}\boldsymbol{b}) + r(\boldsymbol{z})\right). \tag{39}$$

Using Algorithm 1 with this choice of channel and denoising functions returns a set of vectors $\hat{\boldsymbol{\theta}}_1, \cdots, \hat{\boldsymbol{\theta}}_d \in \mathbb{R}^B$, where $\hat{\boldsymbol{\theta}}_i$ contains the $B$ estimates for $\theta_{\star i}$. Hence, these vectors allow to solve the minimization problem (34).

**Intuition of GAMP algorithm**    We are interested in solving the minimization problem (34), which is equivalent to sampling from the distribution

$$p(\boldsymbol{\theta}_1, \ldots, \boldsymbol{\theta}_B) \propto \exp\left(-\beta \mathcal{L}\left(\boldsymbol{\theta}_1, \ldots, \boldsymbol{\theta}_B\right)\right) = \exp\left(-\beta \left(\sum_{\mu=1}^n \ell_{\boldsymbol{p}}(y_\mu, \boldsymbol{\theta}_1^\top \boldsymbol{x}_\mu, \ldots, \boldsymbol{\theta}_B^\top \boldsymbol{x}_\mu) + \sum_{b=1}^B r(\boldsymbol{\theta}_b)\right)\right) \tag{40}$$

in the limit $\beta \to \infty$. Sampling the distribution on a graphical model can be used with Belief Propagation, which iterates messages between different nodes (here the coordinates $\boldsymbol{\theta}_{ij}$ for $i \leq B, j \leq d$). However in high dimensions, Belief Propagation is intractable as it involves computing $d$-dimensional integrals. To alleviate this issue, GAMP only computes the first two moments of the different messages. In the high-dimensional limit, the output of GAMP coincides with the true minimizer of (34).

Similarly to our work, in Aubin et al. [2019], the authors introduce a GAMP algorithm for for a generic coupled system of estimates. They provide a detailed analysis of GAMP and its state evolution to track its behaviour in the asymptotic limit.

## A.1 STATE EVOLUTION EQUATIONS

In this section, we inspect the behavior of Algorithm 1 in the $n, d \to \infty$ limit and derive the asymptotic distribution of $\hat{\boldsymbol{\theta}}_1, \dots, \hat{\boldsymbol{\theta}}_d$. To do so, we start from the more convenient relaxed Belief Propagation (rBP) equations, which are very close to GAMP. In the high-dimensional limit, rBP and GAMP are equivalent. The rBP equations are written,

$$
\begin{cases} \boldsymbol{\omega}_{\mu \to i}^{(t)} &= \sum_{j \neq i} X_{\mu,j} \hat{\boldsymbol{\theta}}_{j \to \mu}^{(t)} \\ \boldsymbol{V}_{\mu \to i}^{(t)} &= \sum_{j \neq i} X_{\mu,j}^2 \hat{\boldsymbol{C}}_{j \to \mu}^{(t)} \end{cases}, \qquad \begin{cases} \boldsymbol{g}_{\text{out} \mu \to i}^{(t)} &= \boldsymbol{g}_{\text{out}}(y_\mu, \boldsymbol{\omega}_{\mu \to i}^{(t)}, \boldsymbol{V}_{\mu \to i}^{(t)}, \boldsymbol{p}_\mu) \\ \partial \boldsymbol{g}_{\text{out} \mu \to i}^{(t)} &= \partial_{\boldsymbol{\omega}} \boldsymbol{g}_{\text{out}}(y_\mu, \boldsymbol{\omega}_{\mu \to i}^{(t)}, \boldsymbol{V}_{\mu \to i}^{(t)}, \boldsymbol{p}_\mu) \end{cases} \tag{41}
$$

$$
\begin{cases} \boldsymbol{b}_{\mu \to i}^{(t)} &= \sum_{\nu \neq \mu} X_{\nu,i} \boldsymbol{g}_{\text{out} \nu \to i}^{(t)} \\ \boldsymbol{A}_{\mu \to i}^{(t)} &= -\sum_{\nu \neq \mu} X_{\nu,i}^2 \partial \boldsymbol{g}_{\text{out} \nu \to i}^{(t)} \end{cases}, \qquad \begin{cases} \hat{\boldsymbol{\theta}}_{i \to \mu}^{(t)} &= \boldsymbol{f}_a(\boldsymbol{b}_{i \to \mu}^{(t)}, \boldsymbol{A}_{i \to \mu}^{(t)}) \\ \hat{\boldsymbol{C}}_{i \to \mu}^{(t)} &= \partial_{\boldsymbol{b}} \boldsymbol{f}_a(\boldsymbol{b}_{i \to \mu}^{(t)}, \boldsymbol{A}_{i \to \mu}^{(t)}). \end{cases} \tag{42}
$$

It turns out that the average asymptotic behavior of these equations can be tracked with some overlap parameters defined as follows:

$$
\boldsymbol{m}^{(t)} \equiv \lim_{d \to \infty} \frac{1}{d} \sum_{i=1}^d \hat{\boldsymbol{\theta}}_i^{(t)} \boldsymbol{\theta}_\star^\top, \qquad\qquad \boldsymbol{Q}^{(t)} \equiv \lim_{d \to \infty} \frac{1}{d} \sum_{i=1}^d \hat{\boldsymbol{\theta}}_i^{(t)} \hat{\boldsymbol{\theta}}_i^{(t)\top} \tag{43}
$$

$$
\boldsymbol{V}^{(t)} \equiv \lim_{d \to \infty} \frac{1}{d} \sum_{i=1}^d \hat{\boldsymbol{C}}_i^{(t)}, \qquad\qquad \rho = \lim_{d \to \infty} \frac{\|\boldsymbol{\theta}_\star\|^2}{d}. \tag{44}
$$

To derive the asymptotic behavior of these overlap parameters, we compute the overlap distributions starting from the rBP equations above.

### A.1.1 Messages Distribution

For convenience, let us define $z_\mu \equiv \sum_{i=1}^d X_{\mu,i} \theta_{\star i} = \boldsymbol{X}_\mu^\top \boldsymbol{\theta}_\star$ and $z_{\mu \to i} \equiv \frac{1}{d} \sum_{j \neq i} X_{\mu,i} \theta_{\star j}$.

**Distribution of $(z_\mu, \boldsymbol{\omega}_{\mu \to i}^{(t)})$** By the Central Limit Theorem, since $(z_\mu, \boldsymbol{\omega}_{\mu \to i}^{(t)})$ are the sum of independent variables, they follow Gaussian distributions in the $d \to \infty$ limit. Therefore, we only need to compute their means, variances, and cross-correlation. Recall that from our assumptions, the random variables $X_{\mu,j}$ are i.i.d. zero-mean Gaussian with variance $1/d$. Hence, the first and second-order statistics read

$$
\mathbb{E}[z_\mu] = \boldsymbol{\theta}_\star^\top \mathbb{E}[\boldsymbol{X}_\mu] = 0 \tag{45}
$$

$$
\mathbb{E}[z_\mu^2] = \sum_{i,j=1}^d \mathbb{E}[X_{\mu,i} X_{\mu,j}] \theta_{\star i} \theta_{\star j} = \sum_{i,j=1}^d \frac{1}{d} \delta_{ij} \theta_{\star i} \theta_{\star j} = \frac{\|\boldsymbol{\theta}_\star\|^2}{d} \overset{d \to \infty}{\longrightarrow} \rho \tag{46}
$$

$$
\mathbb{E}\left[\boldsymbol{\omega}_{\mu \to i}^{(t)}\right] = \sum_{j \neq i} \mathbb{E}[X_{\mu,j}] \hat{\boldsymbol{\theta}}_{j \to \mu}^{(t)} = \boldsymbol{0} \tag{47}
$$

$$
\mathbb{E}\left[\boldsymbol{\omega}_{\mu \to i}^{(t)} (\boldsymbol{\omega}_{\mu \to i}^{(t)})^\top\right] = \sum_{j \neq i}^d \sum_{k \neq i}^d \mathbb{E}[X_{\mu,j} X_{\mu,k}] \hat{\boldsymbol{\theta}}_{j \to \mu}^{(t)} \hat{\boldsymbol{\theta}}_{k \to \mu}^{(t)\top} = \frac{1}{d} \sum_{j \neq i}^d \hat{\boldsymbol{\theta}}_{j \to \mu}^{(t)} \hat{\boldsymbol{\theta}}_{k \to \mu}^{(t)\top} \tag{48}
$$

$$
= \frac{1}{d} \sum_{j=1}^d \hat{\boldsymbol{\theta}}_{j \to \mu}^{(t)} \hat{\boldsymbol{\theta}}_{j \to \mu}^{(t)\top} - \frac{1}{d} \hat{\boldsymbol{\theta}}_{i \to \mu}^{(t)} \hat{\boldsymbol{\theta}}_{i \to \mu}^{(t)\top} \overset{d \to \infty}{\longrightarrow} \boldsymbol{Q}^{(t)} \tag{49}
$$

$$
\mathbb{E}\left[z_\mu \boldsymbol{\omega}_{\mu \to i}^{(t)}\right] = \sum_{j=1}^d \sum_{k \neq i}^d \mathbb{E}[X_{\mu,j} X_{\mu,k}] \hat{\boldsymbol{\theta}}_{k \to \mu}^{(t)} \boldsymbol{\theta}_{\star j} = \frac{1}{d} \sum_{j \neq i} \hat{\boldsymbol{\theta}}_{j \to \mu}^{(t)} \boldsymbol{\theta}_\star \tag{50}
$$

$$
= \frac{1}{d} \sum_{j=1}^d \hat{\boldsymbol{\theta}}_{j \to \mu}^{(t)} \boldsymbol{\theta}_\star - \frac{1}{d} \hat{\boldsymbol{\theta}}_{i \to \mu}^{(t)} \boldsymbol{\theta}_\star \overset{d \to \infty}{\longrightarrow} \boldsymbol{m}^{(t)} \tag{51}
$$

In summary, in the $d \to \infty$ limit :

$$\left(z_\mu, \boldsymbol{\omega}_{\mu \to i}^{(t)}\right) \sim \mathcal{N}\left(0, \begin{bmatrix} \rho & \boldsymbol{m}^{(t)\top} \\ \boldsymbol{m}^{(t)} & \boldsymbol{Q}^{(t)} \end{bmatrix}\right) \tag{52}$$

**Concentration of $\boldsymbol{V}_{\mu \to i}^{(t)}$** In the asymptotic limit, the variances $\boldsymbol{V}_{\mu \to i}^{(t)}$ concentrate around their means, which equates

$$\mathbb{E}\left[\boldsymbol{V}_{\mu \to i}^{(t)}\right] = \sum_{j \neq i}^{d} \mathbb{E}\left[X_{\mu,j}^2\right] \hat{\boldsymbol{C}}^{(t)} = \frac{1}{d} \sum_{j \neq i} \hat{\boldsymbol{C}}_j^{(t)} = \frac{1}{d} \sum_{j=1}^{d} \hat{\boldsymbol{C}}_j^{(t)} - \frac{1}{d} \hat{\boldsymbol{C}}_i^{(t)} \stackrel{d \to \infty}{\longrightarrow} \boldsymbol{V}^{(t)} \tag{53}$$

**Distribution of $\boldsymbol{b}_{\mu \to i}^{(t)}$** Recall from our setting that for a given input $\boldsymbol{x}_\mu$, the corresponding label is distributed as $y_\mu \sim p(\cdot | z_\mu)$. In fact, one can equivalently write $y^\mu = \varphi_0(z_\mu)$ for some (random) function $\varphi_0$. For example, the choice $\varphi_0(x) = x + \sqrt{\Delta}\xi$ corresponds to the linear regression, where $\xi \sim \mathcal{N}(0,1)$ is Gaussian noise scaled by a variance $\Delta \geq 0$. With this representation for $y_\mu$, we have

$$\boldsymbol{b}_{\mu \to i}^{(t)} = \sum_{\nu \neq \mu} X_{\nu,i} \boldsymbol{g}_{\text{out}}(\varphi_0(z_\nu), \boldsymbol{\omega}_{\nu \to i}^{(t)}, \boldsymbol{V}_{\nu \to i}^{(t)}, \boldsymbol{p}_\nu) \tag{54}$$

$$= \sum_{\nu \neq \mu} X_{\nu,i} \boldsymbol{g}_{\text{out}}(\varphi_0(z_{\nu \to i} + \theta_{\star i} X_{\nu,i}), \boldsymbol{\omega}_{\nu \to i}^{(t)}, \boldsymbol{V}_{\nu \to i}^{(t)}, \boldsymbol{p}_\nu) \tag{55}$$

$$= \sum_{\nu \neq \mu} X_{\nu,i} \boldsymbol{g}_{\text{out}}(\varphi_0(z_{\nu \to i}), \boldsymbol{\omega}_{\nu \to i}^{(t)}, \boldsymbol{V}_{\nu \to i}^{(t)}, \boldsymbol{p}_\nu) + X_{\nu,i}^2 \theta_{\star i} \partial_z \boldsymbol{g}_{\text{out}}(\varphi_0(z_{\nu \to i}), \boldsymbol{\omega}_{\nu \to i}^{(t)}, \boldsymbol{V}_{\nu \to i}^{(t)}, \boldsymbol{p}_\nu) + O(d^{-3/2}), \tag{56}$$

where in the last equality we have expanded the denoising function at leading order. Taking expectation on both sides yields

$$\mathbb{E}[\boldsymbol{b}_{\mu \to i}^{(t)}] = \frac{\theta_{\star i}}{d} \sum_{\nu \neq \mu} \partial_z \boldsymbol{g}_{\text{out}}(\varphi_0(z_{\nu \to i}), \boldsymbol{\omega}_{\nu \to i}^{(t)}, \boldsymbol{V}_{\nu \to i}^{(t)}, \boldsymbol{p}_\nu) + O(d^{-3/2}) \tag{57}$$

$$= \frac{\theta_{\star i}}{d} \sum_{\nu = 1}^{n} \partial_z \boldsymbol{g}_{\text{out}}(\varphi_0(z_{\nu \to i}), \boldsymbol{\omega}_{\nu \to i}^{(t)}, \boldsymbol{V}_{\nu \to i}^{(t)}, \boldsymbol{p}_\nu) - \frac{\theta_{\star i}}{d} \partial_z \boldsymbol{g}_{\text{out}}(\varphi_0(z_{\mu \to i}), \boldsymbol{\omega}_{\mu \to i}^{(t)}, \boldsymbol{V}_{\mu \to i}^{(t)}, \boldsymbol{p}_\mu) + O(d^{-3/2}), \tag{58}$$

Note that as $d \to \infty$, it follows from our computations above that for all $\nu$, $(z_{\nu \to i}, \boldsymbol{\omega}_{\nu \to i}^{(t)})$ are identically distributed according to Equation (52). Consequently, by the Law of Large Numbers,

$$\frac{n}{d} \cdot \frac{1}{n} \sum_{\nu = 1}^{n} \partial_z \boldsymbol{g}_{\text{out}}(\varphi_0(z_{\nu \to i}), \boldsymbol{\omega}_{\nu \to i}^{(t)}, \boldsymbol{V}_{\nu \to i}^{(t)}, \boldsymbol{p}_\nu) \stackrel{n,d \to \infty}{\longrightarrow} \alpha \mathbb{E}_{(z,\omega),\boldsymbol{p}}\left[\partial_z \boldsymbol{g}_{\text{out}}(\varphi_0(z), \boldsymbol{\omega}, \boldsymbol{V}^{(t)}, \boldsymbol{p})\right] \equiv \hat{\boldsymbol{m}}^{(t)}, \tag{59}$$

from which we find that

$$\mathbb{E}[\boldsymbol{b}_{\mu \to i}^{(t)}] \stackrel{n,d \to \infty}{\longrightarrow} \theta_{\star i} \hat{\boldsymbol{m}}^{(t)}. \tag{60}$$

The second moment can be computed in a similar fashion:

$$\mathbb{E}[\boldsymbol{b}_{\mu \to i}^{(t)} \boldsymbol{b}_{\mu \to i}^{(t)\top}] = \sum_{\nu \neq \mu} \sum_{\kappa \neq \mu} \mathbb{E}[X_{\nu,i} X_{\kappa,i}] \boldsymbol{g}_{\text{out}}(\varphi_0(z_\nu), \boldsymbol{\omega}_{\nu \to i}^{(t)}, \boldsymbol{V}_{\nu \to i}^{(t)}, \boldsymbol{p}_\nu) \boldsymbol{g}_{\text{out}}(\varphi_0(z_\kappa), \boldsymbol{\omega}_{\kappa \to i}^{(t)}, \boldsymbol{V}_{\kappa \to i}^{(t)}, \boldsymbol{p}_\kappa)^\top \tag{61}$$

$$= \frac{1}{d} \sum_{\nu \neq \mu} \boldsymbol{g}_{\text{out}}(\varphi_0(z_{\nu \to i}), \boldsymbol{\omega}_{\nu \to i}^{(t)}, \boldsymbol{V}_{\nu \to i}^{(t)}, \boldsymbol{p}_\nu) \boldsymbol{g}_{\text{out}}(\varphi_0(z_{\nu \to i}), \boldsymbol{\omega}_{\nu \to i}^{(t)}, \boldsymbol{V}_{\nu \to i}^{(t)}, \boldsymbol{p}_\nu)^\top + O(d^{-2}) \tag{62}$$

$$= \frac{1}{d} \sum_{\nu = 1}^{n} \boldsymbol{g}_{\text{out}}(\varphi_0(z_{\nu \to i}), \boldsymbol{\omega}_{\nu \to i}^{(t)}, \boldsymbol{V}_{\nu \to i}^{(t)}, \boldsymbol{p}_\nu) \boldsymbol{g}_{\text{out}}(\varphi_0(z_{\nu \to i}), \boldsymbol{\omega}_{\nu \to i}^{(t)}, \boldsymbol{V}_{\nu \to i}^{(t)}, \boldsymbol{p}_\nu)^\top + O(d^{-2}) \tag{63}$$

$$\stackrel{n,d \to \infty}{\longrightarrow} \alpha \mathbb{E}_{(z,\omega^{(t)}),\boldsymbol{p}}\left[\boldsymbol{g}_{\text{out}}(\varphi_0(z), \boldsymbol{\omega}^{(t)}, \boldsymbol{V}^{(t)}, \boldsymbol{p}) \boldsymbol{g}_{\text{out}}(\varphi_0(z), \boldsymbol{\omega}^{(t)}, \boldsymbol{V}^{(t)}, \boldsymbol{p})^\top\right] \equiv \hat{\boldsymbol{Q}}^{(t)}. \tag{64}$$

Hence, $\boldsymbol{b}_{\mu \to i}^{(t)} = \theta_{\star i} \hat{\boldsymbol{m}}^{(t)} + \left(\hat{\boldsymbol{Q}}^{(t)}\right)^{1/2} \boldsymbol{\xi}$ with $\boldsymbol{\xi} \sim \mathcal{N}(\boldsymbol{0}, \boldsymbol{I}_B)$.

**Concentration of $A^{(t)}_{\mu \to i}$** It remains to show that the covariances $A^{(t)}_{\mu \to i}$ concentrate. We have

$$A^{(t)}_{\mu \to i} = -\sum_{\nu \neq \mu} X^2_{\nu,i} \partial_{\boldsymbol{\omega}} \boldsymbol{g}_{\text{out}}(y_\nu, \boldsymbol{\omega}^{(t)}_{\nu \to i}, \boldsymbol{V}^{(t)}_{\nu \to i}, \boldsymbol{p}_\nu) \tag{65}$$

$$= -\sum_{\nu \neq \mu} X^2_{\nu,i} \partial_{\boldsymbol{\omega}} \boldsymbol{g}_{\text{out}}(\varphi_0(z_\nu), \boldsymbol{\omega}^{(t)}_{\nu \to i}, \boldsymbol{V}^{(t)}_{\nu \to i}, \boldsymbol{p}_\nu) \tag{66}$$

$$= -\sum_{\nu \neq \mu} X^2_{\nu,i} \partial_{\boldsymbol{\omega}} \boldsymbol{g}_{\text{out}}(\varphi_0(z_{\nu \to i}), \boldsymbol{\omega}^{(t)}_{\nu \to i}, \boldsymbol{V}^{(t)}_{\nu \to i}, \boldsymbol{p}_\nu) + O(d^{-3/2}). \tag{67}$$

Taking the expectation gives

$$\mathbb{E}[A^{(t)}_{\mu \to i}] = -\frac{1}{d} \sum_{\nu \neq \mu} \partial_{\boldsymbol{\omega}} \boldsymbol{g}_{\text{out}}(\varphi_0(z_{\nu \to i}), \boldsymbol{\omega}^{(t)}_{\nu \to i}, \boldsymbol{V}^{(t)}_{\nu \to i}, \boldsymbol{p}_\nu) + O(d^{-3/2}) \tag{68}$$

$$= -\frac{1}{d} \sum_{\nu=1}^{n} \partial_{\boldsymbol{\omega}} \boldsymbol{g}_{\text{out}}(\varphi_0(z_{\nu \to i}), \boldsymbol{\omega}^{(t)}_{\nu \to i}, \boldsymbol{V}^{(t)}_{\nu \to i}, \boldsymbol{p}_\nu) - \frac{1}{d} \partial_{\boldsymbol{\omega}} \boldsymbol{g}_{\text{out}}(\varphi_0(z_{\mu \to i}), \boldsymbol{\omega}^{(t)}_{\mu \to i}, \boldsymbol{V}^{(t)}_{\mu \to i}, \boldsymbol{p}_\mu) + O(d^{-3/2}) \tag{69}$$

$$\overset{n,d \to \infty}{\longrightarrow} -\alpha \mathbb{E}_{(z,\boldsymbol{\omega}^{(t)}),\boldsymbol{p}} \left[ \partial_{\boldsymbol{\omega}} \boldsymbol{g}_{\text{out}}(\varphi_0(z), \boldsymbol{\omega}^{(t)}, \boldsymbol{V}^{(t)}, \boldsymbol{p}) \right] \equiv \hat{\boldsymbol{V}}^{(t)} \tag{70}$$

### A.1.2 Summary

Having shown the distribution of messages and concentration, we are ready to characterize the asymptotic distribution of the estimator:

$$\hat{\boldsymbol{\theta}}_i \sim \boldsymbol{f}_a \left( \theta_{\star i} \hat{\boldsymbol{m}}^{(t)} + \left( \hat{\boldsymbol{Q}}^{(t)} \right)^{1/2} \boldsymbol{\xi}, \hat{\boldsymbol{V}}^{(t)} \right) \quad \forall i \in \{1, \ldots, d\}, \tag{71}$$

where $\boldsymbol{\xi} \sim \mathcal{N}(\boldsymbol{0}, \boldsymbol{I}_B)$.

From that, the definitions of overlaps in Equation (43) at time $t+1$, and the message distributions, we obtain the state-evolution equations of the GAMP algorithm described in Algorithm 1:

$$\begin{cases} \boldsymbol{m}^{(t+1)} &= \mathbb{E}_{\theta_\star, \boldsymbol{\xi}} \left[ \boldsymbol{f}_a \left( \hat{\boldsymbol{m}} \theta_\star + \sqrt{\hat{\boldsymbol{Q}}^{(t)}} \boldsymbol{\xi}, \hat{\boldsymbol{V}}^{(t)} \right) \theta_\star \right] \\ \boldsymbol{Q}^{(t+1)} &= \mathbb{E}_{\theta_\star, \boldsymbol{\xi}} \left[ \boldsymbol{f}_a \left( \hat{\boldsymbol{m}} \theta_\star + \sqrt{\hat{\boldsymbol{Q}}^{(t)}} \boldsymbol{\xi}, \hat{\boldsymbol{V}}^{(t)} \right) \boldsymbol{f}_a \left( \hat{\boldsymbol{m}} \theta_\star + \sqrt{\hat{\boldsymbol{Q}}^{(t)}} \boldsymbol{\xi}, \hat{\boldsymbol{V}}^{(t)} \right)^\top \right] \\ \boldsymbol{V}^{(t+1)} &= \mathbb{E}_{\theta_\star, \boldsymbol{\xi}} \left[ \partial_{\boldsymbol{b}} \boldsymbol{f}_a \left( \hat{\boldsymbol{m}} \theta_\star + \sqrt{\hat{\boldsymbol{Q}}^{(t)}} \boldsymbol{\xi}, \hat{\boldsymbol{V}}^{(t)} \right) \right] \end{cases} \tag{72}$$

where $\boldsymbol{\xi} \sim \mathcal{N}(0, \boldsymbol{I}_B)$, and

$$\begin{cases} \hat{\boldsymbol{m}}^{(t)} &= \alpha \mathbb{E}_{(z,\boldsymbol{\omega}^{(t)}),\boldsymbol{p}} \left[ \partial_z \boldsymbol{g}_{\text{out}}(\varphi_0(z), \boldsymbol{\omega}^{(t)}, \boldsymbol{V}^{(t)}, \boldsymbol{p}) \right] \\ \hat{\boldsymbol{Q}}^{(t)} &= \alpha \mathbb{E}_{(z,\boldsymbol{\omega}^{(t)}),\boldsymbol{p}} \left[ \boldsymbol{g}_{\text{out}}(\varphi_0(z), \boldsymbol{\omega}^{(t)}, \boldsymbol{V}^{(t)}, \boldsymbol{p}) \boldsymbol{g}_{\text{out}}(\varphi_0(z), \boldsymbol{\omega}^{(t)}, \boldsymbol{V}^{(t)}, \boldsymbol{p})^\top \right], \\ \hat{\boldsymbol{V}}^{(t)} &= -\alpha \mathbb{E}_{(z,\boldsymbol{\omega}^{(t)}),\boldsymbol{p}} \left[ \partial_{\boldsymbol{\omega}} \boldsymbol{g}_{\text{out}}(\varphi_0(z), \boldsymbol{\omega}^{(t)}, \boldsymbol{V}^{(t)}, \boldsymbol{p}) \right] \end{cases} \tag{73}$$

where $(z, \boldsymbol{\omega}^{(t)}) \sim \mathcal{N}\left( 0, \begin{bmatrix} \rho & \boldsymbol{m}^{(t)\top} \\ \boldsymbol{m}^{(t)} & \boldsymbol{Q}^{(t)} \end{bmatrix} \right)$.

Let us note that the overlaps $\hat{\boldsymbol{m}}^{(t)}, \hat{\boldsymbol{Q}}^{(t)}, \hat{\boldsymbol{V}}^{(t)}$ can be written slightly differently. For that, first notice that since $(z, \boldsymbol{\omega}^{(t)})$ is Gaussian, so is $z$ conditioned on $\boldsymbol{\omega}^{(t)}$, and in particular $z|\boldsymbol{\omega}^{(t)} \sim \mathcal{N}(\mu_\star(\boldsymbol{\omega}^{(t)}), v_\star)$ with $\mu_\star(\boldsymbol{\omega}^{(t)}) = (\boldsymbol{m}^{(t)})^\top (\boldsymbol{Q}^{(t)})^{-1} \boldsymbol{\omega}^{(t)}$, $v_\star = \rho - (\boldsymbol{m}^{(t)})^\top (\boldsymbol{Q}^{(t)})^{-1} \boldsymbol{m}^{(t)}$. Moreover, using that $p(y|z) = \delta(y - \varphi_0(z))$, we have for an arbitrary function $\boldsymbol{f}$:

$\mathbb{R} \times \mathbb{R}^B \to \mathbb{R}^B$ that

$$\mathbb{E}_{(z,\boldsymbol{\omega}^{(t)})}\left[f(\varphi_0(z), \boldsymbol{\omega}^{(t)})\right] = \mathbb{E}_{\boldsymbol{\omega}^{(t)}}\left[\mathbb{E}_{z|\boldsymbol{\omega}^{(t)}}\left[\boldsymbol{f}(\varphi_0(z), \boldsymbol{\omega}^{(t)})\right]\right] \tag{74}$$

$$= \mathbb{E}_{\boldsymbol{\omega}^{(t)}}\left[\int \mathrm{d}z \mathcal{N}(z|\mu_\star(\boldsymbol{\omega}^{(t)}), v_\star)\boldsymbol{f}(\varphi_0(z), \boldsymbol{\omega}^{(t)})\right] \tag{75}$$

$$= \mathbb{E}_{\boldsymbol{\omega}^{(t)}}\left[\int \mathrm{d}z \mathcal{N}(z|\mu_\star(\boldsymbol{\omega}^{(t)}), v_\star)\int \mathrm{d}y p(y|z)\boldsymbol{f}(y, \boldsymbol{\omega}^{(t)})\right] \tag{76}$$

$$= \mathbb{E}_{\boldsymbol{\omega}^{(t)}}\left[\int \mathrm{d}y \mathcal{Z}_0(y, \mu_\star(\boldsymbol{\omega}^{(t)}), v_\star)\boldsymbol{f}(y, \boldsymbol{\omega}^{(t)})\right], \tag{77}$$

where we have defined $\mathcal{Z}_0(y, \mu, v) \equiv \int \mathrm{d}z \mathcal{N}(z|\mu, v)p(y|z)$. Consequently, we can rewrite

$$\begin{cases} \hat{m}^{(t)} &= \alpha \mathbb{E}_{\boldsymbol{\omega}^{(t)}, \boldsymbol{p}}\left[\int \mathrm{d}y \partial_\mu \mathcal{Z}_0(y, \mu_\star(\boldsymbol{\omega}^{(t)}), v_\star) \cdot \boldsymbol{g}_{\mathrm{out}}(y, \boldsymbol{\omega}^{(t)}, \boldsymbol{V}^{(t)}, \boldsymbol{p})\right] \\ \hat{Q}^{(t)} &= \alpha \mathbb{E}_{\boldsymbol{\omega}^{(t)}, \boldsymbol{p}}\left[\int \mathrm{d}y \mathcal{Z}_0(y, \mu_\star(\boldsymbol{\omega}^{(t)}), v_\star) \cdot \boldsymbol{g}_{\mathrm{out}}(y, \boldsymbol{\omega}^{(t)}, \boldsymbol{V}^{(t)}, \boldsymbol{p})\boldsymbol{g}_{\mathrm{out}}(y, \boldsymbol{\omega}^{(t)}, \boldsymbol{V}^{(t)}, \boldsymbol{p})^\top\right], \\ \hat{V}^{(t)} &= -\alpha \mathbb{E}_{\boldsymbol{\omega}^{(t)}, \boldsymbol{p}}\left[\int \mathrm{d}y \mathcal{Z}_0(y, \mu_\star(\boldsymbol{\omega}^{(t)}), v_\star) \cdot \partial_{\boldsymbol{\omega}} \boldsymbol{g}_{\mathrm{out}}(\varphi_0(z), \boldsymbol{\omega}^{(t)}, \boldsymbol{V}^{(t)}, \boldsymbol{p})\right] \end{cases} \tag{78}$$

where $\boldsymbol{\omega}^{(t)} \sim \mathcal{N}(\boldsymbol{0}, \boldsymbol{Q}^{(t)})$.

### A.1.3 Self-Consistent Equations

In the limit $t \to \infty$, the state-evolution equations derived above yield a set of self-consistent equations:

$$\begin{cases} \boldsymbol{m} &= \mathbb{E}_{\theta_\star, \boldsymbol{\xi}}\left[\boldsymbol{f}_a(\hat{m}\theta_\star + \sqrt{\hat{Q}}\boldsymbol{\xi}, \hat{V})\theta_\star\right] \\ \boldsymbol{Q} &= \mathbb{E}_{\theta_\star, \boldsymbol{\xi}}\left[\left[\boldsymbol{f}_a \boldsymbol{f}_a^\top\right](\hat{m}\theta_\star + \sqrt{\hat{Q}}\boldsymbol{\xi}, \hat{V})\right], \\ \boldsymbol{V} &= \mathbb{E}_{\theta_\star, \boldsymbol{\xi}}\left[\partial_b \boldsymbol{f}_a(\hat{m}\theta_\star + \sqrt{\hat{Q}}\boldsymbol{\xi}, \hat{V})\right] \end{cases} \begin{cases} \hat{m} &= \alpha \mathbb{E}_{\boldsymbol{\omega}, \boldsymbol{p}}\left[\int \mathrm{d}y \partial_\mu \mathcal{Z}_0(y, \mu_\star(\boldsymbol{\omega}), v_\star) \cdot \boldsymbol{g}_{\mathrm{out}}(y, \boldsymbol{\omega}, \boldsymbol{V}, \boldsymbol{p})\right] \\ \hat{Q} &= \alpha \mathbb{E}_{\boldsymbol{\omega}, \boldsymbol{p}}\left[\int \mathrm{d}y \mathcal{Z}_0(y, \mu_\star(\boldsymbol{\omega}), v_\star) \cdot \left[\boldsymbol{g}_{\mathrm{out}} \boldsymbol{g}_{\mathrm{out}}^\top\right](y, \boldsymbol{\omega}, \boldsymbol{V}, \boldsymbol{p})\right] \\ \hat{V} &= -\alpha \mathbb{E}_{\boldsymbol{\omega}, \boldsymbol{p}}\left[\int \mathrm{d}y \mathcal{Z}_0(y, \mu_\star(\boldsymbol{\omega}), v_\star) \cdot \partial_{\boldsymbol{\omega}} \boldsymbol{g}_{\mathrm{out}}(y, \boldsymbol{\omega}, \boldsymbol{V}, \boldsymbol{p})\right] \end{cases} \tag{79}$$

where $\boldsymbol{\xi} \sim \mathcal{N}(0, \boldsymbol{I}_B)$, $\boldsymbol{\omega} \sim \mathcal{N}(\boldsymbol{0}, \boldsymbol{Q})$, and $\mu_\star(\boldsymbol{\omega}) = \boldsymbol{m}^\top \boldsymbol{Q}^{-1}\boldsymbol{\omega}$ and $v_\star = \rho - \boldsymbol{m}^\top \boldsymbol{Q}^{-1}\boldsymbol{m}$ with $\rho = 1/d\|\boldsymbol{\theta}_\star\|_2^2$.

### A.1.4 Channels

**Channel for square loss** When the loss is the square loss $\ell(y, \omega) = \frac{1}{2\Delta}(y - \omega)^2$, we can conveniently write the proximal in a matrix form

$$\mathrm{prox}(y, \boldsymbol{\omega}, \boldsymbol{V}, \boldsymbol{p}) = \arg \min_{\boldsymbol{z} \in \mathbb{R}^B} \frac{1}{2}(\boldsymbol{z} - \boldsymbol{\omega})^\top \boldsymbol{V}^{-1}(\boldsymbol{z} - \boldsymbol{\omega}) + \frac{1}{2\Delta}(\boldsymbol{z} - \boldsymbol{1}_B y)^\top \boldsymbol{P}(\boldsymbol{z} - \boldsymbol{1}_B y), \tag{80}$$

where we have defined $\boldsymbol{P} = \mathrm{Diag}(\boldsymbol{p})$. In that case, the vector $\boldsymbol{z}$ that cancels the derivative of the function to minimize is

$$\boldsymbol{z}_* = \left(\boldsymbol{V}^{-1} + \frac{\boldsymbol{P}}{\Delta}\right)^{-1}\left(\boldsymbol{V}^{-1}\boldsymbol{\omega} + \frac{\boldsymbol{P}}{\Delta}\boldsymbol{1}_B y\right) \tag{81}$$

such that

$$\boldsymbol{g}_{\mathrm{out}}(y, \boldsymbol{\omega}, \boldsymbol{V}, \boldsymbol{p}) = \left(\boldsymbol{I}_B + \frac{\boldsymbol{P}\boldsymbol{V}}{\Delta}\right)^{-1}\frac{\boldsymbol{P}}{\Delta}(\boldsymbol{1}_B y - \boldsymbol{\omega}) \tag{82}$$

$$\partial_{\boldsymbol{\omega}}\boldsymbol{g}_{\mathrm{out}}(y, \boldsymbol{\omega}, \boldsymbol{V}, \boldsymbol{p}) = -\left(\boldsymbol{I}_B + \frac{\boldsymbol{P}\boldsymbol{V}}{\Delta}\right)^{-1}\frac{\boldsymbol{P}}{\Delta} \tag{83}$$

**Channel for logistic loss** In classification tasks one usually uses the logistic loss $\ell(y, z) = \log(1 + e^{-z})$. We thus aim to compute the proximal

$$\text{prox}_{\ell(y,\cdot),\boldsymbol{V}}(\boldsymbol{\omega}) = \arg\min_{\boldsymbol{z} \in \mathbb{R}^B} \sum_{b=1}^{B} p_b \ell(y, z_b) + \frac{1}{2}(\boldsymbol{z} - \boldsymbol{\omega})\boldsymbol{V}^{-1}(\boldsymbol{z} - \boldsymbol{\omega}) \tag{84}$$

We deduce the channel from it. On the other hand, to compute $\partial_{\boldsymbol{\omega}} \boldsymbol{g}_{\text{out}}$, one needs to compute the Hessian of the loss function:

$$\nabla^2 \ell(y, \boldsymbol{z}, \boldsymbol{p}) = \text{Diag}\left(p_1 \sigma'(yz_1), \ldots, p_B \sigma'(yz_B)\right) \tag{85}$$

### A.1.5 Denoiser for $\ell_2$ regularization

In a similar way, the denoiser is written

$$\boldsymbol{f}_a(\boldsymbol{b}, \boldsymbol{A}) = (\lambda \boldsymbol{I}_B + \boldsymbol{A})^{-1} \boldsymbol{b} \tag{86}$$

$$\partial_{\boldsymbol{b}} f_a(\boldsymbol{b}, \boldsymbol{A}) = (\lambda \boldsymbol{I}_B + \boldsymbol{A})^{-1} \tag{87}$$

## A.2 RIDGE REGRESSION

Using the channel for square loss and the denoiser for $\ell_2$ regularization, we can compute the various overlaps for the ridge regression. First, defining $\boldsymbol{R}(\lambda) \equiv (\lambda \boldsymbol{I}_B + \hat{\boldsymbol{V}})^{-1}$, we find that

$$\boldsymbol{m} = \mathbb{E}_{\theta_\star, \boldsymbol{\xi}}\left[\boldsymbol{R}(\lambda)\left(\hat{m}\theta_\star + \sqrt{\hat{Q}}\boldsymbol{\xi}\right)\theta_\star\right] = \boldsymbol{R}(\lambda)\hat{m}\mathbb{E}_{\theta_\star}\left[\theta_\star^2\right] = \boldsymbol{R}(\lambda)\hat{m}\rho \tag{88}$$

$$\boldsymbol{Q} = \mathbb{E}_{\theta_\star, \boldsymbol{\xi}}\left[\boldsymbol{R}(\lambda)\left(\hat{m}\theta_\star + \sqrt{\hat{Q}}\boldsymbol{\xi}\right)\left(\hat{m}\theta_\star + \sqrt{\hat{Q}}\boldsymbol{\xi}\right)^\top \boldsymbol{R}(\lambda)^\top\right] = \boldsymbol{R}(\lambda)\left(\rho\hat{m}\hat{m}^\top + \hat{Q}\right)\boldsymbol{R}(\lambda)^\top \tag{89}$$

$$\boldsymbol{V} = \mathbb{E}_{\theta_\star, \boldsymbol{\xi}}\left[\boldsymbol{R}(\lambda)\right] = \boldsymbol{R}(\lambda). \tag{90}$$

In order to compute the other overlaps, we must first evaluate $\mathcal{Z}_0(y, \mu, v) \equiv \int \mathrm{d}z \mathcal{N}(z|\mu, v)p(y|z)$. Since $p(y|z) = \mathcal{N}(y|z, \Delta)$ for ridge regression, $\mathcal{Z}_0(y, \mu, v)$ is simply the convolution of $\mathcal{N}(y|0, \Delta)$ and $\mathcal{N}(y|\mu, v)$, from which we can conclude $\mathcal{Z}_0(y, \mu, v)$ is equal to the density of $\mathcal{N}(0, \Delta) + \mathcal{N}(\mu, v) = \mathcal{N}(\mu_\star(\boldsymbol{\omega}), v_\star + \Delta)$. Hence, $\mathcal{Z}_0(y, \mu, v) = \mathcal{N}(y|\mu, v + \Delta)$, and we also find that $\partial_\mu \mathcal{Z}_0(y, \mu, v) = \frac{y-\mu}{v+\Delta}\mathcal{N}(y|\mu, v + \Delta)$. Defining $\boldsymbol{G}(\boldsymbol{p}) \equiv (\boldsymbol{I}_2 + \boldsymbol{PV})^{-1}\boldsymbol{P}$ with $\boldsymbol{P} = \text{Diag}(\boldsymbol{p})$, the overlaps are given by

$$\hat{\boldsymbol{m}} = \alpha\mathbb{E}_{\boldsymbol{\omega}, \boldsymbol{p}}\left[\int \mathrm{d}y \mathcal{N}(y|\mu_\star(\boldsymbol{\omega}), v_\star + \Delta)\frac{y - \mu_\star(\boldsymbol{\omega})}{v_\star + \Delta}\boldsymbol{G}(\boldsymbol{p})(\mathbf{1}_B y - \boldsymbol{\omega})\right] \tag{91}$$

$$= \alpha\mathbb{E}_{\boldsymbol{p}}[\boldsymbol{G}(\boldsymbol{p})]\mathbb{E}_{\boldsymbol{\omega}}\left[\int \mathrm{d}y \mathcal{N}(y|\mu_\star(\boldsymbol{\omega}), v_\star + \Delta)\left(\mathbf{1}_B\frac{y^2}{v_\star + \Delta} - \mathbf{1}_B\frac{y\mu_\star(\boldsymbol{\omega})}{v_\star + \Delta} - \frac{y - \mu_\star(\boldsymbol{\omega})}{v_\star + \Delta}\boldsymbol{\omega}\right)\right] \tag{92}$$

$$= \alpha\mathbb{E}_{\boldsymbol{p}}[\boldsymbol{G}(\boldsymbol{p})]\mathbb{E}_{\boldsymbol{\omega}}\left[\left(\mathbf{1}_B\frac{v_\star + \Delta + \mu_\star(\boldsymbol{\omega})^2}{v_\star + \Delta} - \mathbf{1}_B\frac{\mu_\star(\boldsymbol{\omega})^2}{v_\star + \Delta}\right)\right] \tag{93}$$

$$= \alpha\mathbb{E}_{\boldsymbol{p}}[\boldsymbol{G}(\boldsymbol{p})]\mathbf{1}_B \tag{94}$$

$$\hat{\boldsymbol{Q}} = \alpha\mathbb{E}_{\boldsymbol{\omega}, \boldsymbol{p}}\left[\int \mathrm{d}y \mathcal{N}(y|\mu_\star(\boldsymbol{\omega}), v_\star + \Delta)\boldsymbol{G}(\boldsymbol{p})(\mathbf{1}_B y - \boldsymbol{\omega})(\mathbf{1}_B y - \boldsymbol{\omega})^\top \boldsymbol{G}(\boldsymbol{p})^\top\right] \tag{95}$$

$$= \alpha\mathbb{E}_{\boldsymbol{p}}\left[\boldsymbol{G}(\boldsymbol{p})\mathbb{E}_{\boldsymbol{\omega}}\left[\mathbf{1}_{B \times B}(v_\star + \Delta + \mu_\star(\boldsymbol{\omega})^2) - \mathbf{1}_B\mu_\star(\boldsymbol{\omega})\boldsymbol{\omega}^\top - \boldsymbol{\omega}\mathbf{1}_B^\top\mu_\star(\boldsymbol{\omega}) + \boldsymbol{\omega}\boldsymbol{\omega}^\top\right]\boldsymbol{G}(\boldsymbol{p})^\top\right] \tag{96}$$

$$= \alpha\mathbb{E}_{\boldsymbol{p}}\left[\boldsymbol{G}(\boldsymbol{p})\left(\mathbf{1}_{B \times B}(v_\star + \Delta + \boldsymbol{m}^\top \boldsymbol{Q}^{-1}\boldsymbol{m}) - \boldsymbol{m}\mathbf{1}_B^\top - \mathbf{1}_B\boldsymbol{m}^\top + \boldsymbol{Q}\right)\boldsymbol{G}(\boldsymbol{p})^\top\right] \tag{97}$$

$$= \alpha\mathbb{E}_{\boldsymbol{p}}\left[\boldsymbol{G}(\boldsymbol{p})\left(\mathbf{1}_{B \times B}(v_\star + \Delta) + \boldsymbol{B}\boldsymbol{Q}\boldsymbol{B}^\top\right)\boldsymbol{G}(\boldsymbol{p})^\top\right] \tag{98}$$

$$\hat{\boldsymbol{V}} = -\alpha\mathbb{E}_{\boldsymbol{\omega}, \boldsymbol{p}}\left[\int \mathrm{d}y \mathcal{N}(y|\mu_\star(\boldsymbol{\omega}), v_\star + \Delta)(-\boldsymbol{G}(\boldsymbol{p}))\right] = \alpha\mathbb{E}_{\boldsymbol{p}}[\boldsymbol{G}(\boldsymbol{p})], \tag{99}$$

where $\boldsymbol{B} = \mathbf{1}_B\boldsymbol{m}^\top \boldsymbol{Q}^{-1} - \boldsymbol{I}_B$ in Equation (98).

### A.2.1 Summary

Overall, the closed-form expressions for the state-evolution for ridge regression are

$$
\begin{cases}
\hat{\boldsymbol{m}} &= \alpha \mathbb{E}_{\boldsymbol{p}} \left[ \boldsymbol{G}(\boldsymbol{p}) \right] \mathbf{1}_B \\
\hat{\boldsymbol{Q}} &= \alpha \mathbb{E}_{\boldsymbol{p}} \left[ \boldsymbol{G}(\boldsymbol{p}) \left( (v_\star + \Delta) \, \mathbf{1}_{B \times B} + \boldsymbol{B} \boldsymbol{Q} \boldsymbol{B}^\top \right) \boldsymbol{G}(\boldsymbol{p})^\top \right] \\
\hat{\boldsymbol{V}} &= \alpha \mathbb{E}_{\boldsymbol{p}} \left[ \boldsymbol{G}(\boldsymbol{p}) \right]
\end{cases}
,
\begin{cases}
\boldsymbol{m} &= \rho \boldsymbol{R}(\lambda) \hat{\boldsymbol{m}} \\
\boldsymbol{Q} &= \boldsymbol{R}(\lambda) \left( \rho \hat{\boldsymbol{m}} \hat{\boldsymbol{m}}^\top + \hat{\boldsymbol{Q}} \right) \boldsymbol{R}(\lambda)^\top \\
\boldsymbol{V} &= \boldsymbol{R}(\lambda)
\end{cases}
\tag{100}
$$

with $\boldsymbol{G}(\boldsymbol{p}) = (\boldsymbol{I}_B + \boldsymbol{P}\boldsymbol{V})^{-1} \boldsymbol{P}$, $\boldsymbol{P} = \mathrm{Diag}(\boldsymbol{p})$, $\boldsymbol{B} = \mathbf{1}_B \boldsymbol{m}^\top \boldsymbol{Q}^{-1} - \boldsymbol{I}_B$, and $\boldsymbol{R}(\lambda) = \left( \lambda \boldsymbol{I}_B + \hat{\boldsymbol{V}} \right)^{-1}$, and $v_\star = \rho - \boldsymbol{m}^\top \boldsymbol{Q}^{-1} \boldsymbol{m}$.

# B DERIVATION OF THE RESULTS FOR RESIDUAL RESAMPLING

As for pair resampling, one can consider the state-evolution equations of a well-chosen AMP algorithm to compute the conditional bias / variance and the bias and variance of residual bootstrap. Indeed, as for pair resampling, we leverage the fact that the conditional bias and variance, together with the estimates by residual bootstrap, can be written in terms of correlations between estimators trained on different resampled datasets $\mathcal{D}_b^\star$ with same covariates $\boldsymbol{X}$ but resampled labels $y^\star$. Introducing an augmented dataset $\tilde{\mathcal{D}} = (\boldsymbol{x}_i, \boldsymbol{y}_i^\star = (y_{b,i}^\star)_{b=1}^B)_{i=1}^n$ where the labels are now $B$-dimensional vectors comprised of the resampled labels, we see that Equation (26) is mathematically equivalent to the following minimization problem

$$(\hat{\boldsymbol{\theta}})_{b=1}^B = \arg\min_{\boldsymbol{\theta}_1,\ldots,\boldsymbol{\theta}_B \in \mathbb{R}^d} \sum_{b=1}^B \sum_{i=1}^n -\log p(y_{b,i}^\star | \boldsymbol{\theta}_b^\top \boldsymbol{x}_i) + \frac{\lambda}{2}\|\boldsymbol{\theta}_b\|^2 \tag{101}$$

While Equation (101) is equivalent Equation (26), formulating it as a joint minimization over $B$ estimators allow us to solve it using a specific AMP algorithm. As for pair resampling, the state-evolution equations of AMP will yield the correlation between two estimators $\mathbb{E}_{\mathcal{D}_b^\star, \mathcal{D}_{b'}^\star}\left[\hat{\boldsymbol{\theta}}(\mathcal{D}_b^\star)^\top \hat{\boldsymbol{\theta}}(\mathcal{D}_{b'}^\star)\right]$ in the high-dimensional limit. These correlations are sufficient to compute the true variance and its estimation with the residual bootstrap, depending on the resampling process $\mathcal{D}^\star$.

For residual bootstrap, the AMP algorithm is similar to Algorithm 1 to compute the estimators $\hat{\boldsymbol{\theta}}_i$. The main difference with Algorithm 1 is the absence of sample weights $p_i$, as all the covariates $\boldsymbol{x}_i$ are resampled only once. Equivalently, we can consider constant sample weights $p_i = 1 \; \forall i$. Moreover, the labels are now $B$-dimensional.

The overlaps can be computed using the state evolution equations (79) of Algorithm 1, where the 2-dimensional channel function is

$$\boldsymbol{g}_{\text{out}}(\boldsymbol{y}, \boldsymbol{\omega}, \boldsymbol{V}) = \arg\min_{\boldsymbol{z} \in \mathbb{R}^B} \frac{1}{2}(\boldsymbol{z} - \boldsymbol{\omega})^\top \boldsymbol{V}^{-1}(\boldsymbol{z} - \boldsymbol{\omega}) + \sum_{b=1}^B \ell(y_b, z_b) \tag{102}$$

Note that here the channel function takes a vector label as input instead of scalar label. Moreover, the channel function does not depend on any sample weight $\boldsymbol{p}$. This yields the following equations:

$$\begin{cases} \boldsymbol{m} &= \mathbb{E}_{\theta_\star, \boldsymbol{\xi}}\left[\boldsymbol{f}_a(\hat{m}\theta_\star + \sqrt{\hat{Q}}\boldsymbol{\xi}, \hat{\boldsymbol{V}})\theta_\star\right] \\ \boldsymbol{Q} &= \mathbb{E}_{\theta_\star, \boldsymbol{\xi}}\left[\boldsymbol{f}_a(\hat{m}\theta_\star + \sqrt{\hat{Q}}\boldsymbol{\xi}, \hat{\boldsymbol{V}})\boldsymbol{f}_a(\hat{m}\theta_\star + \sqrt{\hat{Q}}\boldsymbol{\xi}, \hat{\boldsymbol{V}})^\top\right] \\ \boldsymbol{V} &= \mathbb{E}_{\theta_\star, \boldsymbol{\xi}}\left[\partial_b \boldsymbol{f}_a(\hat{m}\theta_\star + \sqrt{\hat{Q}}\boldsymbol{\xi}, \hat{\boldsymbol{V}})\right] \end{cases} \tag{103}$$

with $\boldsymbol{\xi} \sim \mathcal{N}(\boldsymbol{0}, \boldsymbol{I}_B)$ and

$$\begin{cases} \hat{m} &= \alpha \mathbb{E}_{\boldsymbol{\omega}}\left[\int d\boldsymbol{y} \partial_\mu \mathcal{Z}_0(\boldsymbol{y}, \mu_\star(\boldsymbol{\omega}), v_\star) \cdot \boldsymbol{g}_{\text{out}}(\boldsymbol{y}, \boldsymbol{\omega}, \boldsymbol{V})\right] \\ \hat{Q} &= \alpha \mathbb{E}_{\boldsymbol{\omega}}\left[\int d\boldsymbol{y} \mathcal{Z}_0(\boldsymbol{y}, \mu_\star(\boldsymbol{\omega}), v_\star) \cdot \boldsymbol{g}_{\text{out}}(\boldsymbol{y}, \boldsymbol{\omega}, \boldsymbol{V})\boldsymbol{g}_{\text{out}}(\boldsymbol{y}, \boldsymbol{\omega}, \boldsymbol{V})^\top\right] , \\ \hat{V} &= -\alpha \mathbb{E}_{\boldsymbol{\omega}}\left[\int d\boldsymbol{y} \mathcal{Z}_0(\boldsymbol{y}, \mu_\star(\boldsymbol{\omega}), v_\star) \cdot \partial_\omega \boldsymbol{g}_{\text{out}}(\boldsymbol{y}, \boldsymbol{\omega}, \boldsymbol{V})\right] \end{cases} \tag{104}$$

where $\boldsymbol{\omega} \sim \mathcal{N}(\boldsymbol{0}, \boldsymbol{Q})$. Now the integrals in Equation (104) carry over vector labels $\boldsymbol{y}$ and the teacher partition $\mathcal{Z}_0$ is

$$\mathcal{Z}_0(\boldsymbol{y}, \mu, v) = \int dz \mathcal{N}(z|\mu, v) \prod_{i=1}^B p(y_i|z) \tag{105}$$

In Equations (103) and (104), $\rho$ is the squared norm $1/d\|\boldsymbol{\theta}_\star\|^2$ of the label-generating vector $\boldsymbol{\theta}_\star$. In the case of conditional resampling, $\boldsymbol{\theta}_\star = 1$ as for pair resampling. However, in the case of residual bootstrap, $\boldsymbol{\theta}_\star$ is replaced by the ERM estimator $\hat{\boldsymbol{\theta}}_\lambda$, and $\rho = 1/d\|\hat{\boldsymbol{\theta}}_\lambda\|^2$. In the high-dimensional limit, $1/d\|\hat{\boldsymbol{\theta}}_\lambda\|^2$ is obtained by running the equations (79) for full resampling, and we have $\rho = Q_{11}^{\text{fr}}$.

**Ridge regression** In the Ridge regression case, the state-evolution equations are given by

$$\begin{cases} \hat{m} &= \alpha \boldsymbol{G} \boldsymbol{1}_B \\ \hat{Q} &= \alpha \boldsymbol{G}\left(v_\star \boldsymbol{1}_{B\times B} + \Delta \boldsymbol{I}_B + \boldsymbol{B}\boldsymbol{Q}\boldsymbol{B}^\top\right)\boldsymbol{G}^\top , \\ \hat{V} &= \alpha \boldsymbol{G} \end{cases} \quad \begin{cases} \boldsymbol{m} &= \rho \boldsymbol{R}(\lambda)\hat{m} \\ \boldsymbol{Q} &= \boldsymbol{R}(\lambda)\left(\rho \hat{m}\hat{m}^\top + \hat{Q}\right)\boldsymbol{R}(\lambda)^\top \\ \boldsymbol{V} &= \boldsymbol{R}(\lambda) \end{cases} \tag{106}$$

with $\boldsymbol{G} = (\boldsymbol{I}_B + \boldsymbol{V})^{-1}$, $\boldsymbol{B} = \boldsymbol{1}_B \boldsymbol{m}^\top \boldsymbol{Q}^{-1} - \boldsymbol{I}_B$, and $\boldsymbol{R}(\lambda) = \left(\lambda \boldsymbol{I}_B + \hat{\boldsymbol{V}}\right)^{-1}$, and $v_\star = \rho - \boldsymbol{m}^\top \boldsymbol{Q}^{-1} \boldsymbol{m}$. Note that $\Delta$ is the variance of the Gaussian noise, which will be 1 for conditional resampling but not for residual bootstrap.

## B.1 RESIDUAL BOOTSTRAP

In residual bootstrap, one uses the ERM estimator trained on the whole dataset $\mathcal{D}$ to sample new labels with fixed input data $X$. Then, to compute the asymptotic behaviour of residual bootstrap, the idea is to solve Equations (103) and (104) where $\boldsymbol{\theta}_\star$ is replaced by $\hat{\boldsymbol{\theta}}_\lambda$. Its squared norm $\|\boldsymbol{\theta}_\star\|_2^2$ will be replaced by $\|\hat{\boldsymbol{\theta}}_\lambda\|^2$ and, in the case of ridge regression, the noise variance is generally replaced by the training square-loss

$$\hat{\Delta} = \frac{1}{n} \sum_{i=1}^n \left(y_i - \hat{\boldsymbol{\theta}}_\lambda^\top \boldsymbol{x}_i\right)^2 \tag{107}$$

Note that $\hat{\Delta}$ will typically underestimate $\Delta$ as $\hat{\boldsymbol{\theta}}_\lambda$ is correlated to $\boldsymbol{x}_i$. In practice, to compute the asymptotics of residual bootstrap, we first run the state-evolution equations to compute the (scalar) overlaps $m^{\mathrm{fr}}, Q^{\mathrm{fr}}, V^{\mathrm{fr}}$ for the ERM estimator. We then plug these overlaps in Equations (103) and (104), yielding new update equations for $\hat{m}, \hat{\boldsymbol{Q}}, \hat{\boldsymbol{V}}$:

$$\begin{cases} \hat{m} &= \alpha \mathbb{E}_{\boldsymbol{\omega}} \left[ \int d\boldsymbol{y} \partial_\omega \mathcal{Z}_0(\boldsymbol{y}, \mu_\star(\boldsymbol{\omega}), \tilde{v}_\star) \cdot \boldsymbol{g}_{\mathrm{out}}(\boldsymbol{y}, \boldsymbol{\omega}, \boldsymbol{V})) \right] \\ \hat{\boldsymbol{Q}} &= \alpha \mathbb{E}_{\boldsymbol{\omega}} \left[ \int d\boldsymbol{y} \mathcal{Z}_0(\boldsymbol{y}, \mu_\star(\boldsymbol{\omega}), \tilde{v}_\star) \cdot \boldsymbol{g}_{\mathrm{out}}(y, \boldsymbol{\omega}, \boldsymbol{V}) \boldsymbol{g}_{\mathrm{out}}(y, \boldsymbol{\omega}, \boldsymbol{V})^\top \right], \\ \hat{\boldsymbol{V}} &= -\alpha \mathbb{E}_{\boldsymbol{\omega}} \left[ \int d\boldsymbol{y} \mathcal{Z}_0(\boldsymbol{y}, \mu_\star(\boldsymbol{\omega}), \tilde{v}_\star) \cdot \partial_{\boldsymbol{\omega}} \boldsymbol{g}_{\mathrm{out}}(y, \boldsymbol{\omega}, \boldsymbol{V}) \right] \end{cases} \tag{108}$$

where $\boldsymbol{\omega} \sim \mathcal{N}(\boldsymbol{0}, \boldsymbol{Q})$. Also note that here, $\tilde{v}_\star = Q_{11}^{\mathrm{fr}} - \boldsymbol{m}^\top \boldsymbol{Q}^{-1} \boldsymbol{m}$ as we replaced $\rho$ by $Q_{11}^{\mathrm{fr}}$, and for ridge regression,

$$\mathcal{Z}_0(y, \mu, v) = \int dz \mathcal{N}(y|z, \tilde{\Delta}) \mathcal{N}(z|\mu, v) = \mathcal{N}(y|\mu, \tilde{\Delta} + v) \tag{109}$$

wherein high-dimensions, the $\ell_2$ loss of $\hat{\boldsymbol{\theta}}_\lambda$ on the training set $\mathcal{D}$ is $\tilde{\Delta} = \frac{1 + \Delta - 2m_1^{\mathrm{fr}} + Q_{11}^{\mathrm{fr}}}{(1 + V_1^{\mathrm{fr}})^2}$, see [Loureiro et al., 2021] for a proof.

# C OVERLAPS AND RATES IN RIDGE REGRESSION

This section is devoted to the simplification of the system of equations in Equation (100). Indeed, while the GAMP algorithm can be run with general $B \geq 1$, we can in fact restrict ourselves to the case $B = 2$ without loss of generality. Since our main goal is to compute the correlation between various independent bootstrap resamples and the resamples are i.i.d, the overlaps will have a simple structure that does not depend on $B$. Once analytical expressions for the overlaps of interest are obtained, the rates of various quantitie like bias and variance are computed in the regime $\alpha \to \infty$.

## C.1 SOLUTION TO THE STATE-EVOLUTION EQUATIONS

Let us simplify the system of equations in Equation (100) assuming $B = 2$:

**Overlaps $V, \hat{V}$**  Note that the matrices $V$ and $\hat{V}$ are diagonal, so that we can denote them as $V = \mathrm{Diag}(v_1, v_2)$ and $\hat{V} = \mathrm{Diag}(\hat{v}_1, \hat{v}_2)$. This is due to the fact that the two estimators are independently computed. As such, combining the two equations for $V$ and $\hat{V}$ in Equation (100), one can write

$$\begin{bmatrix} v_1 & 0 \\ 0 & v_2 \end{bmatrix} = \begin{bmatrix} \frac{1}{\lambda + \alpha \mathbb{E}_{p_1}\left[\frac{p_1}{1+p_1 v_1}\right]} & 0 \\ 0 & \frac{1}{\lambda + \alpha \mathbb{E}_{p_2}\left[\frac{p_2}{1+p_2 v_2}\right]} \end{bmatrix}. \tag{110}$$

Hence for $i = 1, 2$, the overlap $v_i$ is given by the fixed-point equation

$$v_i = \frac{1}{\lambda + \alpha \mathbb{E}_{p_i}\left[\frac{p_i}{1+p_i v_i}\right]}. \tag{111}$$

Moreover, we have $\hat{v}_i = \alpha \mathbb{E}_{p_i}\left[\frac{p_i}{1+p_i v_i}\right] = \frac{1}{v_i} - \lambda$.

**Overlaps $m, \hat{m}$**  Next, we deduce $m$ by combining the $m$ and $\hat{m}$ expressions from Equation (100):

$$\begin{bmatrix} m_1 \\ m_2 \end{bmatrix} = \alpha \begin{bmatrix} \frac{\rho}{\lambda+\hat{v}_1}\mathbb{E}_{p_1}\left[\frac{p_1}{1+p_1 v_1}\right] \\ \frac{\rho}{\lambda+\hat{v}_2}\mathbb{E}_{p_2}\left[\frac{p_2}{1+p_2 v_2}\right] \end{bmatrix} = \begin{bmatrix} \frac{\rho \hat{v}_1}{\lambda+\hat{v}_1} \\ \frac{\rho \hat{v}_2}{\lambda+\hat{v}_2} \end{bmatrix}, \tag{112}$$

so that $m_i = \frac{\rho \hat{v}_i}{\lambda+\hat{v}_i} = \rho(1 - \lambda v_i)$, for $i = 1, 2$. Moreover, $\hat{m}_i = \hat{v}_i$.

**Overlaps $Q, \hat{Q}$**  One can leverage the fact that the matrices $Q, \hat{Q}$ are symmetric. Using the notation

$$Q := \begin{bmatrix} q_1 & q_{1,2} \\ q_{1,2} & q_2 \end{bmatrix}, \quad \hat{Q} := \begin{bmatrix} \hat{q}_1 & \hat{q}_{1,2} \\ \hat{q}_{1,2} & \hat{q}_2 \end{bmatrix} \quad \text{and} \quad Q^{-1} := \begin{bmatrix} q_1' & q_{1,2}' \\ q_{1,2}' & q_2' \end{bmatrix} \tag{113}$$

one can rewrite the equation for $Q$ from Equation (100) as

$$\begin{bmatrix} q_1 & q_{1,2} \\ q_{1,2} & q_2 \end{bmatrix} = \begin{bmatrix} \frac{\rho \hat{m}_1^2 + \hat{q}_1}{(\lambda+\hat{v}_1)^2} & \frac{\rho \hat{m}_1 \hat{m}_2 + \hat{q}_{1,2}}{(\lambda+\hat{v}_1)(\lambda+\hat{v}_2)} \\ \frac{\rho \hat{m}_1 \hat{m}_2 + \hat{q}_{1,2}}{(\lambda+\hat{v}_1)(\lambda+\hat{v}_2)} & \frac{\rho \hat{m}_2^2 + \hat{q}_2}{(\lambda+\hat{v}_2)^2} \end{bmatrix} \iff \begin{cases} q_i = \frac{\rho \hat{m}_i^2 + \hat{q}_i}{(\lambda+\hat{v}_i)^2} = \frac{1}{\rho}m_i^2 + v_i^2 \hat{q}_i, & \text{for } i = 1, 2 \\ q_{1,2} = \frac{\rho \hat{m}_1 \hat{m}_2 + \hat{q}_{1,2}}{(\lambda+\hat{v}_1)(\lambda+\hat{v}_2)} = \frac{1}{\rho}m_1 m_2 + v_1 v_2 \hat{q}_{1,2} \end{cases}. \tag{114}$$

The computations are slightly more involved for $\hat{Q}$, but one can derive that

$$BQB^\top = (m_1^2 q_1' + 2m_1 m_2 q_{1,2}' + m_2^2 q_2')\mathbf{1}_2 + Q - \begin{bmatrix} m^\top \\ m^\top \end{bmatrix} - \begin{bmatrix} m & m \end{bmatrix} \quad \text{and} \quad v_\star = \rho - (m_1^2 q_1' + 2m_1 m_2 q_{1,2}' + m_2^2 q_2'), \tag{115}$$

and consequently the equation for $\hat{Q}$ from Equation (100) reads

$$\begin{bmatrix} \hat{q}_1 & \hat{q}_{1,2} \\ \hat{q}_{1,2} & \hat{q}_2 \end{bmatrix} = \alpha \begin{bmatrix} \mathbb{E}_{p_1}\left[\left(\frac{p_1}{1+p_1 v_1}\right)^2\right](\rho + \Delta - 2m_1 + q_1) & \mathbb{E}_{p_1,p_2}\left[\frac{p_1}{1+p_1 v_1} \cdot \frac{p_2}{1+p_2 v_2}\right](\rho + \Delta - m_1 - m_2 + q_{1,2}) \\ \mathbb{E}_{p_1,p_2}\left[\frac{p_1}{1+p_1 v_1} \cdot \frac{p_2}{1+p_2 v_2}\right](\rho + \Delta - m_1 - m_2 + q_{1,2}) & \mathbb{E}_{p_2}\left[\left(\frac{p_2}{1+p_2 v_2}\right)^2\right](\rho + \Delta - 2m_2 + q_2) \end{bmatrix}$$

(116)

$$\iff \begin{cases} \hat{q}_i = \alpha \mathbb{E}_{p_i}\left[\left(\frac{p_i}{1+p_i v_i}\right)^2\right](\rho + \Delta - 2m_i + q_i), & \text{for } i = 1, 2 \\ \hat{q}_{1,2} = \alpha \mathbb{E}_{p_1,p_2}\left[\frac{p_1}{1+p_1 v_1} \cdot \frac{p_2}{1+p_2 v_2}\right](\rho + \Delta - m_1 - m_2 + q_{1,2}) \end{cases}.$$

(117)

Combining the equations for $q_i$ and $\hat{q}_i$ just derived, one can compute $q_i$ as

$$q_i = \frac{\frac{1}{\rho}m_i^2 + \alpha \mathbb{E}_{p_i}\left[\left(\frac{p_i v_i}{1+p_i v_i}\right)^2\right](\rho + \Delta - 2m_i)}{1 - \alpha \mathbb{E}_{p_i}\left[\left(\frac{p_i v_i}{1+p_i v_i}\right)^2\right]}, \quad \text{for } i = 1, 2$$

(118)

and similarly $q_{1,2}$ is given by

$$q_{1,2} = \frac{\frac{1}{\rho}m_1 m_2 + \alpha \mathbb{E}_{p_1,p_2}\left[\frac{p_1 v_1}{1+p_1 v_1} \cdot \frac{p_2 v_2}{1+p_2 v_2}\right](\rho + \Delta - m_1 - m_2)}{1 - \alpha \mathbb{E}_{p_1,p_2}\left[\frac{p_1 v_1}{1+p_1 v_1} \cdot \frac{p_2 v_2}{1+p_2 v_2}\right]}.$$

(119)

Let us collect these results in the following proposition:

**Proposition C.1.** *Consider two ridge estimators with sampling weights specified by $p_1, p_2$. The set of self-consistent equations in Equation (100) gives a characterization of their overlaps in vector/matrix form for pair resampling. Using the notation*

$$\boldsymbol{V} = \text{Diag}(v_1, v_2), \quad \hat{\boldsymbol{V}} = \text{Diag}(\hat{v}_1, \hat{v}_2), \quad \boldsymbol{Q} = \begin{bmatrix} q_1 & q_{1,2} \\ q_{1,2} & q_2 \end{bmatrix}, \quad \hat{\boldsymbol{Q}} = \begin{bmatrix} \hat{q}_1 & \hat{q}_{1,2} \\ \hat{q}_{1,2} & \hat{q}_2 \end{bmatrix},$$

(120)

*the overlaps of interest can be simplified as follows: each $v_i$ is the unique solution to the fixed-point equation*

$$v_i = \frac{1}{\lambda + \alpha \mathbb{E}_{p_i}\left[\frac{p_i}{1+p_i v_i}\right]},$$

(121)

*while*

$$m_i = \rho(1 - \lambda v_i),$$

(122)

$$q_i = \frac{\frac{1}{\rho}m_i^2 + \alpha \mathbb{E}_{p_i}\left[\left(\frac{p_i v_i}{1+p_i v_i}\right)^2\right](\rho + \Delta - 2m_i)}{1 - \alpha \mathbb{E}_{p_i}\left[\left(\frac{p_i v_i}{1+p_i v_i}\right)^2\right]},$$

(123)

$$q_{1,2} = \frac{\frac{1}{\rho}m_1 m_2 + \alpha \mathbb{E}_{p_1,p_2}\left[\frac{p_1 v_1}{1+p_1 v_1} \cdot \frac{p_2 v_2}{1+p_2 v_2}\right](\rho + \Delta - m_1 - m_2)}{1 - \alpha \mathbb{E}_{p_1,p_2}\left[\frac{p_1 v_1}{1+p_1 v_1} \cdot \frac{p_2 v_2}{1+p_2 v_2}\right]},$$

(124)

*where $\rho = {}^1/d\|\boldsymbol{\theta}_\star\|_2^2$ and $\Delta > 0$.*

*Remark C.2.* When $p_1$ and $p_2$ are identically distributed according to some distribution $\mu$, we get $v_1 = v_2 \equiv v$, $m_1 = m_2 \equiv m$, and $q_1 = q_2 \equiv q$, with

$$\begin{cases} v &= \frac{1}{\lambda + \alpha \mathbb{E}_p\left[\frac{p}{1+pv}\right]} \\ m &= \rho(1 - \lambda v) \\ q &= \frac{\frac{1}{\rho}m^2 + \alpha \mathbb{E}_p\left[\left(\frac{pv}{1+pv}\right)^2\right](\rho + \Delta - 2m)}{1 - \alpha \mathbb{E}_p\left[\left(\frac{pv}{1+pv}\right)^2\right]}, \end{cases}$$

(125)

where $p$ is a random variable distributed according to $\mu$.

*Remark* C.3. When $p_1, p_2$ are independent, the overlap $q_{12}$ can be simplified to

$$q_{1,2} = \frac{\frac{1}{\rho}m_1 m_2 + \alpha \mathbb{E}_{p_1}\left[\frac{p_1 v_1}{1 + p_1 v_1}\right] \cdot \mathbb{E}_{p_2}\left[\frac{p_2 v_2}{1 + p_2 v_2}\right](\rho + \Delta - m_1 - m_2)}{1 - \alpha \mathbb{E}_{p_1}\left[\frac{p_1 v_1}{1 + p_1 v_1}\right] \cdot \mathbb{E}_{p_2}\left[\frac{p_2 v_2}{1 + p_2 v_2}\right]} = \frac{m_1 m_2(\alpha \rho + \rho + \Delta - m_1 - m_2)}{\alpha \rho^2 - m_1 m_2}. \tag{126}$$

**Residual Resampling** The system of equations for residual resampling in Equation (106) is almost identical to Equation (100), and in fact simpler as it does not involve expectations. Hence, following the same approach and notation as above, one can solve it to determine the overlaps of interests.

**Proposition C.4.** *Consider two ridge estimators. The set of self-consistent equations in Equation* (106) *gives a characterization of their overlaps in vector/matrix form for residual resampling. Using the notation*

$$\boldsymbol{V} = \mathrm{Diag}(v_1, v_2), \quad \hat{\boldsymbol{V}} = \mathrm{Diag}(\hat{v}_1, \hat{v}_2), \quad \boldsymbol{Q} = \begin{bmatrix} q_1 & q_{1,2} \\ q_{1,2} & q_2 \end{bmatrix}, \quad \hat{\boldsymbol{Q}} = \begin{bmatrix} \hat{q}_1 & \hat{q}_{1,2} \\ \hat{q}_{1,2} & \hat{q}_2 \end{bmatrix}, \tag{127}$$

*the overlaps of interest are such that $v \equiv v_1 = v_2$, $m \equiv m_1 = m_2$, $q \equiv q_1 = q_2$. In particular, $v$ is the unique solution to the fixed-point equation*

$$v = \frac{1}{\lambda + \frac{\alpha}{1+v}}, \tag{128}$$

*while*

$$m = \rho(1 - \lambda v), \tag{129}$$

$$q = \frac{\frac{1}{\rho}m^2 + \alpha\left(\frac{v}{1+v}\right)^2(\rho + \Delta - 2m)}{1 - \alpha\left(\frac{v}{1+v}\right)^2} = \frac{m^2(\alpha\rho + \rho + \Delta - 2m)}{\alpha\rho^2 - m^2}, \tag{130}$$

$$q_{1,2} = \frac{\frac{1}{\rho}m^2 + \alpha\left(\frac{v}{1+v}\right)^2(\rho - 2m)}{1 - \alpha\left(\frac{v}{1+v}\right)^2} = \frac{m^2(\alpha\rho + \rho - 2m)}{\alpha\rho^2 - m^2}, \tag{131}$$

*where $\rho = 1/d\|\boldsymbol{\theta}_\star\|_2^2$ and $\Delta > 0$.*

## C.1.1 Full Resampling Overlaps

To compute overlaps between two independent learners performing ERM on their own dataset, we consider a single dataset of size $2n$ split evenly between the learners. This is achieved by using sampling weights $p_1, p_2$ with joint distribution given by $\mu(p_1, p_2) = \frac{1}{2}\mathbb{1}\{p_1 = 1, p_2 = 0\} + \frac{1}{2}\mathbb{1}\{p_1 = 0, p_2 = 1\}$. Since $p_1, p_2$ have the same marginals, Remark C.2 applies. Note also that here we are in the high-dimensional regime with $2n/d \to 2\alpha$. With this, the fixed-point equation for $v$ becomes $v = \frac{1}{\lambda + \frac{\alpha}{1+v}}$ and can be solved exactly. Overall, the overlaps are given by

$$\begin{cases} v &= \frac{1 - \lambda - \alpha + \sqrt{(\alpha + \lambda - 1)^2 + 4\lambda}}{2\lambda} \\ m &= \rho(1 - \lambda v) \\ q &= \frac{\frac{1}{\rho}m^2 + \alpha\left(\frac{v}{1+v}\right)^2(\rho + \Delta - 2m)}{1 - \alpha\left(\frac{v}{1+v}\right)^2} = \frac{m^2(\alpha\rho + \rho + \Delta - 2m)}{\alpha\rho^2 - m^2} \\ q_{1,2} &= \frac{m^2}{\rho} \end{cases} \tag{132}$$

by Proposition C.1. In the following, we refer to these overlaps as $v_i^{\mathrm{fr}}, m_i^{\mathrm{fr}}, q_i^{\mathrm{fr}}$ and $q_{1,2}^{\mathrm{fr}}$.

## C.1.2 Residual Resampling Overlaps

The overlaps are given by Proposition C.4:

$$\begin{cases} v &= \frac{1 - \lambda - \alpha + \sqrt{(\alpha + \lambda - 1)^2 + 4\lambda}}{2\lambda} \\ m &= \rho(1 - \lambda v) \\ q &= \frac{m^2(\alpha\rho + \rho + \Delta - 2m)}{\alpha\rho^2 - m^2} \\ q_{1,2} &= \frac{m^2(\alpha\rho + \rho - 2m)}{\alpha\rho^2 - m^2} \end{cases} \tag{133}$$

In the following, we refer to these overlaps as $v_i^{\text{rr}}, m_i^{\text{rr}}, q_i^{\text{rr}}$ and $q_{1,2}^{\text{rr}}$.

### C.1.3 Subsampling Overlaps

To compute overlaps between two independent learners that perform subsampling at rate $r_1, r_2$ of the same dataset, we must consider $p_1 \sim \text{Bern}(r_1)$ and $p_2 \sim \text{Bern}(r_2)$ with $p_1$ independent of $p_2$. The fixed-point equations for $v_i$ become $v_i = \frac{1}{\lambda + \frac{\alpha r_i}{1+v_i}}$ and can be solved exactly to yield $v_i = \frac{1-\lambda-\alpha r_i + \sqrt{(\alpha r_i + \lambda - 1)^2 + 4\lambda}}{2\lambda}$ for $i = 1, 2$. Note also that Remark C.3 applies here. By Proposition C.1, we get

$$
\begin{cases}
v_i &= \frac{1-\lambda-\alpha r_i + \sqrt{(\alpha r_i + \lambda - 1)^2 + 4\lambda}}{2\lambda} \\
m_i &= \rho(1 - \lambda v_i) \\
q_i &= \frac{\frac{1}{\rho}m_i^2 + \alpha r_i \left(\frac{v_i}{1+v_i}\right)^2 (\rho + \Delta - 2m)}{1 - \alpha r_i \left(\frac{v}{1+v}\right)^2} = \frac{m_i^2(\alpha \rho r_i + \rho + \Delta - 2m_i)}{\alpha \rho^2 r_i - m_i^2} \\
q_{1,2} &= \frac{m_1 m_2(\alpha \rho + \rho + \Delta - m_1 - m_2)}{\alpha \rho^2 - m_1 m_2},
\end{cases}
\tag{134}
$$

for $i = 1, 2$. In the following, we refer to these overlaps as $v_i^{\text{ss}}, m_i^{\text{ss}}, q_i^{\text{ss}}$ and $q_{1,2}^{\text{ss}}$.

### C.1.4 Pairs Bootstrap Overlaps

To compute overlaps between two independent learners that perform pairs bootstrap resampling of the same dataset, we must consider $p_1, p_2 \overset{\text{i.i.d.}}{\sim} \text{Poi}(1)$, so that Remark C.2 and Remark C.3 apply. By Proposition C.1, the overlaps are thus given by

$$
\begin{cases}
v &= \frac{1}{\lambda + \alpha \mathbb{E}_p\left[\frac{p}{1+pv}\right]} \\
m &= \rho(1 - \lambda v) \\
q &= \frac{\frac{1}{\rho}m^2 + \alpha \mathbb{E}_p\left[\left(\frac{pv}{1+pv}\right)^2\right](\rho + \Delta - 2m)}{1 - \alpha \mathbb{E}_p\left[\left(\frac{pv}{1+pv}\right)^2\right]} \\
q_{1,2} &= \frac{m^2(\alpha \rho + \rho + \Delta - 2m)}{\alpha \rho^2 - m^2},
\end{cases}
\tag{135}
$$

with $p \sim \text{Poi}(1)$.

*Remark* C.5. For $\lambda > 0$, the variance is thus equal to

$$
\widehat{\text{Var}}_{\text{pb}} = q - q_{1,2} = \frac{\frac{1}{\rho}m^2 + \alpha \mathbb{E}_p\left[\left(\frac{pv}{1+pv}\right)^2\right](\rho + \Delta - 2m)}{1 - \alpha \mathbb{E}_p\left[\left(\frac{pv}{1+pv}\right)^2\right]} - \frac{m^2(\alpha \rho + \rho + \Delta - 2m)}{\alpha \rho^2 - m^2},
\tag{136}
$$

with $v$ and $m$ defined in Equation (135). Setting $\lambda = 0$ (which only makes sense for $\alpha > 1$), the variance becomes

$$
\widehat{\text{Var}}_{\text{pb}} = \frac{\rho + \alpha \mathbb{E}_p\left[\left(\frac{pv}{1+pv}\right)^2\right](\Delta - \rho)}{1 - \alpha \mathbb{E}_p\left[\left(\frac{pv}{1+pv}\right)^2\right]} - \frac{\alpha \rho - \rho + \Delta}{\alpha - 1}
\tag{137}
$$

$$
= \Delta \left( \frac{\alpha \mathbb{E}_p\left[\left(\frac{pv}{1+pv}\right)^2\right]}{1 - \alpha \mathbb{E}_p\left[\left(\frac{pv}{1+pv}\right)^2\right]} - \frac{1}{\alpha - 1} \right)
\tag{138}
$$

$$
= \Delta \left( \frac{1}{1 - \alpha \mathbb{E}_p\left[\left(\frac{pv}{1+pv}\right)^2\right]} - \frac{\alpha}{\alpha - 1} \right),
\tag{139}
$$

where $v$ is the unique solution to the fixed point equation $v = \frac{1}{\alpha \mathbb{E}_p\left[\frac{p}{1+pv}\right]}$. We thus recover Theorem 2 from Karoui and Purdom [2018] since this is equivalent to writing

$$\widehat{\mathrm{Var}_{\mathrm{pb}}} = \Delta \left( \frac{\kappa}{1 - \kappa - f(\kappa)} - \frac{1}{1 - \kappa} \right), \tag{140}$$

where $\kappa = \frac{1}{\alpha}$, $f(\kappa) := \mathbb{E}_p\left[\frac{1}{(1+pv)^2}\right]$, and $v$ is the unique solution of $\mathbb{E}_p\left[\frac{1}{1+pv}\right] = 1 - \kappa$.

In the following, we refer to the overlaps as $v_i^{\mathrm{pb}}, m_i^{\mathrm{pb}}, q_i^{\mathrm{pb}}$ and $q_{1,2}^{\mathrm{pb}}$.

### C.1.5 Residual Bootstrap Overlaps

To compute overlaps between two independent learners that perform bootstrap resampling, we follow the explanation in Appendix B.1. It states that the overlaps for the residual bootstrap are given by those of the residual resampling, with $\rho$ replaced by $\tilde{\rho} = q^{\mathrm{fr}}$ and $\Delta$ replaced by $\tilde{\Delta} = \frac{\rho + \Delta - 2m^{\mathrm{fr}} + q^{\mathrm{fr}}}{(1+v^{\mathrm{fr}})^2}$. Hence, Proposition C.4 gives

$$\begin{cases} v &= \frac{1-\lambda-\alpha+\sqrt{(\alpha+\lambda-1)^2+4\lambda}}{2\lambda} \\ m &= \tilde{\rho}(1-\lambda v) \\ q &= \frac{m^2(\alpha\tilde{\rho}+\tilde{\rho}+\tilde{\Delta}-2m)}{\alpha\tilde{\rho}^2-m^2} \\ q_{1,2} &= \frac{m^2(\alpha\tilde{\rho}+\tilde{\rho}-2m)}{\alpha\tilde{\rho}^2-m^2}. \end{cases} \tag{141}$$

In the following, we refer to these overlaps as $v_i^{\mathrm{rb}}, m_i^{\mathrm{rb}}, q_i^{\mathrm{rb}}$ and $q_{1,2}^{\mathrm{rb}}$.

### C.1.6 Overlaps between Distinct Resampling Methods

Certain quantities of interest require to compute the correlation between two estimators which use different resampling methods. In the high-dimensional regime, this corresponds to the overlap $q_{1,2}$ where the sampling weights $p_1, p_2$ are independent. In that case, Remark C.3 applies and Proposition C.1 yields

$$\begin{cases} v_i &= \frac{1}{\lambda+\alpha\mathbb{E}_{p_i}\left[\frac{p_i}{1+p_iv_i}\right]} \\ m_i &= \rho(1-\lambda v_i) \\ q_{12} &= \frac{m_1m_2(\alpha\rho+\rho+\Delta-m_1-m_2)}{\alpha\rho^2-m_1m_2}, \end{cases} \tag{142}$$

for $i = 1, 2$. In particular, the overlap between full resampling and pairs bootstrap is given by

$$q_{1,2}^{\mathrm{fr,pb}} := \frac{m^{\mathrm{fr}}m^{\mathrm{pb}}(\alpha\rho + \rho + \Delta - m^{\mathrm{fr}} - m^{\mathrm{pb}})}{\alpha\rho^2 - m^{\mathrm{fr}}m^{\mathrm{pb}}}, \tag{143}$$

the overlap between full resampling and subsampling at rate $r$ is given by

$$q_{1,2}^{\mathrm{fr,ss}} := \frac{m^{\mathrm{fr}}m^{\mathrm{ss}}(\alpha\rho + \rho + \Delta - m^{\mathrm{fr}} - m^{\mathrm{ss}})}{\alpha\rho^2 - m^{\mathrm{fr}}m^{\mathrm{ss}}}. \tag{144}$$

## C.2 LARGE $\alpha$ RATES

In this section, we compute the rates of quantities of interest (variances, biases) in the $\alpha \to \infty$ limit, which are summarized in Table 1. The approach is mathematically standard: for each overlap, we compute its series expansion at $\alpha \to \infty$ up to a desired order. Let us illustrate this with an example.

Consider the full resampling overlap $v^{\mathrm{fr}}$ computed in Appendix C.1.1:

$$v^{\mathrm{fr}} = \frac{1 - \lambda - \alpha + \sqrt{(\alpha + \lambda - 1)^2 + 4\lambda}}{2\lambda}. \tag{145}$$

To compute its series expansion at $\alpha \to \infty$, we substitute $\alpha$ with $1/\beta$ in the equation above, and then compute its Taylor series at $\beta \to 0$. Letting

$$h(\beta) := \frac{1 - \lambda - \frac{1}{\beta} + \sqrt{(\frac{1}{\beta} + \lambda - 1)^2 + 4\lambda}}{2\lambda}, \tag{146}$$

one can apply this strategy and determine the Taylor expansion up to order 2 for $v^{\text{fr}}$ by evaluating

$$\lim_{\beta \to 0} h(\beta) = \lim_{\beta \to 0} \frac{\beta(1-\lambda) - 1 + \sqrt{(\beta(\lambda-1)+1)^2 + 4\lambda\beta^2}}{2\lambda\beta} = 0 \tag{147}$$

$$\lim_{\beta \to 0} h'(\beta) = \lim_{\beta \to 0} \frac{\frac{1}{\beta^2} - \frac{((\frac{1}{\beta}+\lambda-1)\frac{1}{\beta^2})}{\sqrt{(\frac{1}{\beta}+\lambda-1)^2+4\lambda}}}{2\lambda} = 1 \tag{148}$$

$$\lim_{\beta \to 0} h''(\beta) = \lim_{\beta \to 0} \frac{-\frac{2}{\beta^3} + \frac{2(\frac{1}{\beta}+\lambda-1)}{\beta^3\sqrt{(\frac{1}{\beta}+\lambda-1)^2+4\lambda}} + \frac{1}{\beta^4\sqrt{(\frac{1}{\beta}+\lambda-1)^2+4\lambda}} - \frac{(\frac{1}{\beta}+\lambda-1)^2}{\beta^4\left((\frac{1}{\beta}+\lambda-1)^2+4\lambda\right)^{3/2}}}{2\lambda} = 2(1-\lambda), \tag{149}$$

from which we conclude that for $\beta \to 0$,

$$h(\beta) = h(\beta) + h'(\beta)\beta + \frac{1}{2}h''(\beta)\beta^2 + O(\beta^3) = \beta + (1-\lambda)\beta^2 + O(\beta^3) \tag{150}$$

or equivalently, substituting back $\alpha = 1/\beta$,

$$v^{\text{fr}} = \frac{1}{\alpha} + \frac{1-\lambda}{\alpha^2} + O\left(\frac{1}{\alpha^3}\right) \tag{151}$$

for $\alpha \to \infty$. The computation of all overlaps are carried out in the same fashion, and we use the Mathematica software [Wolfram Research] to automate these computations.

### C.2.1 Full Resampling Rates

From the overlaps computed in Appendix C.1.1, we retrieve the limiting behaviors

$$\begin{cases} v^{\text{fr}} & \stackrel{\alpha \to \infty}{\simeq} \frac{1}{\alpha} + \frac{1-\lambda}{\alpha^2} + O\left(\frac{1}{\alpha^3}\right) \\ m^{\text{fr}} & \stackrel{\alpha \to \infty}{\simeq} \rho - \frac{\rho\lambda}{\alpha} + \frac{\rho\lambda(\lambda-1)}{\alpha^2} + O\left(\frac{1}{\alpha^3}\right) \\ q^{\text{fr}} & \stackrel{\alpha \to \infty}{\simeq} \rho + \frac{\Delta-2\lambda\rho}{\alpha} + \frac{\Delta(1-2\lambda)+\rho\lambda(3\lambda-2)}{\alpha^2} + O\left(\frac{1}{\alpha^3}\right) \\ q^{\text{fr}}_{1,2} & \stackrel{\alpha \to \infty}{\simeq} \rho - \frac{2\rho\lambda}{\alpha} + \frac{\rho\lambda(3\lambda-2)}{\alpha^2} + O\left(\frac{1}{\alpha^3}\right), \end{cases} \tag{152}$$

so that the variance is given by

$$\text{Var}_{\mathcal{D}}(\hat{\boldsymbol{\theta}}_\lambda) = q^{\text{fr}} - q^{\text{fr}}_{1,2} \stackrel{\alpha \to \infty}{\simeq} \frac{\Delta}{\alpha} + O\left(\frac{1}{\alpha^2}\right) \tag{153}$$

and the bias is

$$\text{Bias}^2_{\mathcal{D}}(\hat{\boldsymbol{\theta}}_\lambda) = \rho + q^{\text{fr}}_{1,2} - 2m^{\text{fr}} \stackrel{\alpha \to \infty}{\simeq} \frac{\rho\lambda^2}{\alpha^2} + O\left(\frac{1}{\alpha^3}\right). \tag{154}$$

### C.2.2 Residual Resampling Rates

From the overlaps computed in Appendix C.1.2, we retrieve the limiting behaviors

$$\begin{cases} v^{\text{rr}} & \stackrel{\alpha \to \infty}{\simeq} \frac{1}{\alpha} + \frac{1-\lambda}{\alpha^2} + O\left(\frac{1}{\alpha^3}\right) \\ m^{\text{rr}} & \stackrel{\alpha \to \infty}{\simeq} \rho - \frac{\rho\lambda}{\alpha} + \frac{\rho\lambda(\lambda-1)}{\alpha^2} + O\left(\frac{1}{\alpha^3}\right) \\ q^{\text{rr}} & \stackrel{\alpha \to \infty}{\simeq} \rho + \frac{\Delta-2\rho\lambda}{\alpha} + \frac{\Delta(1-2\lambda)+\lambda(3\lambda-2)}{\alpha^2} + O\left(\frac{1}{\alpha^3}\right) \\ q^{\text{rr}}_{1,2} & \stackrel{\alpha \to \infty}{\simeq} \rho - \frac{2\rho\lambda}{\alpha} + \frac{\rho\lambda(3\lambda-2)}{\alpha^2} + O\left(\frac{1}{\alpha^3}\right), \end{cases} \tag{155}$$

so that the variance is given by

$$\mathrm{Var}_{\mathcal{D}|\boldsymbol{X}}(\hat{\boldsymbol{\theta}}_\lambda) = q^{\mathrm{rr}} - q^{\mathrm{rr}}_{1,2} \overset{\alpha\to\infty}{\simeq} \frac{\Delta}{\alpha} + O\left(\frac{1}{\alpha^2}\right) \tag{156}$$

and the bias is

$$\mathrm{Bias}^2_{\mathcal{D}|\boldsymbol{X}}(\hat{\boldsymbol{\theta}}_\lambda) = \rho + q^{\mathrm{rr}}_{1,2} - 2m^{\mathrm{rr}} \overset{\alpha\to\infty}{\simeq} \frac{\rho\lambda^2}{\alpha^2} + O\left(\frac{1}{\alpha^3}\right). \tag{157}$$

### C.2.3   Rates of Overlaps between Distinct Resampling Methods

From the overlaps computed in Appendix C.1.6, we retrieve the limiting behaviors

$$\begin{cases} q^{\mathrm{fr,ss}}_{1,2} & \overset{\alpha\to\infty}{\simeq} \rho + \frac{r\Delta - \rho\lambda(r+1)}{r\alpha} + \frac{r^2\Delta + \rho\lambda(\lambda + r(\lambda + (\lambda-1)r) - 1) - r\Delta\lambda(r+1)}{r^2\alpha^2} + O\left(\frac{1}{\alpha^3}\right) \\ q^{\mathrm{fr,pb}}_{1,2} & \overset{\alpha\to\infty}{\simeq} \rho + \frac{\Delta - 2\lambda\rho}{\alpha} + \frac{\Delta(1-2\lambda) + 3\rho\lambda(\lambda-1)}{\alpha^2} + O\left(\frac{1}{\alpha^3}\right). \end{cases} \tag{158}$$

### C.2.4   Subsampling and Jackknife Rates

From the overlaps computed in Appendix C.1.3, we retrieve the limiting behaviors

$$\begin{cases} v^{\mathrm{ss}}_i & \overset{\alpha\to\infty}{\simeq} \frac{1}{r_i\alpha} + \frac{1-\lambda}{r_i^2\alpha^2} + O\left(\frac{1}{\alpha^3}\right) \\ m^{\mathrm{ss}}_i & \overset{\alpha\to\infty}{\simeq} \rho - \frac{\rho\lambda}{r_i\alpha} + \frac{\rho\lambda(\lambda-1)}{r_i^2\alpha^2} + O\left(\frac{1}{\alpha^3}\right) \\ q^{\mathrm{ss}}_i & \overset{\alpha\to\infty}{\simeq} \rho + \frac{\Delta - 2\rho\lambda}{r_i\alpha} + \frac{\Delta(1-2\lambda) + \rho\lambda(3\lambda-2)}{r_i^2\alpha^2} + O\left(\frac{1}{\alpha^3}\right) \\ q^{\mathrm{ss}}_{1,2} & \overset{\alpha\to\infty}{\simeq} \rho + \frac{\Delta r_1 r_2 r - 2\rho\lambda}{r_1 r_2\alpha} + \frac{\Delta + \frac{(\lambda-1)\lambda\rho}{r_1^2} + \frac{\lambda(\lambda\rho - \Delta r_2)}{r_1 r_2} + \frac{(\lambda-1)\lambda\rho}{r_2^2} - \frac{\Delta\lambda}{r_2}}{\alpha^2} + O\left(\frac{1}{\alpha^3}\right), \end{cases} \tag{159}$$

so that the variance when subsampling at rate $r_1 = r_2 \equiv r$ is given by

$$\widehat{\mathrm{Var}_{\mathrm{ss}}} = \frac{q^{\mathrm{ss}} - q^{\mathrm{ss}}_{1,2}}{1-r} \overset{\alpha\to\infty}{\simeq} \frac{\Delta}{\alpha r} + O\left(\frac{1}{\alpha^2}\right). \tag{160}$$

and the bias is

$$\widehat{\mathrm{Bias}^2_{\mathrm{ss}}} = \frac{q^{\mathrm{ss}}_{1,2} + q^{\mathrm{fr}} - 2q^{\mathrm{fr,ss}}_{1,2}}{(1-r)^2} \overset{\alpha\to\infty}{\simeq} \frac{\rho\lambda^2}{\alpha^2 r^2} + O\left(\frac{1}{\alpha^3}\right). \tag{161}$$

The Jackknife variances and biases are computed by taking the limit $r \to 1$, and we get

$$\widehat{\mathrm{Var}_{\mathrm{jk}}} = \lim_{r\to 1} \frac{q^{\mathrm{ss}} - q^{\mathrm{ss}}_{1,2}}{1-r} \overset{\alpha\to\infty}{\simeq} \frac{\Delta}{\alpha} + O\left(\frac{1}{\alpha^2}\right). \tag{162}$$

and

$$\widehat{\mathrm{Bias}^2_{\mathrm{jk}}} = \lim_{r\to 1} \frac{q^{\mathrm{ss}}_{1,2} + q^{\mathrm{fr}} - 2q^{\mathrm{fr,ss}}_{1,2}}{(1-r)^2} \overset{\alpha\to\infty}{\simeq} \frac{\rho\lambda^2}{\alpha^2} + O\left(\frac{1}{\alpha^3}\right). \tag{163}$$

### C.2.5   Pairs Bootstrap Rates

The computation of rates in this case are less straightforward given that the overlaps depend on the evaluation of various expectations (see Appendix C.1.4). Let us consider $v^{\mathrm{pb}}$ first, which is given by the fixed-point equation

$$v^{\mathrm{pb}} = \frac{1}{\lambda + \alpha\mathbb{E}_p\left[\frac{p}{1+pv^{\mathrm{pb}}}\right]}. \tag{164}$$

We use the Ansatz that $v^{\mathrm{pb}}$ behaves as $1/\alpha$ in the $\alpha \to \infty$ limit, and hence write it as $v^{\mathrm{pb}} = \frac{\tilde{v}}{\alpha}$. Since $\frac{1}{1+x} = 1 - x + O(x^2)$ for $x \to 0^+$, we get

$$\tilde{v} = \frac{\alpha}{\lambda + \alpha\mathbb{E}_p\left[\frac{p}{1+\frac{p\tilde{v}}{\alpha}}\right]} \approx \frac{\alpha}{\lambda + \alpha\mathbb{E}_p\left[p(1 - \frac{p\tilde{v}}{\alpha})\right]} = \frac{\alpha}{\lambda + \alpha - 2\tilde{v}}. \tag{165}$$

This can be solved exactly and

$$\tilde{v} = \frac{\alpha + \lambda - \sqrt{(\alpha + \lambda)^2 - 8\alpha}}{4} \Rightarrow v^{\mathrm{pb}} = \frac{\alpha + \lambda - \sqrt{(\alpha + \lambda)^2 - 8\alpha}}{4\alpha} \overset{\alpha \to \infty}{\simeq} \frac{1}{\alpha} + \frac{2 - \lambda}{\alpha^2} + O\left(\frac{1}{\alpha^3}\right). \tag{166}$$

Overlaps $m^{\mathrm{pb}}$ and $q_{1,2}^{\mathrm{pb}}$ are thus given by

$$m^{\mathrm{pb}} \overset{\alpha \to \infty}{\simeq} \rho - \frac{\rho\lambda}{\alpha} + \frac{\rho\lambda(\lambda - 2)}{\alpha^2} + O\left(\frac{1}{\alpha^3}\right) \tag{167}$$

$$q_{1,2}^{\mathrm{pb}} \overset{\alpha \to \infty}{\simeq} \rho + \frac{\Delta - 2\rho\lambda}{\alpha} + \frac{\Delta(1 - 2\lambda) + \rho\lambda(3\lambda - 4)}{\alpha^2} + O\left(\frac{1}{\alpha^3}\right). \tag{168}$$

Overlap $q^{\mathrm{pb}}$ involves the evaluation of $\mathbb{E}_p\left[\left(\frac{pv^{\mathrm{pb}}}{1 + pv^{\mathrm{pb}}}\right)^2\right]$, which can be computed using the same approximation as in Equation (165):

$$\mathbb{E}_p\left[\left(\frac{pv^{\mathrm{pb}}}{1 + pv^{\mathrm{pb}}}\right)^2\right] \approx \mathbb{E}_p\left[\left(pv^{\mathrm{pb}}(1 - pv^{\mathrm{pb}})\right)^2\right] \tag{169}$$

$$= \mathbb{E}_p\left[(pv^{\mathrm{pb}})^2 - 2(pv^{\mathrm{pb}})^3 + (pv^{\mathrm{pb}})^4\right] \tag{170}$$

$$= 2(v^{\mathrm{pb}})^2 - 10(v^{\mathrm{pb}})^3 + 15(v^{\mathrm{pb}})^4, \tag{171}$$

where the last equality is obtained since $p \sim \mathrm{Pois}(1)$. This yields

$$q^{\mathrm{pb}} \overset{\alpha \to \infty}{\simeq} 1 + \frac{2(\Delta - \rho\lambda)}{\alpha} + \frac{2\Delta(1 - 2\lambda) + \rho\lambda(3\lambda - 4)}{\alpha^2} + O\left(\frac{1}{\alpha^3}\right). \tag{172}$$

so that the variance in the $\alpha \to \infty$ limit is thus given by

$$\widehat{\mathrm{Var}_{\mathrm{pb}}} = q^{\mathrm{pb}} - q_{1,2}^{\mathrm{pb}} \overset{\alpha \to \infty}{\simeq} \frac{\Delta}{\alpha} + O\left(\frac{1}{\alpha^2}\right) \tag{173}$$

and the bias is

$$\widehat{\mathrm{Bias}_{\mathrm{pb}}^2} = q_{1,2}^{\mathrm{pb}} + q^{\mathrm{fr}} - 2q_{1,2}^{\mathrm{fr,pb}} \overset{\alpha \to \infty}{\simeq} \frac{\rho\lambda^2}{\alpha^4} + O\left(\frac{1}{\alpha^5}\right). \tag{174}$$

### C.2.6 Residual Bootstrap Rates

From the overlaps computed in Appendix C.1.5, we retrieve the limiting behaviors

$$\begin{cases} v^{\mathrm{rb}} & \overset{\alpha \to \infty}{\simeq} \frac{1}{\alpha} + \frac{1 - \lambda}{\alpha^2} + O\left(\frac{1}{\alpha^3}\right) \\ m^{\mathrm{rb}} & \overset{\alpha \to \infty}{\simeq} \rho + \frac{\Delta - 3\rho\lambda}{\alpha} + \frac{\Delta(1 - 3\lambda) + 3\rho\lambda(2\lambda - 1)}{\alpha^2} + O\left(\frac{1}{\alpha^3}\right) \\ q^{\mathrm{rb}} & \overset{\alpha \to \infty}{\simeq} \rho + \frac{2(\Delta - 2\lambda\rho)}{\alpha} + \frac{\Delta(1 - 6\lambda) + 2\rho\lambda(5\lambda - 2)}{\alpha^2} + O\left(\frac{1}{\alpha^3}\right) \\ q_{1,2}^{\mathrm{rb}} & \overset{\alpha \to \infty}{\simeq} \rho + \frac{\Delta - 4\rho\lambda}{\alpha} + \frac{\Delta(1 - 4\lambda) + 2\rho\lambda(5\lambda - 2)}{\alpha^2} + O\left(\frac{1}{\alpha^3}\right), \end{cases} \tag{175}$$

so that the variance is

$$\widehat{\mathrm{Var}_{\mathrm{rb}}} = q^{\mathrm{rb}} - q_{1,2}^{\mathrm{rb}} \overset{\alpha \to \infty}{\simeq} \frac{\Delta}{\alpha} + O\left(\frac{1}{\alpha^2}\right) \tag{176}$$

and the bias is

$$\widehat{\mathrm{Bias}_{\mathrm{rb}}^2} = q_{1,2}^{\mathrm{rb}} + q^{\mathrm{fr}} - 2m^{\mathrm{rb}} \overset{\alpha \to \infty}{\simeq} \frac{\rho\lambda^2}{\alpha^2} + O\left(\frac{1}{\alpha^3}\right). \tag{177}$$

### C.2.7 Differences between Rates

Recall that pairs bootstrap and subsampling aim to estimate bias and variance with respect to the joint distribution $p_\theta(y, \boldsymbol{x})$, while residual bootstrap seeks to estimate the bias and variance with respect to the conditional distribution $p_\theta(y|\boldsymbol{x})$. To understand how good each estimate of the bias and variance is, we compute for each resampling method the difference between their estimate and the true value. For the variances, this results in

$$\left|\widehat{\mathrm{Var}}_{\mathrm{ss}} - \mathrm{Var}_{\mathcal{D}}(\hat{\boldsymbol{\theta}}_\lambda)\right| \stackrel{\alpha\to\infty}{\simeq} \frac{\Delta(1-r)}{\alpha r} + \frac{\Delta\left((1-2\lambda)(1-r^2)+r\right)}{\alpha^2 r^2} + O\left(\frac{1}{\alpha^3}\right)$$

$$\left|\widehat{\mathrm{Var}}_{\mathrm{jk}} - \mathrm{Var}_{\mathcal{D}}(\hat{\boldsymbol{\theta}}_\lambda)\right| \stackrel{\alpha\to\infty}{\simeq} \frac{\Delta}{\alpha^2} + O\left(\frac{1}{\alpha^3}\right)$$

$$\left|\widehat{\mathrm{Var}}_{\mathrm{pb}} - \mathrm{Var}_{\mathcal{D}}(\hat{\boldsymbol{\theta}}_\lambda)\right| \stackrel{\alpha\to\infty}{\simeq} \frac{\Delta(4\lambda+7)}{\alpha^3} + O\left(\frac{1}{\alpha^4}\right)$$

$$\left|\widehat{\mathrm{Var}}_{\mathrm{rb}} - \mathrm{Var}_{\mathcal{D}|\boldsymbol{X}}(\hat{\boldsymbol{\theta}}_\lambda)\right| \stackrel{\alpha\to\infty}{\simeq} \frac{\Delta}{\alpha^2} + O\left(\frac{1}{\alpha^3}\right)$$

while the biases are given by

$$\left|\widehat{\mathrm{Bias}^2_{\mathrm{ss}}} - \mathrm{Bias}^2_{\mathcal{D}}(\hat{\boldsymbol{\theta}}_\lambda)\right| \stackrel{\alpha\to\infty}{\simeq} \frac{\rho\lambda^2(r^2-1)}{r^2\alpha^2} + \frac{\lambda^2\left(\rho\left(2\lambda - 2(\lambda-1)r^3 - (3-2\lambda)r - 2\right) - \Delta r\right)}{r^3\alpha^3} + O\left(\frac{1}{\alpha^4}\right)$$

$$\left|\widehat{\mathrm{Bias}^2_{\mathrm{jk}}} - \mathrm{Bias}^2_{\mathcal{D}}(\hat{\boldsymbol{\theta}}_\lambda)\right| \stackrel{\alpha\to\infty}{\simeq} \frac{\lambda^2(\rho(2\lambda-3)-\Delta)}{\alpha^3} + O\left(\frac{1}{\alpha^4}\right)$$

$$\left|\widehat{\mathrm{Bias}^2_{\mathrm{pb}}} - \mathrm{Bias}^2_{\mathcal{D}}(\hat{\boldsymbol{\theta}}_\lambda)\right| \stackrel{\alpha\to\infty}{\simeq} \frac{\rho\lambda^2}{\alpha^2} + O\left(\frac{1}{\alpha^3}\right)$$

$$\left|\widehat{\mathrm{Bias}^2_{\mathrm{rb}}} - \mathrm{Bias}^2_{\mathcal{D}|\boldsymbol{X}}(\hat{\boldsymbol{\theta}}_\lambda)\right| \stackrel{\alpha\to\infty}{\simeq} \frac{\lambda^2(2\lambda\rho-\Delta)}{\alpha^3} + O\left(\frac{1}{\alpha^4}\right).$$

# D ASYMPTOTICS OF PREDICTION VARIANCE

The focus of our work is the variance of estimators with respect to the resampling of the training set. However, one can also be interested in computing the *prediction variance*, often defined as

$$\text{Var}_{\boldsymbol{x},y}\left(y - \hat{y}\left(\boldsymbol{x}\right)\right) \tag{178}$$

where now the training set is fixed, and the variance is taken with respect to the new test sample $\boldsymbol{x}, y$. In a linear model where $\hat{y} = \hat{\boldsymbol{\theta}}_\lambda^\top \boldsymbol{x}$ and in our setting defined in Equation (1), the prediction variance is equal to the test error of the ERM estimator. Indeed :

$$\text{Var}_{\boldsymbol{x},y}\left(y - \hat{y}\left(\boldsymbol{x}\right)|\mathcal{D}\right) = \mathbb{E}\left[(y - \hat{\boldsymbol{\theta}}_\lambda^\top \boldsymbol{x})^2\right] + \mathbb{E}\left[(y - \hat{\boldsymbol{\theta}}_\lambda^\top \boldsymbol{x})\right]^2 \tag{179}$$

$$= \mathbb{E}\left[(y - \hat{\boldsymbol{\theta}}_\lambda^\top \boldsymbol{x})^2\right] = \varepsilon_g \tag{180}$$

because $\mathbb{E}\left[(y - \hat{\boldsymbol{\theta}}_\lambda^\top \boldsymbol{x})\right]^2 = 0$. In the case of Ridge regression,

$$\varepsilon_g = \rho - 2m^{\text{fr}} + Q_{11}^{\text{fr}} + \sigma^2. \tag{181}$$

Note that at optimal $\lambda = \sigma^2$ ($\lambda = 1$ in our case), the performance of the ERM estimator is equal the posterior variance of the Bayes-optimal, as

$$\text{Var}_{\text{bo}} = \rho - q^{\text{bo}} \tag{182}$$

$$= \rho - 2m^{\text{bo}} + q^{\text{bo}} \tag{183}$$

$$= \rho - 2m^{\text{fr}} + Q_{11}^{\text{fr}}, \tag{184}$$

where Equation (183) follows from the *Nishimori condition* $m^{\text{bo}} = q^{\text{bo}}$, and Equation (184) is due to the fact that $\hat{\boldsymbol{\theta}}_\lambda = \mathbb{E}\left[\boldsymbol{\theta}|\mathcal{D}\right]$ for optimal $\lambda$.

# E ADDITIONAL DETAILS FOR NUMERICAL EXPERIMENTS

The state evolution equations for the resampling methods are written in the Julia language [Bezanson et al., 2017] and are available on the Github repository `https://github.com/SPOC-group/BootstrapAsymptotics` that also contains the code used to reproduce the plots. The code leverages libraries such as `NLSolvers.jl` for optimization [Mogensen and Riseth, 2018], `QuadGK.jl` and `HCubature.jl` for integration [Johnson, 2013, 2017, Genz and Malik, 1980], `MLJLinearModels.jl` for estimation of GLMs [Jul, 2023a], as well as various utilities for statistical functions [Jul, 2024b, 2023b], performance [Jul, 2024a] and plotting [Breloff, 2024]. The code to compute the posterior variance of the Bayes-optimal estimator is written in Rust and is available at `https://github.com/spoc-group/double_descent_uncertainty`. All the experiments were run on a computer with the following specifications: 16 GB RAM, Apple M1 Pro CPU.

## E.1 EFFECTS OF FINITE $B$

In Section 5, we studied the behavior of resampling methods in the limit $B \to \infty$. However, in practice $B$ is usually not very large, and the finiteness of $B$ has an impact on the estimated bias and variances. Indeed :

$$\widehat{\text{Var}} = \frac{1}{dB}\sum_{b=1}^{B}\left\|\hat{\boldsymbol{\theta}}_b - \frac{1}{B}\sum_{b=1}^{B}\hat{\boldsymbol{\theta}}_b\right\|^2 = \frac{1}{dB}\sum_{b=1}^{B}\|\hat{\boldsymbol{\theta}}_b - \mathbb{E}_{\mathcal{D}^\star}\left[\hat{\boldsymbol{\theta}}\right]\|^2 + \frac{1}{d}\|\mathbb{E}_{\mathcal{D}^\star}\left[\hat{\boldsymbol{\theta}}\right] - \frac{1}{B}\sum_{b=1}^{B}\hat{\boldsymbol{\theta}}_b\|^2$$

where second term vanishes as $B \to \infty$. Note that our framework allows us to compute the $\widehat{\text{Var}}(B)$ for a finite number of Bootstrap resamples $B$, as we get asymptotically

$$\widehat{\text{Var}}(B) = \frac{B-1}{B}\lim_{B\to\infty}\widehat{\text{Var}}$$

where $\widehat{\mathrm{Var}}$ is the variance plotted in Figure 1 and Figure 3.

Likewise, the estimator of the bias with finite $B$ can be computed and equates

$$\widehat{\mathrm{Bias}}(B) = \widehat{\mathrm{Bias}} + \frac{1}{B}\widehat{\mathrm{Var}}$$

where $\frac{1}{B}\widehat{\mathrm{Var}}$ is due to finite sampling and vanishes as $B \to \infty$. Note that the overlaps computed with our state-evolution equations allow us to compute $\widehat{\mathrm{Bias}}(B)$ at any $B$.

