# OpenReview forum: "Analysis of Bootstrap and Subsampling in High-dimensional Regularized Regression"
_auai.org/UAI/2024/Conference — UAI 2024 spotlight_

### Official Review · Reviewer_6Sqs · 2024-02-29

**Q2-1 Originality-Novelty:** 3
**Q2-2 Correctness-Technical Quality:** 3
**Q2-5 Clarity Of Writing:** 4

**Q1 Summary And Contributions:**

This paper provides a tight asymptotic analysis of the biases and variances estimated by several resampling methods. The analysis leads to three findings: (1) these methods exhibit a double-descent phenomenon; (2) they are consistent only when the ratio $n/d$ is large; (3) they are not consistent when $n/d<1$, even with an optimal regularization.

**Q2-3 Extent To Which Claims Are Supported By Evidence:**

4: Excellent: all claims are supported by very convincing evidence (in the form of comprehensive experimental evaluation, rigorous mathematical proofs, detailed (pseudo-)code, precise references, well-motivated and realistic assumptions) and the authors deliver what they promise.

**Q2-4 Reproducibility:**

4: Excellent: key resources (e.g. proofs, code, data) are available and key details (e.g. proof sketches, experimental setup) are comprehensively described for competent researchers to confidently and easily reproduce the main results.

**Q3 Main Strengths:**

- This paper is well-written.
- This paper provides a non-trivial asymptotic analysis of the bias and variance terms estimated by resampling methods for generalized linear models.
- It indicates that only when $\alpha=n/d\gg 1$, the error estimation is consistent.
- It also indicates that when $\alpha<1$, the resampling methods do not work.

**Q4 Main Weakness:**

- The authors do not fully explain how the resampling methods behave when $\alpha>1$ and $\alpha=O(1)$. Can the authors clarify this situation in the rebuttal stage?

**Q5 Detailed Comments To The Authors:**

- Can we use the derived theories to improve existing resampling methods?

**Q9 Complying With Reviewing Instructions:**

Yes

---

> ### Author Rebuttal · Authors · 2024-04-05
>
> We thank the reviewer for their comments
>
> > The authors do not fully explain how the resampling methods behave when $\alpha>1$ and $\alpha=O(1)$. Can the authors clarify this situation in the rebuttal stage?
>
> In this regime, we observe from Figure 1 that subsampling and jacknife overestimate the true bias and variance (solid red line), both at $\lambda = 10^{-2}$ and $\lambda = 1$. However, they estimate the true bias / variance more accurately than pair bootsrap and must be preferred in this regime. On the other hand, residual bootstrap will underestimate the true variance with both regularization, but estimates the true bias rather accurately. We will add this discussion of the results in this regime in Section 5.
>
> We hope we have answered the reviewer's question and would be happy to provide further details.
>
> > Can we use the derived theories to improve existing resampling methods?
>
> This is a very interesting question. Our theories can analyze a wide range of resampling methods, for instance they can be easily extended to analyze k-fold cross validation. We thus think that our teacher-student model can be used as a test bed to analyze the performance of new resampling methods, similarly to what is done in [El Karoui, Purdom 2018] where the authors design new resampling scheme based on their analysis of pair and residual bootstrap for M-estimators.

---

### Official Review · Reviewer_hSUt · 2024-03-13

**Q2-1 Originality-Novelty:** 3
**Q2-2 Correctness-Technical Quality:** 3
**Q2-5 Clarity Of Writing:** 3

**Q1 Summary And Contributions:**

The paper provides a theoretical analysis of resampling estimates of uncertainty in high-dimensional GLMs. The main results are derived from approximate message passing techniques established previously for the original (not resampled) estimators. The authors find interesting phenomena such as double-descent behavior and inconsistency in the overparametrized regime. The theoretical analysis is validated with a few numerical experiments.

**Q2-3 Extent To Which Claims Are Supported By Evidence:**

3: Good: the main claims are supported by convincing evidence (in the form of adequate experimental evaluation, proofs, (pseudo-)code, references, assumptions).

**Q2-4 Reproducibility:**

1: Poor: key details (e.g. proof sketches, experimental setup) are incomplete/unclear, or key resources (e.g. proofs, code, data) are unavailable.

**Q3 Main Strengths:**

* A theoretical analysis of resampling methods for high-dimensional GLMs is timely and interesting.
* The paper uses the modern machinery of approximate message passing techniques to tackle the problems.
* The insights gained from the analysis are intriguing.

**Q4 Main Weakness:**

* Main results are stated in a very tedious and implicit form. There are multiple key quantities that are only defined implicitly through a rather complicated set of equations. It would be great if this could be made more explicit/interpretable, at least for a special type of model/estimator.
* The "proofs" in the appendix are basically impossible to check without doing all the work myself.  I would urge the authors to at least make it somewhat self-contained by introducing the notation and core ideas/results of AMP in the appendix. You cannot expect readers o look all of that up somewhere else. Further, much of the remainder of the appendix are statements of formulas without any justification. Please provide more context, so it is clear why these formulas are true.

**Q5 Detailed Comments To The Authors:**

* What is the black line in Figure 1?
* Theorem 4.2, variance (27): Does $Q^{rb}$ really solve (17), (18)? Or is it (25), (26)?
* p.4: You mention that the pair bootstrap is asymptotically equivalent to using iid Poisson weights. I think you need to be careful with the wording here. Indeed the sequence of weight vectors converges in distribution to iid Poisson weights. However, the "error" in this convergence may still affect the asymptotics for the bootstrap. In some cases, it was shown to be negligible (Van der Vaart and Wellner, Weak convegence, 2023, Chapter 3.7), but this is not automatic.

**Q9 Complying With Reviewing Instructions:**

Yes

---

> ### Author Rebuttal · Authors · 2024-04-05
>
> We thank the reviewer for their comments. To adress the main weakness :
>
> > "Main results are stated in a very tedious and implicit form [...] "
>
> >"The "proofs" in the appendix are basically impossible to check without doing all the work myself. [ ... ] "
>
> We refer the reviewer to the common answer to all reviewers where we address these two points : we will write the equations in Section 4 in a simpler form where possible, and we will add a section dedicated to the special case of subsampling where the equations of Theorem 4.1 are simpler and more understandable. Concerning the appendices, we will add all the steps required to derive the state evolution equations and explain better the core concepts of GAMP.
>
> Concerning other comments :
>
> > "What is the black line in Figure 1?"
>
> This line corresponds to the posterior variance of the Bayes optimal estimator, defined in equation (8) and discussed in Section 5.
>
> > "Theorem 4.2, variance (27): Does $Q^{rb}$ really solve (17), (18)? Or is it (25), (26)?"
>
> $Q^{rb}$ indeed solves (25), (26), we will fix this mistake.

---

### Official Review · Reviewer_ELCh · 2024-03-14

**Q2-1 Originality-Novelty:** 3
**Q2-2 Correctness-Technical Quality:** 3
**Q2-5 Clarity Of Writing:** 3

**Q1 Summary And Contributions:**

The authors investigate resampling methods such as subsampling, bootstrap and the jacknife in high dimensions. The authors provide tight asymptotic results on the biases and variances estimated by these methods. The authors also use a new proof technique based on the Approximate Message Passing (AMP) scheme. Finally, the authors provide new insights based on their analysis: Resampling techniques face significant challenges in high-dimensional contexts. In particular, convergence rates are only good when the number of samples is sufficiently large in comparison to the number of dimensions.

**Q2-3 Extent To Which Claims Are Supported By Evidence:**

4: Excellent: all claims are supported by very convincing evidence (in the form of comprehensive experimental evaluation, rigorous mathematical proofs, detailed (pseudo-)code, precise references, well-motivated and realistic assumptions) and the authors deliver what they promise.

**Q2-4 Reproducibility:**

4: Excellent: key resources (e.g. proofs, code, data) are available and key details (e.g. proof sketches, experimental setup) are comprehensively described for competent researchers to confidently and easily reproduce the main results.

**Q3 Main Strengths:**

Studying bootstrap and subsampling in high-dimensional regularized regression seems to be an important and still open problem. The authors identify this open issue and thoroughly analyze it: They look at different generalized linear models, different resampling schemes and both frequentist and bayesian notions of bias and variance. Many parts of the paper are well written and the authors provide very detailed references to the literature. The proofs are very detailed (however because the proofs require extensive background knowledge on techniques that I am not familiar with, I have only superficially checked them).  Moreover, the authors provide new insights based on their analysis on resampling in high-dimensions (see Summary and Contributions) which seem interesting and should pave the way for further research.

**Q4 Main Weakness:**

The setting that the authors consider seems somewhat restricted: First, they only consider a fixed ratio of dimension and sample size $\alpha=n/d$, so not even convergence of $n/d$ to some $\alpha$. Second, and this is my **major concern at the moment** (this point has been sufficiently addressed in the rebuttal), the distribution of their samples looks quite restricted: The $x_i$'s are sampled from a normal distribution where the variance decreases with $d$. So for moderately large $d$, the $x_i$'s should all be approximately $0$, right? In other existing work, for example Karoui and Purdom [2018] there is at least some discussion on the distribution of the samples and why the assumptions are as they are.

**Q5 Detailed Comments To The Authors:**

My main concern at the moment is that the setting looks very restrictive:

- Major concern: The distribution of the samples looks quite restricted: The $x_i$'s are sampled from a normal distribution where the variance decreases with $d$. So for moderately large $d$, the $x_i$'s should all be approximately $0$, right? In other existing work, for example Karoui and Purdom [2018] there at least seems to be some discussion on the distribution of the samples, and if I see it correctly, the variance also does not decrease with increasing dimension $d$.
- Minor concern: The setting that $\alpha=n/d$ also looks somewhat restrictive to me. Karoui and Purdom [2018] were looking at convergence of $n/d$ to some $\alpha$, which is more general. However, I also see that the setting you consider is difficult and starting with fixed $\alpha=n/d$ is still valuable.

Moreover:
I think the presentation of the results in Section 4 can be improved. That is partly due to the mathematics being difficult, but also because:

- You use the abbreviation "fr" in equation (16) before you introduce it a page later. I think it would help if you first introduce "fr" and then write down equation (16)
- You do not give much intuition behind the terms that appear in equation (16); I think at least roughly explaining some of these terms would help. Also, some terms in equation (16) look quite similar, for example $Q^t_{11}$ and $Q^{fr}\_{11}$, I am wondering whether it is possible to just define $Q^t\_{11}$ and additionally allow $t=fr$ next to $t\in${$pb,ss,jk$}.
- I find the logical flow of Theorem 4 not optimal. You first introduce these self-consistent equations. However, at that point, the reader does not know how $Q$ and $m$ are defined. I find that particularly problematic because I would have expected some notation like $Q^t$ or $m^t$, but because this does not appear, I wonder at this point how you stack all the $Q$-elements in equation (16) in one $Q$-matrix. The relieve only comes in equations (19)-(20). But also then, one somewhat needs to guess as a reader that one either puths $Q^t_{11}$,... and so on in the $Q$-matrix, or $Q^{fr}_{11}$.
- I think you should introduce the particular product notation between $p$ and $V$ when you introduce $G(p)$.
- In equation (21), you write $Q^{t,fr}_{12}$; that however has not been defined; you have only defined $Q^{fr,t}\_{12}$. I suppose both terms are the same?
- In equation (23) you use "rb" and "rr" as abbreviations. What is "rr" standing for? I think you have not introduced that.
- In equation (24) you call the refitted estimator $\hat{\theta}_{\lambda,k}$, however in Section 2.2, you call it $\hat{\theta}\_k$ or $\hat{\theta}\_\lambda(\mathcal{D}_k^*)$. If both of the estimators are the same (which I think at the moment), why not use consistent notation?

Other detailed comments to the main paper:


- Abstract: According to the UAI-template, mathematical formulas in the abstract are to be avoided.
- Introduction: You write: " ... not only in the absolute performance of $\hat{\theta}$ ...". What do you mean by absolute performance?
- Section 2: I find it somewhat confusing that you always write $p(y|z)$ for your likelihood because you have used $Z_i$ in the introduction, and now the $z$ looks like there is some relation to $Z_i$, however, that is not really the case, right?
- Section 2: You denote the Gaussian likelihood by $p(y|z)=\mathcal{N}(z|y,\Delta)$. First, I think you should at least once introduce the notation $\mathcal{N}(z|y,\Delta)$ and particularly explain what the vertical bar means. Second, shouldn't it be $\mathcal{N}(y|z,\Delta)$ instead?
- Section 2.2: You have not introduced $B$ before you use it in equations (11) and (12).
- Last paragraph of Section 2.2: You write $Bias_k$, shouldn't it be $Bias_t$ instead?
- Footnote 1: I think you should write "Since the $\mathcal{D}_k^*$'s are independent ..." to indicate that you mean several of the $\mathcal{D}_k^*$'s and not just one.
- Equation (14): Maybe you can add that $\theta\in\mathbb{R}^d$ as you have done for the other optimization problems.
- Section 4.1. Maybe you can refer to the exact section of Karaoui and Purdom [2018] when you cite the fact about the multinomial and Poisson distribution, given that this referenced paper is quite long.
- Section 4.2: You write "The key difference is that the covariate $x_i$ remain constant ...". There is a singular-plural issue here. Either you have "the covariates $x_i$ remain ..." or "the covariate $x_i$ remains ...".
- References: You have the Karoui-references not all at the same place because you have different spellings of that name. That is confusing.

Regarding the supplement:
For me, it seemed that there are rather many typos (and other issues) in the appendix; I think the authors should do better proofreading here. For example:

- Section A: You have not introduced the abbreviation GAMP, only AMP, and that only in the main paper. I think it is good to introduce that abbreviation here again.
- Section A: You write in a paragraph regarding subsampling: "... from a Bernoulli distribution with probability $r\in(0,1]$ ...". In the main paper you had the open interval $r\in(0,1)$.
- Section A: In the paragraph that starts with "The novelty of ..." there are the following typos: "... study the performance of GAMP for several estimators" --> "...studying the performance of GAMP for several estimators. Next sentence, shouldn't it be "properties" instead of "property"?. Then, at the end of the paragraph, you also have "paralel"-->"parallel", "guaranties" --> "guarantees" and "independantly" --> "independently".
- In the next paragraph: "...coupled system of estimate." --> "...coupled system of estimates." Moreover, in your channel function, you sometimes seem to write $V$ in bold font, sometimes not.
- Before equation (48): You write: "... consider constant sample weights $p_i=1\forall i$." There could be more emptyspace here in that mathematical expression in fron of $\forall i$.
- Section C.1.6: "cooresponds" --> "corresponds".


The proofs, especially in Sections A and B require substantial background knowledge, I am not familiar with these proof techniques, and I have only superficially worked through these proofs.

### References:
- Karoui and Purdom [2018]: Noureddine El Karoui and Elizabeth Purdom. Can we trust
the bootstrap in high-dimensions? the case of linear mod-
els. Journal of Machine Learning Research, 19(5):1–66, 2018.

**Q9 Complying With Reviewing Instructions:**

Yes

---

> ### Author Rebuttal · Authors · 2024-04-05
>
> We thank the reviewer for their constructive review and exhaustive list of comments. We will make sure to integrate them in the camera-ready version of the paper, in particular those concerning inconcistencies in the notation.
>
> **To address the main weaknesses :**
>
> > they only consider a fixed ratio of dimension and sample size
>
> This assumption is made for convenience in our computations, but all of our results also hold if we assume a limit $n/d \rightarrow \alpha$ instead.
>
> > The distribution of their samples looks quite restricted
>
> In our model, the teacher weights $\theta_{\star} \sim N(0, I_d)$ are sampled from the Gaussian distribution with unit covariance. So the purpose of having the input $ {x}$ scale as $1/\sqrt{d}$ is that the local field $\theta_{\star}^T  {x}$ is of order one.
> This assumption in our model is not restrictive, and alternatively we could define our model (1) as
>
> $
> y_i \sim p( \cdot | \frac{ \theta_{\star}^T  {x}_i }{\sqrt{d}} ),  {x}_i \sim N(0, I_d)
> $
>
> where the scaling by $\sqrt{d}$ is done in the likelihood function. Moreover, we would like to point out that this is a standard assumption in the in the high-dimensional statistics litterature, see for example [1] or [2].
>
> Finally, note that our tools allow to extend our study to more complex and realistic data structures such as random features, as done in [3] or [4]. However, the paper is already dense enough, we thus leave the study of more complex data distributions to future work.
>
> **To answer additional comments :**
>
> > What do you mean by absolute performance?
>
> We refer to the generalization error of the estimator.
>
> > What is "rr" standing for? I think you have not introduced that
>
> rr refers to residual resampling (defined in section 4), we will introduce the acronym in the revised version
>
> > You do not give much intuition behind the terms that appear in equation (16)
>
> Intuitively, $m$ is the correlation between the teacher and the resample average, while the matrix $Q$ is essentially a Gram matrix between two independent resamplings. We will add this description in the camera-ready.
>
> > I am wondering whether it is possible to just define $Q^t_{11}$ and additionally allow $t = fr$ next to $t = \{ pb, ss, jk \}$
>
> The full resampling $fr$ corresponds to a resampling of the whole dataset $D$ and not to a reweighting of the $ {x}_i$, contrary to $pb, ss, jk$, which is why it is not treated the same way. It would be possible to unify $fr$ and the other resampling methods but would unnecessarily complexify the notation.
>
> > I find the logical flow of Theorem 4 not optimal [ ... ]
>
> We will add the superscript $t$ in equations (17, 18) and explain how the matrix $ {Q}$ is defined from "stacking" the overlaps defined in equation (16) :
>
> $
>  {Q}^t = \begin{pmatrix} Q_{11}^t & Q_{12}^t \\\\ Q_{12}^t & Q_{11}^t \end{pmatrix}
> $
>
> whereas
>
> $
>  {Q}^{t, fr} = \begin{pmatrix} Q_{11}^t & Q_{12}^{t,fr} \\\\ Q_{12}^{t, fr} & Q_{11}^{fr} \end{pmatrix}
> $
>
> Likewise, the vector $ {m}^t = (m_1^t, m_1^t)$ and $ {m}^{fr} = (m_1^{fr}, m_1^{fr})$.To give an intuition of the results, we will explain that the matrix $ {Q}^t$ represents the Gram matrix of two estimators trained on the same dataset with two independent resample, while $Q^{t, fr}$ is the Gram matrix of one resampled estimator and the ERM estimator trained on the full dataset, while the vector $ {m}$ contains the correlation between the estimator and the teacher $\theta_{\star}$.
>
> > The proofs, especially in Sections A and B require substantial background knowledge, I am not familiar with these proof techniques, and I have only superficially worked through these proofs.
>
> We refer the reviewer to the common answer to all the reviewers.
>
> [1] arxiv:0907.3574
>
> [2] arxiv:1803.06964
>
> [3] proceedings.mlr.press/v206/clarte23a/clarte23a.pdf
>
> [4] proceedings.mlr.press/v162/loureiro22a.html

---

### Official Review · Reviewer_zPQL · 2024-03-20

**Q2-1 Originality-Novelty:** 3
**Q2-2 Correctness-Technical Quality:** 3
**Q2-5 Clarity Of Writing:** 3

**Q10 Ethical Concerns:**

No.

**Q1 Summary And Contributions:**

The authors analyze various resampling techniques in the high-dimensional setting, where the number of features and the number of observations grow - while keeping the ratio of both at a fixed level. They analyze the bias and variance of estimators of the parameters in a regression and a classification scenario - in both cases with ridge penalization - and provide asymptotic descriptions

**Q2-3 Extent To Which Claims Are Supported By Evidence:**

3: Good: the main claims are supported by convincing evidence (in the form of adequate experimental evaluation, proofs, (pseudo-)code, references, assumptions).

**Q2-4 Reproducibility:**

3: Good: key resources (e.g. proofs, code, data) are available and key details (e.g. proofs, experimental setup) are sufficiently well-described for competent researchers to confidently reproduce the main results.

**Q3 Main Strengths:**

The paper tackles an important issue in ML. The paper is written clearly and has extensive appendices supporting their claims.

**Q4 Main Weakness:**

The appendices might be too extensive, I did not have the time to review them.

**Q5 Detailed Comments To The Authors:**

Dear authors, thank you for this submission, I liked reading your paper. Perhaps one minor comment: In the ML community, the term “regression” is often reserved for the scenario of a continuous target variable (which is confusing, I agree, since logistic regression is then a “classification” technique). At first sight, I asked myself why you do not treat the case of a binary or categorical target variable as well – until I figured out that you indeed do this. Perhaps consider changing the title and the second paragraph in the second column on the first page to avoid readers thinking you stick to y \in \R.

**Q9 Complying With Reviewing Instructions:**

Yes

---

> ### Author Rebuttal · Authors · 2024-04-05
>
> > The appendices might be too extensive
>
> Unfortunately, the length of the appendices is partially due to the nature of the mathematical tools. It is not possible not shorten the appendices, but we refer the reviewer to the common answer where we explain how we will make them clearer in the revised version.
>
> > Perhaps consider changing the title and the second paragraph in the second column on the first page to avoid readers thinking you stick to $y \in R$
>
> We agree that the use of "regression" can sometimes be confusing. As suggested, we will make it clear in the introduction that we also study classification tasks.

---

### Official Review · Reviewer_Vccu · 2024-03-27

**Q2-1 Originality-Novelty:** 4
**Q2-2 Correctness-Technical Quality:** 4
**Q2-5 Clarity Of Writing:** 4

**Q1 Summary And Contributions:**

The paper looks at the performance resampling methods for estimating uncertainty of the performance of generalized linear models in the  high dimensional setting. ​ Toward this end the authors provide the asymptotic bias and variance from different resampling schemes, and show that the schemes are incapable of providing reliable error analyses unless n/d > 1. The application of the results are given for ridge regression and logistic regression.

**Q2-3 Extent To Which Claims Are Supported By Evidence:**

3: Good: the main claims are supported by convincing evidence (in the form of adequate experimental evaluation, proofs, (pseudo-)code, references, assumptions).

**Q2-4 Reproducibility:**

3: Good: key resources (e.g. proofs, code, data) are available and key details (e.g. proofs, experimental setup) are sufficiently well-described for competent researchers to confidently reproduce the main results.

**Q3 Main Strengths:**

This paper does an excellent job of studying the ability of resampling to be used in error analysis for GLMs in the high dimensional setting. This topic is obviously very relevant to modern practice, and the results here are non-intuitive and I believe provide valuable insight into the behavior of high dimensional models. The authors do a nice job of explaining the problem setting in clear terms, and providing concise derivations of the aspects of the resampling procedure which leads to errors in the estimates. The discussion and application to ridge regression and logistic regression help with broader accessibility of the paper.

**Q4 Main Weakness:**

Overall, I think this paper is quite strong. It would have been nice to see some empirical evidence, but I don't think that substantially detracts too much from the overall quality of the paper.

**Q5 Detailed Comments To The Authors:**

As I mention above, I found this paper to be very well written, and quite interesting. The authors do a really nice job of stepping through the argument and technical details. One question I had was whether similar results hold in the case of the multiplicative bootstrap (like what is used in Wu & Sun, 2023). Also, it would improve the overall quality of the paper if some amount of empirical evidence was provided to demonstrate the findings.

**Q9 Complying With Reviewing Instructions:**

Yes

---

> ### Author Rebuttal · Authors · 2024-04-05
>
> We thank the reviewer for their comments, and their appreciation of our work.
>
> > It would have been nice to see some empirical evidence
>
>  We agree that it would be interesting to see to what extent our results hold on real datasets. However, the goal of our work is to compare the performance of resampling methods in a model where the data-generating process is known. This allows us e.g. to compare with the performance of the bayes optimal estimator (which is not possible with real data).
>
> > One question I had was whether similar results hold in the case of the multiplicative bootstrap
>
> If the reviewers refer to the paper arxiv:2309.03354, then our theories can be used to study multiplied bootstrap. Indeed, it consists of minimizing
> $
> L^k = \sum_{i = 1}^N w_{k, i} (y_i - x_i^Tb)^2
> $
> with a multiplier matrix $w_{k ,n}$. Our model can accomodate any choice of distribution of $w_{k, n}$ under the conditions that 1) the $w_{k, n}$ are identically distributed 2) for a fixed $k$, $w_{k, n}$ and $w_{k, n'}$ are independent for $n \neq n'$ and 3) for fixed $n$, all the pairs $(w_{k, n}, w_{k', n})$ for ${k \neq k'}$ follow the same distribution. Of course, the performance of bootstrap then depends on the choice of $w_{k, n}$

---

### Meta-Review · Area_Chair_xP65 · 2024-04-16

The submission provides a significant contribution to the understanding of resampling methods in high-dimensional settings. The paper offers a detailed asymptotic analysis. The paper is a strong contribution to the field. It advances our understanding of the limitations and capabilities of resampling methods in high-dimensional statistical analysis. The clarity of writing, alongside the rigor of the theoretical work, positions this paper as a valuable resource for both theoreticians and practitioners in statistics.

Pros:
+ Ground-breaking ideas in the performance of resampling methods for high-dimensional generalized linear models (GLMs).
+ Provides comprehensive theoretical analysis, including asymptotic bias and variance.
+ Results are applicable to commonly used statistical methods such as ridge regression and logistic regression.
+ High clarity of writing, with the paper being well-organized and technically sound.
+ Claims are well-supported by evidence, and the setup allows for replication and verification by other researchers.

Cons:
- The paper could have stronger empirical evidence.
- Some comments suggest that while the appendices are extensive, they may be overly complex, making it hard for readers to fully engage with the material without extensive background knowledge.


For the camera-ready version of your paper, please integrate feedback from the reviews thoroughly, especially concerning clarity and mathematical notation consistency. Make sure all mathematical terms and notations are introduced and consistently used throughout the paper.